# Inference-time optimization for experiment-grounded protein ensemble generation

**Advaith Maddipatla** [1]  **Anar Rzayev** [1]  **Marco Pegoraro** [1]
**Martin Pacesa** [2]  **Paul Schanda** [1]  **Ailie Marx** [3 4]  **Sanketh Vedula** [1 5 6 † *]  **Alex M. Bronstein** [1 7 *]

## Abstract

Protein function relies on dynamic conformational ensembles, yet current generative models like AlphaFold3 (Abramson et al., 2024) often fail to produce ensembles that match experimental data. Recent experiment-guided generators attempt to address this by steering the reverse diffusion process. However, these methods are limited by fixed sampling horizons and sensitivity to initialization, often yielding thermodynamically implausible results. We introduce a general inference-time optimization framework to solve these challenges. First, we optimize over latent representations to maximize ensemble log-likelihood, rather than perturbing structures post hoc. This approach eliminates dependence on diffusion length, removes initialization bias, and easily incorporates external constraints. Second, we present novel sampling schemes for drawing Boltzmann-weighted ensembles. By combining structural priors from AlphaFold3 with force-field-based priors, we sample from their product distribution while balancing experimental likelihoods. Our results show that this framework consistently outperforms state-of-the-art guidance, improving diversity, physical energy, and agreement with data in X-ray crystallography and NMR, often fitting the experimental data better than deposited PDB (Burley et al., 2017) structures. Finally, inference-time optimization experiments maximizing ipTM scores reveal that perturbing AlphaFold3 embeddings can artificially inflate model confidence. This exposes a vulnerability in current design metrics, whose mitigation could offer a pathway to reduce false discovery rates in binder engineering.

## 1. Introduction

Proteins are dynamic and cooperative systems that populate multiple conformational states and/or interact with diverse molecular partners. In many cases, biological function is determined not by a single structure, but by an ensemble whose state populations govern binding, catalysis, and allosteric interaction (Furnham et al., 2006). Proteins further act through transient complexes with other proteins and metabolites, making conformational heterogeneity central to molecular recognition. Hence, capturing such heterogeneity is critical to design tasks, including binder engineering, enzyme active-site optimization, and protein-protein interface design, where success depends on generating and ranking plausible alternative conformations.

Modern sequence-conditioned predictors (Jumper et al., 2021) provide strong priors over protein structure. Diffusion-based models such as AlphaFold3 (Abramson et al., 2024) (AF3) can generate high-quality structures, but often underrepresent rare functional states and can fail to fully match experimental observations, particularly for flexible parts that exist in multiple conformations (Alderson et al., 2023). This motivates methods that incorporate experimental measurements and design objectives during inference.

Recently, there has been growing attention toward guiding sequence-conditioned diffusion models with differentiable likelihood terms during inference (Dhariwal & Nichol, 2021). In experimental structure determination, these terms are derived from forward models, including NOE-based distance likelihoods for NMR, real-space density agreement for crystallographic maps, and cryo-EM likelihoods that score agreement between predicted structures and reconstructed electrostatic potential maps (Maddipatla et al., 2025a; Raghu et al., 2025).

---

[*]Equally contributing senior author. [†]Work done at the Institute of Science and Technology, Austria. [1]Institute of Science and Technology, Austria. [2]University of Zurich, Switzerland. [3]Tel Hai – University of Kiryat Shmona in the Galilee, Israel. [4]Migal – Galilee Research Institute, Israel. [5]Princeton University, NJ, USA. [6]Broad Institute of MIT and Harvard, MA, USA. [7]Technion – Israel Institute of Technology, Haifa, Israel. Correspondence to: Sanketh Vedula <svedula@princeton.edu>, Alex M. Bronstein <Alexander.Bronstein@ist.ac.at>, Advaith Maddipatla <sai.maddipatla@ist.ac.at>.

*Proceedings of the $43^{rd}$ International Conference on Machine Learning*, Seoul, South Korea. PMLR 306, 2026. Copyright 2026 by the author(s).

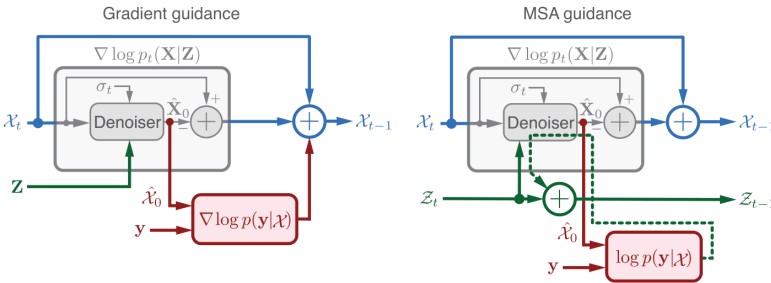

*Figure 1.* **Gradient guidance versus inference-time optimization in experiment-guided AlphaFold3.** (Left) In gradient guidance, the Pairformer conditioning variable $\mathbf{Z}$ is fixed, and experimental gradients $\nabla_{\mathcal{X}} \log p(\mathbf{y}|\mathcal{X})$ are applied directly to the coordinates $\mathcal{X}$ during reverse diffusion. (Right) In inference-time optimization (MSA Guidance), conditioning embeddings $\mathcal{Z}$ are updated using the experimental likelihood $\log p(\mathbf{y}|\mathcal{X})$, while structures are denoised via reverse diffusion conditioned on the optimized embeddings. The dotted line shows the gradient flow.

In parallel, many protein design workflows rely on objectives that are not physical measurements but are nonetheless used operationally for generation and selection. In binder and protein-protein interface design, BindCraft-like pipelines (Pacesa et al., 2025) commonly backpropagate through structure predictors to optimize losses that include model-internal confidence metrics such as interface predicted Template Modeling (ipTM) and predicted Template Modeling (pTM). These metrics are used both to guide design updates and to rank candidate complexes, making them natural conditioning objectives in practical design settings.

In current pipelines, these objectives are typically optimized using coordinate-space guidance during the denoising process. While effective, this approach has two key drawbacks. First, the guidance is tightly coupled to the denoising trajectory, making results sensitive to initialization and scheduling. This sensitivity can complicate convergence and lead to suboptimal solutions when the denoising process operates under a limited step budget. Second, coordinate-space guidance primarily focuses on enforcing agreement between the ensemble and the conditioning signal, but it does not specify how to translate a set of samples into thermodynamically meaningful states. For instance, in solution-state NMR, many conformations can satisfy the experimental restraints; however, additional modeling assumptions (e.g., energy functions) are introduced to obtain thermodynamically plausible ensemble weights. Our framework addresses this by decoupling conditioning from the specifics of the denoising schedule and incorporating energy-based reweighting to obtain thermodynamically consistent ensemble statistics when desired.

## 2. Contributions and Main Results

In what follows, we outline the main contributions of this work and summarize the key empirical results.

### 2.1. Contributions

**Inference-time optimization.** We introduce a novel inference-time optimization (IT-Optimization) framework for sequence-conditioned diffusion models that treats AF3

as a learned structural prior. We condition sampling by updating Pairformer trunk embeddings using gradients of any differentiable objective, including experiment-derived likelihoods, confidence metrics, or combinations thereof. Unlike coordinate-space guidance along a fixed denoising trajectory, representation-space updates persist across sampling runs and bias subsequent trajectories toward target-satisfying structures (see Figure 8). The framework is agnostic to the internal diffusion iterations and the underlying protein generative model (Figure 14). It composes with any modified sampling scheme, functioning as a meta-guidance layer that shapes the conditioning landscape before coordinate-level steering takes effect. Empirically, representation-space conditioning significantly and consistently outperforms coordinate-space guidance and serves as an effective upstream step for downstream coordinate-level refinement (see Figures 1, 2).[1]

**Energy-weighted sampling.** We combine the AF3 structural prior with an external force-field prior via energy-based reweighting, enabling Boltzmann-weighted ensemble statistics rather than uniform weighting of accepted samples. This preserves agreement with the conditioning signal while biasing populations toward thermodynamically plausible regions, yielding ensembles with improved energetic profiles.

### 2.2. Main results

**NMR.** We evaluate our approach on solution-state NMR structure determination, where protein conformational ensembles are inferred from nuclear Overhauser effect (NOE) measurements (Vögeli, 2014), which provide inter-proton distance restraints via cross-peaks. Using proteins from the NMRDB dataset (Klukowski et al., 2024), IT-optimization substantially reduces NOE restraint violations relative to NOE-guided AF3 (see Figure 4B, D). To obtain thermodynamically plausible conformational ensembles, we apply energy-reweighted sampling (and its IT-Optimization analogue). This further reduces restraint violations while simultaneously producing ensembles with lower effective

---

[1] AF3, AlphaFold, and AlphaFold3 are used interchangeably; IT-Opt, IT-Optimization, and Inference-time Optimization are used interchangeably.

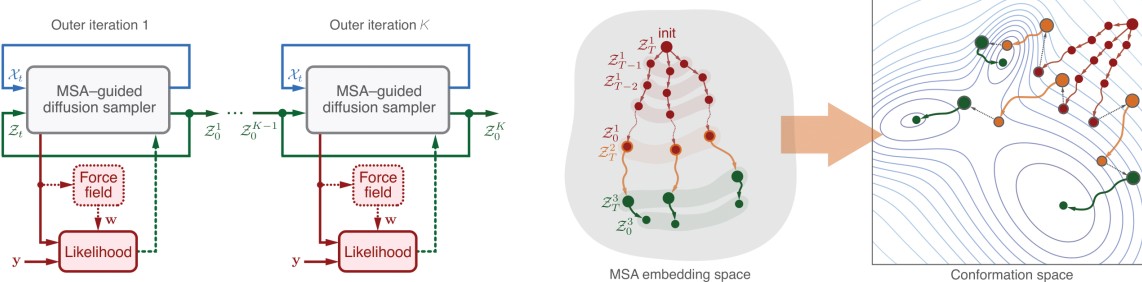

*Figure 2.* **Nested inference-time optimization of AlphaFold3 conditioning embeddings.** (Left) The outer loop runs $K$ diffusion processes, each initializing a new reverse diffusion trajectory from noise while carrying forward the optimized conditioning embeddings $\mathcal{Z}$ from the previous diffusion process. Within each diffusion process, experimental likelihood gradients update the embeddings (inner loop), which then condition subsequent denoising steps; optional force-field-based Boltzmann weights $\mathbf{w}$ bias ensemble statistics toward thermodynamically plausible conformations. (Right) Conceptually, successive diffusion trajectories explore the MSA embedding space (left panel), with embeddings refined across outer iterations. The resulting optimized embeddings induce ensembles in conformation space (right panel) that concentrate on regions consistent with experimental observations while preserving structural diversity.

energies under the AMBER99SB (Hornak et al., 2006) force field (see Figure 4B, D).

**Crystallography.** We further evaluate our method on X-ray crystallographic benchmarks comprising sequence segments that either exhibit alternative conformations (altlocs) or correspond to unrestrained short regions adopting context-dependent structures (protein-bound peptides). We observe that the proposed IT-Optimized AF3 framework consistently outperforms experiment-guided AF3 (Maddipatla et al., 2025a) across all evaluated X-ray benchmarks, achieving lower $R_{\text{work}}$ and $R_{\text{free}}$ values, improved local density alignment (see Figure 4A, C, and Figure 3), and greater reproducibility across random seeds (Figure 15), while consistently generating physically plausible structures (see Table 17). Importantly, whereas current approaches require terminal restraints to model peptides and struggle to recover widely separated altlocs, IT-optimization enables restraint-free modeling and improves recovery of separated altlocs (Figure 3).

**ipTM guidance.** We examine IT-Optimization of the interface predicted Template Modeling score (ipTM) to assess its behavior as an optimization objective for protein-protein complex prediction and to characterize the sensitivity of ipTM-directed perturbations in AF3 embedding space. We curate a benchmark comprising three representative settings: (i) complexes with low baseline ipTM scores (Wee & Wei, 2024), (ii) complexes involving transiently ordered regions of the N-terminal domain of the tumor suppressor protein p53 with different structurally characterized binding partners (Oren, 1999), and (iii) de novo protein-binder complexes from BindCraft. Across these cases, we observe that ipTM-based inference-time optimization can, in some instances, improve complex predictions where unguided AF3 performs poorly, but the effect is not uniform across systems (see Figure 7). We also find that ipTM values can be increased to high-confidence levels through very

small perturbations of the embedding space (0.01%). While such optimization is sometimes associated with improved structural agreement (see Figures 9, 11), in other cases it produces predictions with high ipTM that do not correspond to experimentally accurate interfaces, as confirmed across complementary structural metrics including interaction prediction score from aligned errors (ipSAE) (Dunbrack Jr, 2025), DockQ (Mirabello & Wallner, 2024), fraction of native contacts (Fnat), and hydrogen-bond recovery (see Figures 10 and 13). Together, these results indicate that ipTM and related confidence metrics should be interpreted cautiously when used as direct optimization objectives. They suggest that the embedding space contains regions where confidence can be increased without corresponding gains in structural accuracy, with implications for protein-binder design workflows that rely on such metrics for optimization and ranking.

## 3. Conditional ensemble generation

**Notation.** We denote the amino acid sequence of a protein by $\mathbf{a}$ and its folded three-dimensional structure by $\mathbf{X} = (\mathbf{x}_1, \ldots, \mathbf{x}_m)$, where $\mathbf{x}_i \in \mathbb{R}^3$ denotes the Cartesian coordinate of the $i$-th atom and $m$ is the total number of atoms in the structure. We further denote by $\mathbf{Z} = (\mathbf{s}, \mathbf{z})$ the conditioning variable produced by AF3's Pairformer module.[2] Here $\mathbf{s} \in \mathbb{R}^{|\mathbf{a}| \times c_s}$ and $\mathbf{z} \in \mathbb{R}^{|\mathbf{a}| \times |\mathbf{a}| \times c_z}$ denote the single and pairwise representations respectively. Together, these variables form learned continuous representations that integrate co-evolutionary information from the multiple sequence alignment (MSA) with pairwise positional and contextual features introduced by the Pairformer.

**Problem statement.** Given a protein sequence $\mathbf{a}$ and an experimental observation $\mathbf{y}$, our goal is to optimize a set of

---

[2]"MSA embeddings" and "AF3 embeddings" are used interchangeably to denote outputs of the Pairformer.

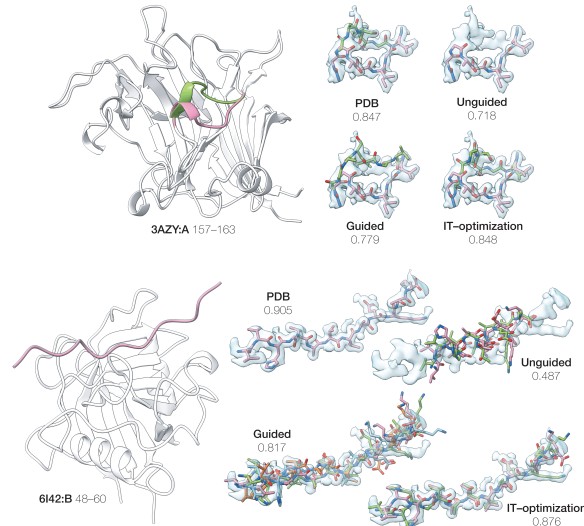

*Figure 3.* **Inference-time (IT) optimization improves structural accuracy over guided and unguided baselines.** (Top) `3AZY:A` (1.65Å) exhibits bimodal distribution at residues 157-163. Unguided AF3 predicts a single mode, while guidance produces a bimodal ensemble with a poorly fit backbone at one of the modes. IT-optimization recovers both modes with accurate density fit, matching the PDB. (Bottom) For `6I42:B` (1.38Å), AF3 mispredicts the bound 13-residue peptide. Guidance improves backbone placement but poorly predicts side-chains, whereas IT-optimization yields accurate backbone and side-chain agreement. Numbers beneath each ensemble indicate cosine similarity to $F_{\mathrm{o}}$.

$n$ conditioning variables $\boldsymbol{\mathcal{Z}} = \{\mathbf{Z}^1, \ldots, \mathbf{Z}^n\}$ that induce an ensemble of corresponding structures $\boldsymbol{\mathcal{X}} = \{\mathbf{X}^1, \ldots, \mathbf{X}^n\}$. In AF3, the conditioning variable is a deterministic transformation $\mathbf{Z}(\mathbf{a})$ of the sequence $\mathbf{a}$ produced by the Pairformer. We relax this relation into a separable prior $p(\boldsymbol{\mathcal{Z}}|\mathbf{a}) = p(\mathbf{Z}^1|\mathbf{a}) \cdots p(\mathbf{Z}^n|\mathbf{a})$ preventing $\mathbf{Z}$ from drifting arbitrarily in the embedding space and penalizing off-manifold solutions. In what follows, we assume a simple Gaussian prior $p(\mathbf{Z}|\mathbf{a}) = \mathcal{N}(\mathbf{Z}(\mathbf{a}), \sigma^2)$. Invoking Bayes' theorem, we obtain the posterior

$$p(\boldsymbol{\mathcal{Z}}|\mathbf{y}, \mathbf{a}) \propto p(\boldsymbol{\mathcal{Z}}|\mathbf{a})p(\mathbf{y}|\boldsymbol{\mathcal{Z}}, \mathbf{a}) = p(\boldsymbol{\mathcal{Z}}|\mathbf{a})p(\mathbf{y}|\boldsymbol{\mathcal{Z}}),$$

Marginalizing out $\boldsymbol{\mathcal{X}}$ yields

$$p(\mathbf{y}|\boldsymbol{\mathcal{Z}}) = \int p(\mathbf{y}|\boldsymbol{\mathcal{X}})p(\boldsymbol{\mathcal{X}}|\boldsymbol{\mathcal{Z}})d\boldsymbol{\mathcal{X}} = \mathbb{E}_{\boldsymbol{\mathcal{X}} \sim p(\boldsymbol{\mathcal{X}}|\boldsymbol{\mathcal{Z}})}p(\mathbf{y}|\boldsymbol{\mathcal{X}}),$$

where $p(\mathbf{y} \mid \boldsymbol{\mathcal{X}})$ is a generally inseparable likelihood that quantifies the agreement between the structural ensemble $\boldsymbol{\mathcal{X}}$ and the experimental observation $\mathbf{y}$. This data term is specified by forward models reflecting the physics of the experimental measurement. We consider three data terms in this work: (i) NOE distance restraints (Section 3.1); (ii) real-space crystallographic density maps (Section 3.2); and (iii) the ipTM confidence score (Section 3.3).

This leads to the exact MAP problem

$$\boldsymbol{\mathcal{Z}}^* = \arg\max_{\boldsymbol{\mathcal{Z}}} \log \mathbb{E}_{\boldsymbol{\mathcal{X}} \sim p(\boldsymbol{\mathcal{X}}|\boldsymbol{\mathcal{Z}})} p(\mathbf{y} \mid \boldsymbol{\mathcal{X}}) p(\boldsymbol{\mathcal{Z}}|\mathbf{a}). \quad (1)$$

To avoid the hardly-tractable log of the expectation, we invoke Jensen's inequality $\log \mathbb{E} p \geq \mathbb{E} \log p$ to obtain an evidence lower bound (ELBO)-type objective

$$\boldsymbol{\mathcal{Z}}^* = \arg\max_{\boldsymbol{\mathcal{Z}}} \mathbb{E}_{\boldsymbol{\mathcal{X}} \sim p(\boldsymbol{\mathcal{X}}|\boldsymbol{\mathcal{Z}})} \log p(\mathbf{y} \mid \boldsymbol{\mathcal{X}}) + \log p(\boldsymbol{\mathcal{Z}}|\mathbf{a}). \quad (2)$$

The maximization of this lower bound can be viewed as a form of variational inference with $\boldsymbol{\mathcal{Z}}$ acting as a variational parameter and the AF3 prior itself acting as a restricted family of priors, $q_{\boldsymbol{\mathcal{Z}}}(\boldsymbol{\mathcal{X}}) = p(\boldsymbol{\mathcal{X}}|\boldsymbol{\mathcal{Z}})$. The expectation is approximated by sampling from AF3.

### 3.1. Nuclear Overhauser effect restraints

Nuclear Overhauser Effect (NOE) distance restraints derived from NMR spectroscopy encode pairwise proximity constraints between non-bonded atoms $(i, j) \in \mathcal{P}$. The likelihood of observing an NOE effect between two atoms depends on their time-averaged distance, $r$, as $1/r^6$. However, as other parameters, such as local dynamics, play a role, determining precise distances, although possible, is not commonly done (Vögeli, 2014), and only lower and upper distance bounds, $[\underline{d}_{ij}, \bar{d}_{ij}]$, are used. Given an ensemble $\boldsymbol{\mathcal{X}}$ and the set of restraints $\mathbf{D} = \{(\underline{d}_{ij}, \bar{d}_{ij}) : (i, j) \in \mathcal{P}\}$, the log-likelihood is

$$\log p(\mathbf{D} \mid \boldsymbol{\mathcal{X}}) = \\ - \sum_{(i,j) \in \mathcal{P}} \left( \left[\underline{d}_{ij} - d_{ij}(\boldsymbol{\mathcal{X}})\right]_+^2 + \left[d_{ij}(\boldsymbol{\mathcal{X}}) - \bar{d}_{ij}\right]_+^2 \right) \quad (3)$$

where $[\cdot]_+ = \max(\cdot, 0)$ and $d_{ij}(\boldsymbol{\mathcal{X}}) = \frac{1}{n} \sum_{k=1}^n \|\mathbf{x}_i^{(k)} - \mathbf{x}_j^{(k)}\|$ is the ensemble-averaged interatomic distance.

### 3.2. Crystallographic electron densities

The observed real-space electron density map $F_{\mathrm{o}} : \mathbb{R}^3 \to \mathbb{R}$ represents an ensemble over the molecular conformations present in a crystal lattice. In X-ray crystallography, 2D diffraction images aggregate scattering contributions from a large population of molecules, such that the resulting electron density reflects an average over conformational heterogeneity (Hahn, 1983). Typical crystallographic pipelines attempt to fit an ensemble $\boldsymbol{\mathcal{X}}$ that best explains $F_{\mathrm{o}}$ (Murshudov et al., 2011; Agirre et al., 2023). This objective can be formulated as maximizing the likelihood $\log p(F_{\mathrm{o}} \mid \boldsymbol{\mathcal{X}})$. We define the log-likelihood as the negative $L_1$ distance between the observed and ensemble-averaged calculated density maps

$$\log p(F_{\mathrm{o}}|\boldsymbol{\mathcal{X}}) = - \int_\Omega \left\| F_{\mathrm{o}}(\boldsymbol{\xi}) - \frac{1}{n} \sum_{k=1}^n F_{\mathrm{c}}(\boldsymbol{\xi}; \mathbf{X}^k) \right\|_1 d\boldsymbol{\xi} \quad (4)$$

where, the loss is computed over the asymmetric unit cell $\Omega$ and $F_{\mathrm{c}}(\boldsymbol{\xi}; \mathbf{X}^k)$ is the theoretical electron density computed

from the atomic coordinates of structure $\mathbf{X}^k$ (See Appendix C.4 for details).

### 3.3. Interface predicted template modeling score

The interface predicted template modeling (ipTM) score, introduced with AlphaFold-Multimer and reported by AF3, is a learned confidence metric that quantifies the reliability of predicted inter-chain interfaces in multimeric protein structures (Evans et al., 2022; Abramson et al., 2024). A high ipTM prediction indicates strong model confidence in the relative placement and contacts between chains, while a low ipTM values typically correspond to poorly formed or disordered interfaces. Unlike experimental observations, ipTM is not a physical measurement; instead, it is derived from a neural network trained to estimate structural accuracy by comparing predicted interfaces against known protein complexes (Jumper et al., 2021; Abramson et al., 2024; Passaro et al., 2025). In our framework, we treat ipTM as a surrogate likelihood that provides a differentiable signal for assessing interfacial consistency. The data term can be defined as,

$$\log p(\mathbf{y} \mid \boldsymbol{\mathcal{X}}) \propto \sum_{i=1}^{n} \mathrm{ipTM}(\mathbf{X}^i) \qquad (5)$$

where $\mathbf{y}$ denotes the (unknown) true interface configuration. Unlike other likelihoods, these terms are separable. By maximizing this likelihood, we bias the ensemble toward interface geometries that are assigned higher confidence by the model, thereby promoting coherent inter-chain arrangements.

## 4. Inference-time Optimization

AF3 (Abramson et al., 2024) generates protein structure ensembles using a diffusion-based generative model (Ho et al., 2020; Karras et al., 2022) defined over the coordinates of $\mathbf{X}$. The reverse diffusion process follows the stochastic differential equation (SDE):

$$d\mathbf{X} = -\left(\frac{1}{2}\mathbf{X} + \nabla_{\mathbf{X}} \log p_t(\mathbf{X} \mid \mathbf{Z})\right) \sigma_t dt + \sqrt{\sigma_t} \mathbf{N} \quad (6)$$

where $\sigma_t$ denotes the diffusion noise schedule, $\mathbf{N} \sim \mathcal{N}(\mathbf{0}, \mathbf{I})$ is sampled from a standard normal distribution, and $\nabla_{\mathbf{X}} \log p_t(\mathbf{X} \mid \mathbf{Z})$ is the learned score function. Protein structures are generated by numerically integrating this SDE from an initial noise sample drawn from $\mathcal{N}(\mathbf{0}, \sigma_T \mathbf{I})$ at time $t = T$ down to $t = 0$. In a typical guidance setting (Maddipatla et al., 2024; 2025b;a), a non-i.i.d. ensemble $\boldsymbol{\mathcal{X}} = \{\mathbf{X}^1, \dots, \mathbf{X}^n\}$ is generated by integrating an experi-

mental likelihood term into the reverse SDE.

$$d\begin{bmatrix}\mathbf{X}^1 \\ \vdots \\ \mathbf{X}^n\end{bmatrix} = -\left(\frac{1}{2}\begin{bmatrix}\mathbf{X}^1 \\ \vdots \\ \mathbf{X}^n\end{bmatrix} + \begin{bmatrix}\nabla_{\mathbf{X}^1} \log p_t(\mathbf{X}^1|\mathbf{Z}^1) \\ \vdots \\ \nabla_{\mathbf{X}^n} \log p_t(\mathbf{X}^n|\mathbf{Z}^n)\end{bmatrix}\right)\sigma_t dt$$
$$- \eta\nabla_{\boldsymbol{\mathcal{X}}} \log p\left(\mathbf{y}|D_\theta(\boldsymbol{\mathcal{X}};\boldsymbol{\mathcal{Z}},t)\right)\sigma_t dt + \sqrt{\sigma_t}\begin{bmatrix}\mathbf{N}^1 \\ \vdots \\ \mathbf{N}^n\end{bmatrix} \quad (7)$$

Here, $\nabla_{\boldsymbol{\mathcal{X}}} \log p\left(\mathbf{y}|D_\theta(\boldsymbol{\mathcal{X}};\boldsymbol{\mathcal{Z}},t)\right)$ biases the SDE toward ensembles that maximize agreement with $\mathbf{y}$. $D_\theta(\boldsymbol{\mathcal{X}};\boldsymbol{\mathcal{Z}},t)$ denotes AF3's denoised structure prediction at diffusion step $t$, $\eta$ controls the guidance strength, and $\mathbf{N}^i \sim \mathcal{N}(\mathbf{0}, \mathbf{I})$. Despite its flexibility, the reverse diffusion process is limited: it operates with a fixed number of discretization steps, exhibits sensitivity to the initial noise realization, and its convergence rate is constrained by the noise schedule $\sigma_t$.

As a remedy, we propose a nested optimization framework consisting of an outer loop and an inner loop to solve the optimization problem in Equation 2 on a batch of embeddings $\boldsymbol{\mathcal{Z}} = \{\mathbf{Z}^1, \dots, \mathbf{Z}^n\}$ at inference-time.

**Outer loop (Exploration).** The outer loop performs a global optimization over $\boldsymbol{\mathcal{Z}}$. Due to the iterative nature of the diffusion process, we treat each outer iteration $k \in \{1, \dots, K\}$, as a fresh sampling of the diffusion noise $\boldsymbol{\mathcal{X}}_T^k = \{\mathbf{X}_T^{k,i} \sim \mathcal{N}(\mathbf{0}, \sigma_T \mathbf{I})\}_{i=1}^n$ which initializes a new diffusion trajectory. This resampling promotes exploration across distinct diffusion paths and reduces sensitivity to starting noise. Crucially, this framework drives $\boldsymbol{\mathcal{Z}}$ to generalize across noise realizations rather than overfitting to a particular diffusion trajectory. See Algorithm 1 (lines 3-5) for details and Figures 1, 2 as an illustration.

**Inner loop (Joint refinement).** Within each outer loop, we run an inner loop that simulates the reverse diffusion process from $\boldsymbol{\mathcal{X}}_T^k$ to the denoised counterpart $\boldsymbol{\mathcal{X}}_0^k$. At each reverse diffusion step $t \in [T, \dots, 0]$, we perform two coupled updates.

**Embedding updates.** We update $\boldsymbol{\mathcal{Z}}_t^k$ by doing gradient ascent on the experimental likelihood with respect to the embeddings:

$$\boldsymbol{\mathcal{Z}}_{t-1}^k = \boldsymbol{\mathcal{Z}}_t^k + \eta_z \Big(\nabla_{\boldsymbol{\mathcal{Z}}_t^k} \log p\Big(\mathbf{y} \mid D_\theta(\boldsymbol{\mathcal{X}}_t^k; \boldsymbol{\mathcal{Z}}_t^k, t)\Big)$$
$$+ \lambda_{\mathrm{p}} \nabla_{\boldsymbol{\mathcal{Z}}_t^k} \log p(\boldsymbol{\mathcal{Z}}_t^k|\mathbf{a})\Big) \qquad (8)$$

where $\eta_z$ is the learning rate and $\lambda_p$ is the prior weight. This update shifts the learned conditioning manifold toward regions of the embedding space that are more consistent with $\mathbf{y}$. Since the embeddings parameterize the structure generation process, this step indirectly biases subsequent structural samples toward experimentally faithful conformations.

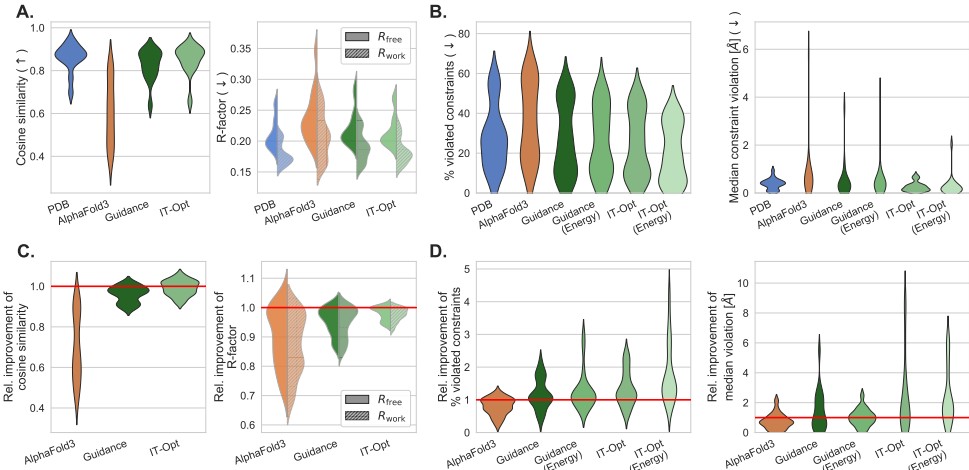

*Figure 4.* **Summary statistics for NMR and X-ray crystallography benchmarks.** Violin plots compare AlphaFold3, Guidance, and IT-Opt (with energy-reweighted variants for NMR) against deposited PDB structures. In panels **C, D**, relative improvements are computed so that *values above the red line at 1 indicate better fit than the deposited PDB.* **(A)** Distributions of $F_o$ and $F_c$ cosine similarity (left) and $R_{\text{work}}$ and $R_{\text{free}}$ (right), with relative improvements over deposited PDB structures summarized in panel **C**. See Tables 11-13 for complete results. **(B)** Distributions of the percentage of violated distance restraints (left) and median violation magnitude (right), with relative improvements summarized in panel **D**. See Tables 3, 4 for complete results.

**Reverse SDE.** The updated embeddings $\mathcal{Z}_{t-1}^k = \{\mathbf{Z}_{t-1}^{k,1}, \ldots, \mathbf{Z}_{t-1}^{k,n}\}$ are injected into the SDE, yielding the following reverse step for each ensemble member $\mathbf{X}^{k,i} \in \mathcal{X}^k$:

$$d\mathbf{X}^{k,i} = -\left(\frac{1}{2}\mathbf{X}^{k,i} + \nabla_{\mathbf{X}^{k,i}} \log p_t(\mathbf{X}^{k,i}|\mathbf{Z}_{t-1}^{k,i})\right)\sigma_t dt$$
$$+ \sqrt{\sigma_t}\mathbf{N}^i.$$

Note that the score function is predicted by the model conditioned on the updated embeddings. With this alternating procedure, the conditioning variables act as a persistent memory of the experimental manifold across diffusion steps. Importantly, the embeddings obtained at the end of the inner loop $\mathcal{Z}_0^k$ are used to initialize the next outer iteration $\mathcal{Z}_T^{k+1}$, enabling cumulative refinement rather than restarting the optimization as shown in Figures 8 and 14. See Algorithm 1, lines 6-8, for details and Figures 1, 2 as an illustration.

Our framework thus serves as a meta-initialization; in our experiments, the embeddings $\mathcal{Z}_0^K$ obtained after the convergence of the outer loops are used as conditioning variables for the coordinate-space guidance procedure in Equation 7. A detailed pseudocode is provided in Algorithm 2, with additional implementation details in Appendix C.2 and runtime analysis in Appendix C.2.1. We further study the convergence properties of inference-time updates to the diffusion model's input embeddings in Appendix B.

**Boltzmann re-weighting.** While the log-likelihood in Equation 8 uniformly weights ensemble members, physical ensembles in solution follow a Boltzmann distribution (Barducci et al., 2011). To bias the ensemble toward thermodynamically plausible conformations, we refine the AF3-

---

**Algorithm 1** IT-Optimization

1: **Input:** Initial trunk embeddings $\mathcal{Z}$; experimental observation $\mathbf{y}$; noise schedule $[\sigma_T, \ldots, \sigma_0]$; learning rate $\eta_z$; ensemble size $n$; outer iterations $K$; prior weight $\lambda_p$
2: **Output:** Optimized trunk embeddings $\mathcal{Z}$
3: **for** $k = 1$ to $K$          ▷ Outer loop
4:      $\mathcal{X} \sim \sigma_T \cdot [\mathbf{N}^1, \ldots, \mathbf{N}^n]^T$     $\mathbf{N}^i \sim \mathcal{N}(\mathbf{0}, \mathbf{I})$
5:      **for** $t \in [T-1, \ldots, 0]$       ▷ Inner loop
6:          $\mathcal{Z} \leftarrow \mathcal{Z} + \nabla_{\mathcal{Z}}(\eta_z \log p(\mathbf{y}|D_\theta(\mathcal{X}; \mathcal{Z}, t)) + \lambda_p \log p(\mathcal{Z}|\mathbf{a}))$
7:          $\mathcal{X} \leftarrow \text{ReverseSDE}(\mathcal{X}; \mathcal{Z}, t, \sigma_t)$     ▷ Eq 6
8:      **end for**
9: **end for**
10: return $\mathcal{Z}$

---

induced structural prior by tilting it with a differentiable energy function $E_\phi : \mathbb{R}^{m \times 3} \to \mathbb{R}$

$$\pi(\mathcal{X}|\mathcal{Z}) \propto p(\mathcal{X}|\mathcal{Z}) \cdot \exp(-\beta \sum_i E_\phi(\mathbf{X}^i)), \quad (9)$$

where $\beta = 1.68 \text{ kcal}^{-1} \text{ mol}$ is the inverse temperature at physiological conditions ($T = 300 \text{ K}$). This formulation interprets the force field as a thermodynamic prior that biases AF3 towards energetically favorable conformations. Because the normalizing constant of the tilted distribution $\pi$ is intractable, we adopt a self-normalized importance sampling (SNIS) perspective (Skreta et al., 2025). Expectations under $\pi$ are approximated by assigning Boltzmann weights $w^i$ to samples drawn from the AF3 prior,

$$w^i = \frac{\exp(-\beta E_\phi(\mathbf{X}^i))}{\sum_{k=1}^n \exp(-\beta E_\phi(\mathbf{X}^k))}. \quad (10)$$

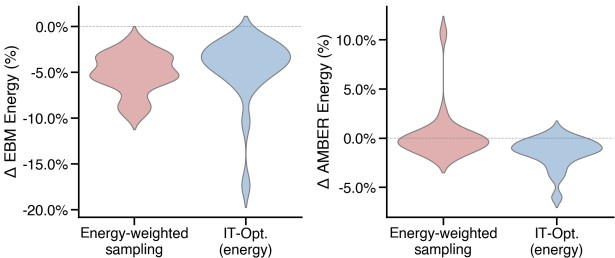

*Figure 5.* **Energy-weighted inference improves thermodynamic stability of the ensemble.** Energy changes relative to a uniformly weighted baseline (dashed line) for energy-weighted sampling and energy-weighted IT-Opt. Left: ProteinEBM. Right: AMBER99. Negative change means more stable structures.

Under this reweighting, experimental observables are evaluated as Boltzmann-weighted ensemble averages, emphasizing low-energy conformations while preserving consistency with the AF3 prior. Additional implementation details, including temperature annealing and energy smoothing strategies, are provided in Appendix C.3.4.

## 5. Experiments

### 5.1. Electron density-based IT-Optimization

**Alternative Conformations.** In loop regions, density maps often exhibit multimodal backbone density due to alternative conformations (altlocs). As shown by Rosenberg et al. (2024), AF3 typically collapses such regions to a single mode, failing to accurately capture conformational heterogeneity. While guidance-based methods can recover multimodal ensembles, they degrade when alternative states are well separated (Maddipatla et al., 2025a). We illustrate this limitation using the laminarinase catalytic domain (Figure 3), where chain A of 3AZY (1.65 Å) contains a seven-residue altloc region. AF3 recovers only one mode, and guidance produces a bimodal ensemble but misfits the backbone for one mode. In contrast, IT-Optimization accurately captures both modes, including sidechain placement. Figure 15 further shows that IT-Optimization yields significantly more consistent ensembles across 5 random seeds than guidance on the same proteins, while generating physically plausible ensembles (Table 17). Figure 4A, C show that similar trends are observed across additional altloc-containing structures. Additional benchmarks are provided in Tables 14-16 (bottom).

**Protein-bound peptides.** Short peptide chains are challenging for AF3 due to their high conformational flexibility. Guidance-based approaches simplify this setting by fixing the peptide N-C termini and optimizing only internal residues (Maddipatla et al., 2025a). Here, we remove this constraint and fit the entire peptide directly into the density map using both guidance and IT-optimization, and without

the need to anchor any atoms or residues. Figure 3 shows chain B of 6I42 (1.38Å), a 13-residue peptide from the *α-synuclein-cyclophilin A complex*. AF3 fails to reproduce the experimental structure, and guidance improves only the backbone placement but yields incorrect side-chain packing. In contrast, IT-Optimization generates ensembles with well-fitted backbone and side-chain arrangements. Moreover, as shown in Figure 8, iterative optimization enables continued refinement until convergence, whereas guidance is limited to a fixed number of iterations. Lastly, Figure 4A, C show consistent improvements across the full benchmark. Additional benchmarks are provided in Tables 14-16 (top).

### 5.2. NOE-based IT-Optimization

We benchmark the proposed framework on a set of 20 protein structures from the NMRDB dataset (Klukowski et al., 2024) (Table 1). As shown in Figure 4B & D, IT-Optimization with uniformly weighted NMR log-likelihood (Equation 3) consistently reduces both the number of violated NOE restraints and the median violation distance compared to guided and unguided baselines. These improvements are robust to the density of available restraints: subsampling NOEs to as little as 10% of the full set yields validity losses (Equation 38) on par with the reference PDB ensemble, with no sign of overfitting under sparse constraints (Table 10). Incorporating energy reweighting (Equation 10) using force-field-predicted energies further improves NOE satisfaction while promoting thermodynamically plausible ensembles (Table 4). As a representative example, we consider the solution-NMR ensemble of an uncharacterized *Rhodospirillum rubrum* protein (PDB 2K0M, BMRB 15652). Figure 6 shows that IT-Optimization reduces restraint violations relative to competing methods, with additional gains when combined with energy reweighting. This also results in improved structural agreement with the reference ensemble (Figure 12). Lastly, as shown in Figure 5 and Table 7, energy-weighted variants produce lower-energy ensembles with higher effective sample size (ESS) under the optimization force field (ProteinEBM (Roney et al., 2025)) than their uniformly weighted counterparts. Additional benchmarks are provided in Tables 5 and 6.

### 5.3. ipTM-based IT-Optimization

We systematically analyzed the effects of inference-time optimization of the ipTM objective on predicted protein complexes, with the goal of characterizing how such optimization interacts with the AF3 embedding space. Given the widespread use of ipTM as a confidence and ranking metric in large-scale complex screening and binder design, it is important to understand how perturbations in the AF3 embeddings influence ipTM values and the resulting structural outputs.

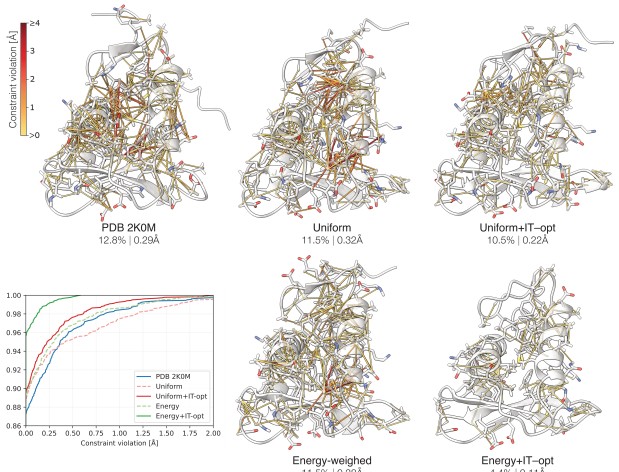

*Figure 6.* **Inference-time optimization, with and without energy weighting, reduces NOE restraint violations.** NOE constraint violations in 2K0M and ensembles from PDB, Guidance (Uniform), Inference-time Optimization (Uniform + IT-opt), Energy-weighted Guidance, and Energy-weighted IT-Opt. Violated constraints are shown as lines and colored by violation magnitude. The percentage of violated constraints and their median violation distance are reported below each structure. *Bottom left*: cumulative distribution of violation magnitudes across all five generated ensembles.

We conduct the following perturbation experiment. For each target complex, we perform inference-time optimization of the AF3 embeddings to increase the ipTM score (and related weighted metrics). Because AF3's ipTM predictor depends jointly on the AF3 embeddings and the predicted complex structure, it is sensitive to perturbations in the embedding space. Optimization is performed by iteratively perturbing the AF3 embeddings and resampling structures using the AF3 diffusion model to increase ipTM. A maximum relative perturbation budget is enforced on the AF3 embeddings; upon reaching this limit, optimization is halted, and the resulting embeddings are used for structure sampling.

The evaluated complexes comprise three distinct classes. Firstly, complexes involving the tumor suppressor p53 (Oren, 1999), determined either by NMR – focusing on the transiently ordered N-terminal domain (2K8F, 2L14, and 2LY4) – or by X-ray crystallography, characterizing the core domain interactions (1YCS). Here, unguided AF3 predictions exhibit low baseline confidence in terms of both pTM and ipTM. These systems, therefore, represent challenging test cases for confidence-driven inference-time optimization. Secondly, we consider de novo designed protein–protein interfaces generated using BindCraft; (9HAD, 9HAF). These targets typically exhibit higher baseline confidence and provide a complementary regime for assessing the sensitivity of ipTM optimization in designed binding scenarios. Lastly, we consider 8Q70 a natural heterodimer with a domain swap.

**Sensitivity of ipTM to embedding perturbations.** Using

the inference-time optimization procedure described above, we first examine how the ipTM-driven objective (Eq. 5) evolves under continued optimization. Across all evaluated targets, both ipTM and pTM components can be systematically increased within the imposed perturbation budget (Figure 7). Notably, these increases are often achieved with only very small relative changes to the embedding ($\approx 0.01\%$), indicating that the confidence objective is highly sensitive to local directions in embedding space. However, the figure further shows that increases in confidence are not consistently accompanied by improvements in structural accuracy: while confidence scores generally increase monotonically with perturbation budget, RMSD to experiment is target-dependent and may remain unchanged or even worsen in some cases, particularly in the absence of MSA input. This decoupling holds under interface-specific metrics as well, with ipSAE, DockQ, Fnat, and hydrogen-bond recovery largely unchanged across most targets despite substantial confidence gains (Figures 10 & 13). Together, these results suggest that ipTM is highly sensitive to local variations in the embedding space and can be driven to high-confidence regimes via small perturbations, but that such increases do not necessarily imply improved agreement with experiment.

**ipTM-guidance improves complex predictions in some cases.** Due to this sensitivity, ipTM optimization can sometimes coincide with improved interfacial geometry and experimental agreement. In the crystallographic complex 1YCS, ipTM-based IT-Optimization recovers a helix-mediated contact missed by AF3 (Figure 11A) and increases the recovery of native inter-chain hydrogen bonds, yielding a higher average contact recovery rate, reflecting a larger fraction of experimentally observed contacts recovered per sampled structure (Table 21). A similar effect is observed for the NMR complex 2LY4, which consists of a globular target bound to the transiently ordered N-terminal domain of p53. While unguided AF3 samples both an exterior binding mode associated with higher ipTM scores and an interior binding mode supported by the experimental NMR ensemble, ipTM-based inference-time optimization consistently favors the interior binding configuration across random initializations (Figure 11B). This shift is accompanied by a substantial enrichment in experimentally observed inter-chain hydrogen bonds within the peptide helical region, indicating that ipTM optimization in this case promotes structures that better reflect the experimentally supported binding mode (Table 22). In low-information settings, ipTM-based inference-time optimization generally does not correct major structural errors. For the designed complex 9HAF without MSA input, AF3 mispredicts one chain (27% agreement). Optimization increases secondary-structure agreement to 59.2% (Figure 9), but does not recover the correct fold, suggesting confidence-driven optimization can partially reduce errors but is not reliably predictive in low-information regimes.

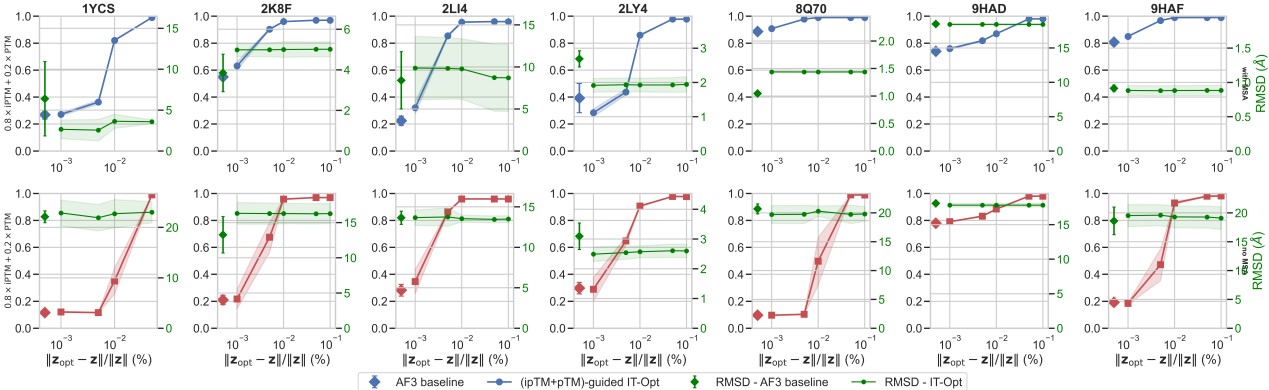

*Figure 7.* **Effect of inference-time optimization on ipTM across biomolecular complexes.** The figure shows the impact of inference-time optimization targeting $0.8 \times \text{ipTM} + 0.2 \times \text{pTM}$ across seven biomolecular complexes, evaluated with and without MSA input (top and bottom rows, respectively). Diamonds denote baseline AF3 predictions at zero perturbation, while curves show optimized solutions obtained under increasing relative perturbation budgets $\|\mathcal{Z}_{\text{opt}} - \mathcal{Z}\|/\|\mathcal{Z}\|$ in AF3 embedding space; shaded regions indicate variability across samples drawn given the optimized embedding. The left axis reports confidence, while the right axis reports backbone RMSD to experiment. Across several targets, we observe that relatively small embedding perturbations can increase confidence scores. However, these increases are not uniformly accompanied by improvements in structural agreement with experiment, particularly in the no-MSA setting. These results suggest that while embedding-space optimization can effectively modulate confidence metrics, confidence improvements should be interpreted cautiously as indicators of experimental accuracy.

# 6. Related Work

**Experimental guidance.** Recent methods treat AF3 as a sequence-conditioned prior and incorporate experimental likelihoods from cryo-EM, NMR, and X-ray crystallography into the reverse SDE (Maddipatla et al., 2024; Raghu et al., 2025; Maddipatla et al., 2025b;a). Our method directly optimizes the Pairformer embeddings, thereby avoiding limitations imposed by the diffusion schedule and finite denoising steps. This approach further relates to enhanced-sampling methods like metadynamics (Barducci et al., 2011), as we utilize differentiable force fields to recover experimentally faithful, Boltzmann-distributed ensembles. Our Boltzmann reweighting draws on the broader paradigm of sampling from Boltzmann densities with learned generative models (Noé et al., 2019; Akhound-Sadegh et al., 2024), though it requires no additional training and instead performs posthoc reweighting of AF3 samples at inference time. Related work has also explored the conformational landscape using RMSD-based repulsive potentials as a guidance signal (Richman et al., 2025).

**MSA manipulation.** Prior works have shown that manipulating the MSA can recover distinct conformational states (Wayment-Steele et al., 2024) or steer predictions toward desired structural targets (Bryant & Noé, 2024; Fadini et al., 2026) by optimizing AlphaFold2's MSA profile. However, these methods require expensive backpropagation through the Evoformer and assume a single structural target. We avoid these bottlenecks by optimizing a batch of embeddings directly. This allows for efficient inference-time steering and enables the generation of ensembles that capture the experimental heterogeneity inherent to proteins.

**ipTM benchmarking and confidence.** The interface predicted template modeling (ipTM), introduced with AF-Multimer, is a confidence metric designed to assess the accuracy of predicted protein–protein interfaces by focusing on inter-chain geometry (Evans et al., 2022). Benchmark studies show that ipTM correlates well with interface accuracy metrics such as DockQ (Mirabello & Wallner, 2024) across diverse heteromeric datasets (Zhai et al., 2025). However, ipTM does not capture all structural failure modes, particularly in flexible or disordered regions (Wee & Wei, 2024), motivating the development of refined interface confidence metrics such as actual interface pTM (actifpTM) (Varga et al., 2025) and interaction prediction score from aligned errors (ipSAE) (Dunbrack Jr, 2025).

# 7. Conclusion

In this work, we introduce an inference-time optimization (IT-Opt) framework that updates Pairformer embeddings to generate structural ensembles consistent with experimental data. Using a nested optimization scheme, the method mitigates key limitations of traditional guidance, including initialization bias and fixed sampling horizons. It also enables thermodynamically consistent ensemble generation. Across NMR and X-ray crystallography benchmarks, IT-Opt improves agreement with experimental observables and recovers challenging structural features. We further show that confidence metrics such as ipTM can be artificially inflated by small perturbations, revealing an important limitation of current confidence frameworks. In future work, we aim to extend this framework to additional structural modalities, including single-particle cryo-EM.

## Acknowledgments

Ailie Marx acknowledges the financial support of the Helmsley Fellowships Program for Sustainability and Health. Alex Bronstein is supported by the Israeli Science Foundation grant 1834/24 and ISTA HPC Cluster. Paul Schanda is supported by the Austrian Science Fund (FWF, grant numbers I5812-B and I6223). Martin Pacesa is supported by the European Research Council (ERC) Starting Grant 101220545 (NAMPify) and by the University Research Priority Program of the University of Zurich (URPP) Innovative Therapies in Rare Diseases (ITINERARE). Sanketh Vedula is supported in part by funding from the Eric and Wendy Schmidt Center at the Broad Institute of MIT and Harvard.

## Impact statement

This work introduces inference-time optimization methods for generating experiment-grounded protein ensembles using AF3. Our approach enables more accurate and thermodynamically plausible ensemble generation from X-ray crystallographic and NMR data, potentially accelerating structure determination workflows. The Boltzmann reweighting framework bridges machine learning predictions with physically meaningful conformational distributions. Our ipTM optimization analysis reveals vulnerabilities in confidence metrics used for protein-binder design, which could help reduce false discovery rates in therapeutic development. Given the central role of proteins in biological processes and drug discovery, these advances may benefit both basic research and biomedical applications. We foresee no specific ethical concerns arising from this work.

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

# A. Additional figures, tables, and pseudocodes

## A.1. Figures

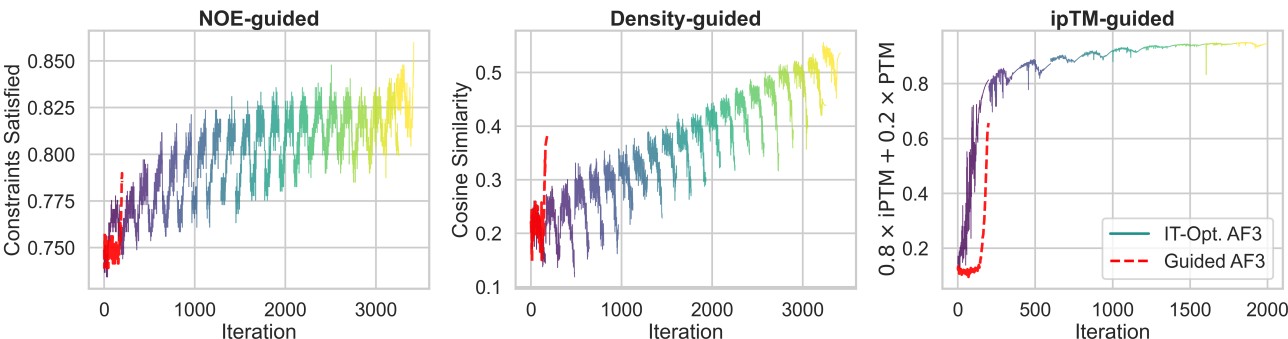

*Figure 8.* **Convergence behavior of guided versus inference-time optimized (IT-Opt) AlphaFold3 across experimental modalities.** Convergence curves are shown for NOE-based IT-Opt on an NMR structure (`1U0P`), density-based IT-Opt on an X-ray structure (`6I42`), and ipTM-based IT-Opt on a protein complex (`2L14`). In each panel, dashed red curves correspond to standard guided AlphaFold3, where guidance is applied during sampling and with a fixed budget of 200 diffusion iterations. Solid curves show inference-time optimization, where embedding updates are applied at each diffusion step (with early stopping at $t = 160$ in inner loop) and reused across successive diffusion rounds. In the final outer loop, (the last 200 iterations of IT-Opt), the optimized embeddings initialize coordinate-space guidance, yielding the late-stage performance improvement characteristic of traditional guidance.

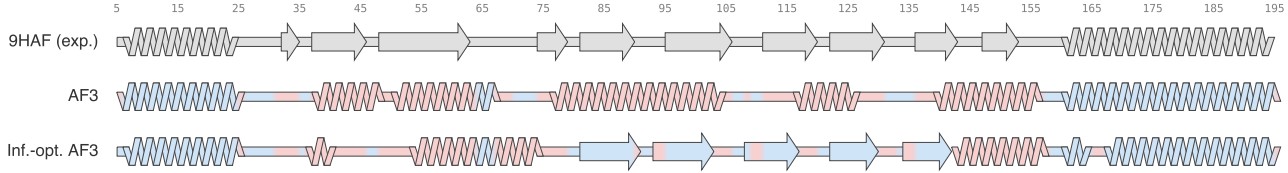

*Figure 9.* **Secondary-structure comparison for complex `9HAF` under ipTM-based inference-time optimization without MSA.** Schematic comparison of secondary-structure assignments along chain A of complex `9HAF`. The top row shows the experimentally determined secondary structure, while the middle and bottom rows correspond to the unguided AF3 prediction without MSA input and the IT-optimized AF3 prediction targeting the combined ipTM+pTM objective, respectively. Helices and $\beta$-strands are shown as ribbons and arrows, with residue-wise agreement (blue) and disagreement (red) relative to the experimental annotation indicated by color. In the unguided prediction, the structure is dominated by helical assignments and exhibits low secondary-structure agreement with the experimental annotation (41.9%). Following ipTM-based IT-optimization, large regions of misassigned helices are corrected, and extended $\beta$-strand regions are recovered, increasing secondary-structure agreement to 59.2%.

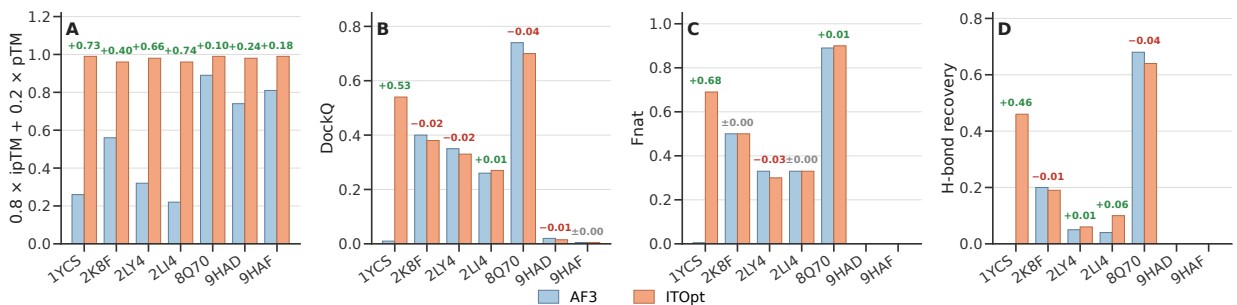

*Figure 10.* **ipTM-based IT-Optimization inflates confidence without improving structural accuracy.** The figure shows the impact of IT-Optimization across seven biomolecular complexes using $0.8 \times \text{ipTM} + 0.2 \times \text{pTM}$ at a relative perturbation budget ($\|\mathcal{Z}_{\text{opt}} - \mathcal{Z}\| / \|\mathcal{Z}\|$) of 0.1 in AF3 embedding space. **(A)** IT-Opt consistently increases confidence scores. **(B-D)** Despite this, structural accuracy is largely unchanged in $6/7$ cases across DockQ (Mirabello & Wallner, 2024) **(B)**, Fraction of native contacts (Fnat) **(C)**, and Hydrogen-bond recovery **(D)**. The single improving case (`1YCS`) shows confidence gains aligned with genuine structural recovery across all metrics. Together, these results indicate that ipTM is susceptible to inflation by minor embedding perturbations. Per-target deltas (IT-Opt $-$ AF3) are shown above each bar pair; green and red indicate gains and losses respectively, while gray indicates no change.

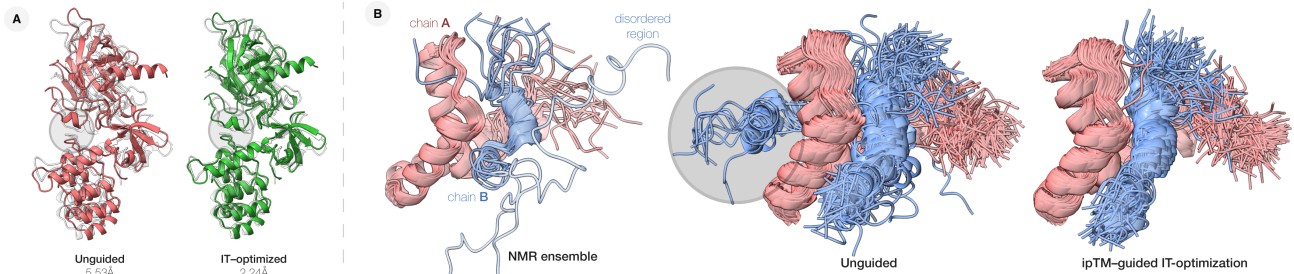

*Figure 11.* **Selected examples where ipTM-based inference-time optimization improves agreement with experiment.** (A) Complex `1YCS`. The experimental crystal structure is shown in white, the unguided AlphaFold3 prediction in red, and the ipTM-optimized prediction in green. In this case, the unguided AF3 prediction fails to form a specific helix-helix contact (left), whereas ipTM-based IT-Opt recovers this interaction (right), resulting in closer agreement with the experimental structure. (B) Complex `2LY4`, comprising a globular target (chain A) and a peptide derived from `p53`. The experimental NMR ensemble (left) supports an interior binding mode. Across 100 unguided AF3 samples (middle), two binding modes are observed: an exterior conformation associated with higher ipTM (0.55) and an interior conformation with lower ipTM (0.22). In this example, ipTM-based IT-Opt (right) consistently samples the interior binding mode across 100 random initializations, producing structures that better align with the experimental ensemble.

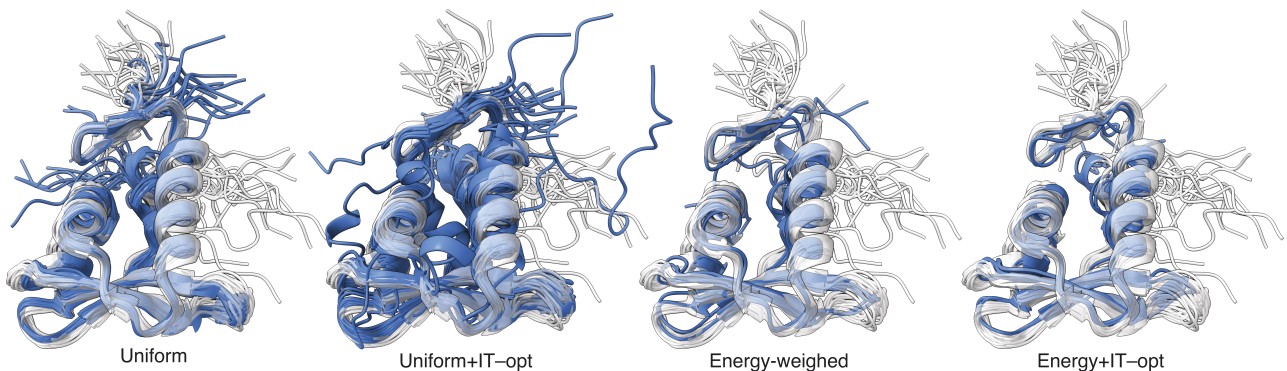

*Figure 12.* Conformational ensembles for `2K0M` from Guidance (uniform), Inference-time Optimization (uniform), Energy-weighted Guidance, and Energy-weighted Inference-time Optimization (blue) overlaid with the NMR ensemble derived from the same NOESY data (white). Uniform ensembles weight all sampled conformations equally, whereas energy-weighted ensembles display only conformations with non-negligible Boltzmann weight.

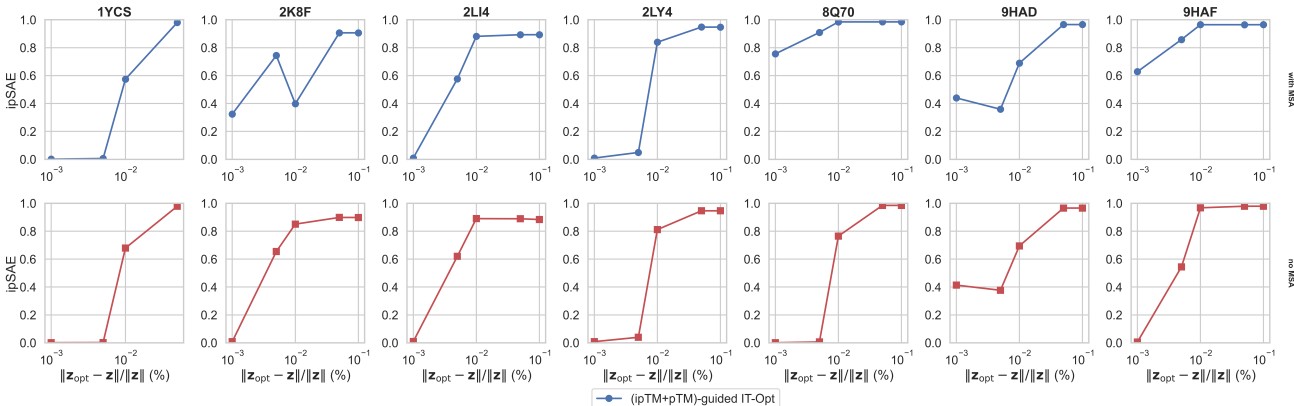

*Figure 13.* **Effect of inference-time optimization on Interaction Prediction Score from Aligned Errors (ipSAE) across biomolecular complexes.** The figure depicts the ipSAE values obtained as a result of (ipTM + pTM)-based inference-time optimization across seven biomolecular complexes, evaluated with and without MSA input. Curves show optimized solutions obtained under increasing relative perturbation budgets $\|\mathcal{Z}_{opt} - \mathcal{Z}\|/\|\mathcal{Z}\|$ in AF3 embedding space. Across targets, increases in the optimized (ipTM + pTM) objective are accompanied by consistent increases in ipSAE, with ipSAE frequently approaching near-saturation values at moderate perturbation magnitudes. Similar trends are observed in both MSA-rich and MSA-poor settings, although sensitivity to perturbation varies across targets. These results indicate that, like ipTM, ipSAE is highly sensitive to small embedding-space perturbations and can be driven to high values through inference-time optimization.

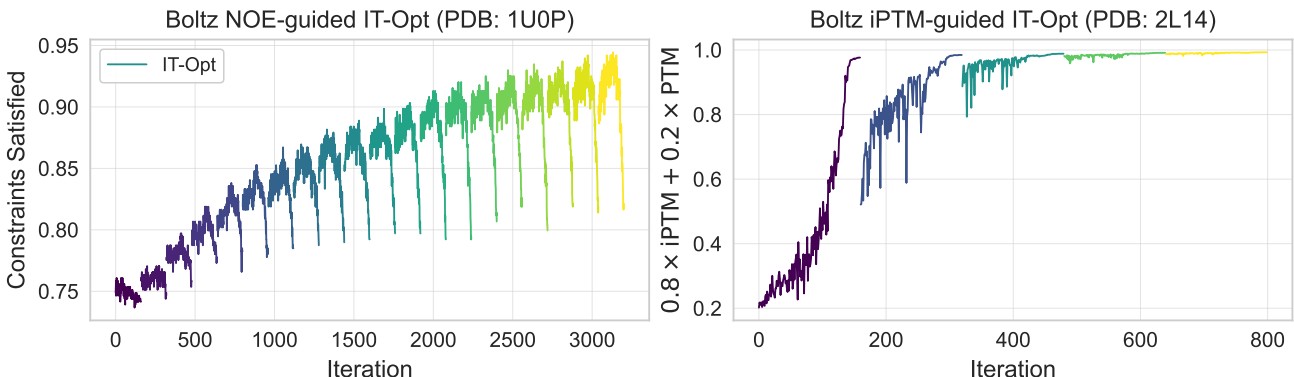

*Figure 14.* **IT-Optimization convergence generalizes across protein generative models.** Convergence curves for NOE-based IT-Optimization on `1U0P` (left) and ipTM-based IT-Optimization on `2L14` (right) using Boltz-2 (Passaro et al., 2025), following the same optimization protocol as in Figure 8. The monotonic outer-loop improvement matches the behavior observed for Protenix and AlphaFold3 under analogous likelihood signals, demonstrating that IT-Opt is agnostic to the underlying protein generative model.

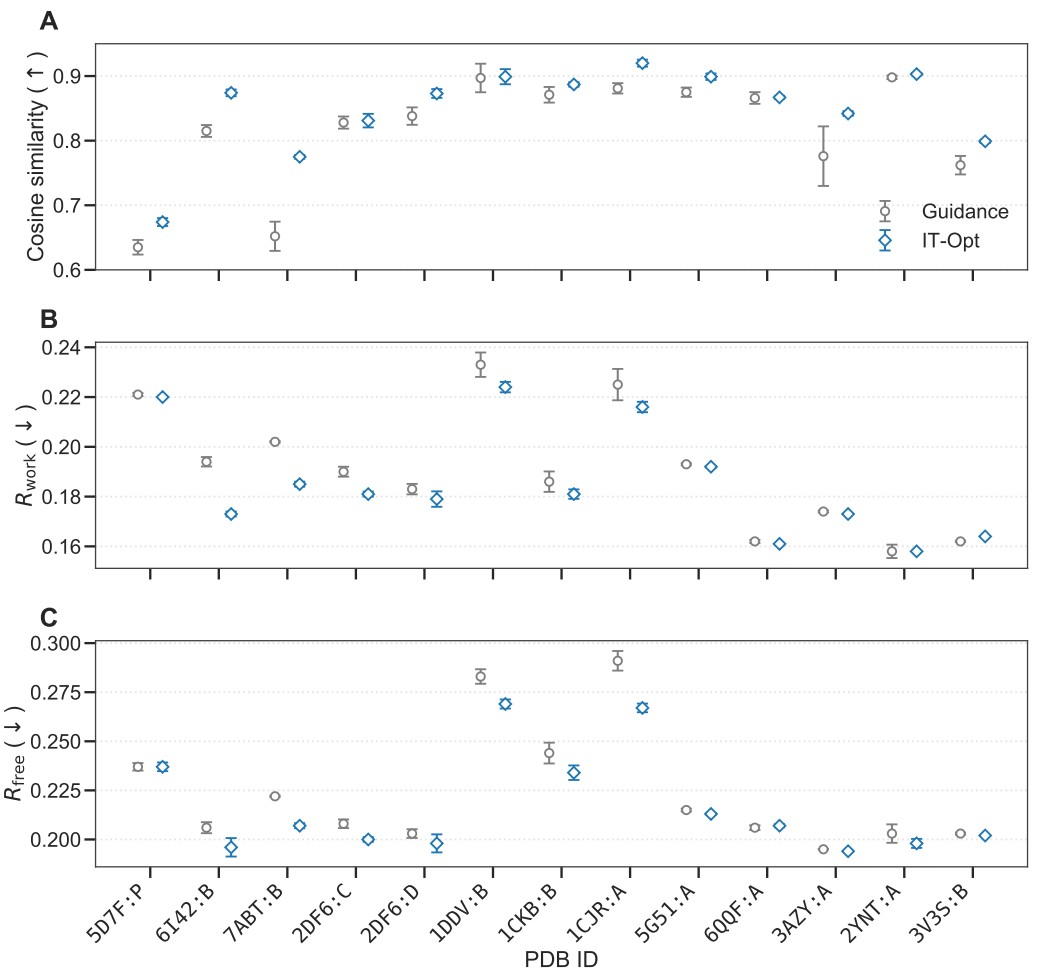

*Figure 15.* **IT-Optimization exhibits improved stability compared to Guidance.** Crystallographic metrics computed across five independent random-seed runs on all proteins from Table 2 for electron density-guided AlphaFold3 (*Guidance*) and electron density-based Inference-time Optimization (*IT-Opt*). Markers indicate the mean and error bars the standard deviation across runs. IT-Opt consistently achieves improved agreement with the experimental electron density, reflected by higher cosine similarity (**A**), lower $R_{\text{work}}$ (**B**), and lower $R_{\text{free}}$ (**C**), while also exhibiting reduced run-to-run variability across all three metrics.

## A.2. Tables

*Table 1.* **NMR protein structures used for evaluation in our experiments**. For each structure, we report the amino acid sequence length and the number of experimentally derived Nuclear Overhauser Effect (NOE) distance restraints.

| PDB ID | Seq Length | # NOEs |
|--------|-----------|--------|
| 2K53 | 70 | 932 |
| 6SVC | 35 | 1147 |
| 1U0P | 27 | 198 |
| 1DEC | 39 | 493 |
| 6SOW | 58 | 1120 |
| 2KPB | 26 | 267 |
| 5L82 | 37 | 485 |
| 2MRW | 40 | 159 |
| 1D3Z | 76 | 2727 |
| 2B5B | 36 | 96 |
| 1RKL | 36 | 218 |
| 2KNS | 33 | 233 |
| 2RN7 | 108 | 353 |
| 2K0M | 104 | 1834 |
| 2HEQ | 84 | 485 |
| 2K57 | 55 | 982 |
| 1PQX | 91 | 1490 |
| 1YEZ | 68 | 1237 |
| 2KCD | 120 | 1075 |
| 2HN8 | 38 | 719 |

*Table 2.* **X-ray crystallography protein used for evaluation in our experiments**. For each structure, we report the chain ID, the residue range, the region sub-sequence, and the resolution of the density map. The top group consists of protein-bound peptides; the bottom group consists of alternate locations (altlocs).

| PDB ID | Residue Range | Region Sequence | Resolution (Å) |
|---|---|---|---|
| 5D7F:P | 1–10 | GPWDSVARVL | 1.30 |
| 6I42:B | 48–60 | VVHGVATVAEKTK | 1.38 |
| 7ABT:B | 1–8 | PRPRPRPR | 1.31 |
| 2DF6:C | 180–197 | PPVIAPRPEHTKSIYTRS | 1.30 |
| 2DF6:D | 180–191 | PPVIAPRPEHTK | 1.30 |
| 1DDV:B | 1001–1006 | TPPSPF | 1.90 |
| 1CKB:B | 1–8 | PPPVPPRR | 1.90 |
| 1CJR:A | 1–15 | KETAAAKFERQHMDS | 2.30 |
| 5G51:A | 291–295 | SASDQ | 1.45 |
| 6QQF:A | 69–74 | TPGSRN | 1.95 |
| 3AZY:A | 157–163 | GGASIGV | 1.65 |
| 2YNT:A | 183–185 | GNG | 1.60 |
| 3V3S:B | 246–249 | AQER | 1.90 |

*Table 3.* **Quantitative evaluation of NOE restraint violations for structures from Table 1**. The violation percentage (**Viol %** (↓)) denotes the fraction of NOE distance restraints that are not satisfied, while the violation distance (**Viol. Å** (↓)) reports the median distance of unsatisfied NOE restraints. Here, *Guidance* refers to ensembles derived using NOE-guided AlphaFold3 (Maddipatla et al., 2025b), whereas *IT-Opt* denotes ensembles derived using NOE-based inference-time optimization. Entries highlighted in **green** correspond to ensembles that outperform all baseline methods, excluding the PDB, while entries highlighted in **blue** outperform all baselines, including the PDB.

| | PDB | | AlphaFold3 | | AlphaFlow | | Guidance | | IT-Opt | |
|---|---|---|---|---|---|---|---|---|---|---|
| PDB ID | Viol. % | Viol. Å | Viol. % | Viol. Å | Viol. % | Viol. Å | Viol. % | Viol. Å | Viol. % | Viol. Å |
| 2K53 | 29.6 | 0.35 | 41.1 | 0.53 | 40.5 | 0.47 | 28.1 | 0.20 | **20.6** | **0.13** |
| 6SVC | 51.8 | 0.94 | 50.6 | 0.44 | 49.7 | 0.41 | 51.4 | 0.38 | **44.4** | **0.33** |
| 1U0P | 57.5 | 0.13 | 60.4 | 0.35 | 53.1 | 0.34 | 50.2 | 0.22 | **44.4** | **0.20** |
| 1DEC | 12.4 | 0.07 | 32.1 | 0.75 | 45.6 | 2.84 | 19.9 | 0.36 | **14.4** | **0.23** |
| 6SOW | 32.0 | 0.51 | 43.3 | 0.69 | 43.7 | 0.71 | 30.0 | 0.42 | **28.6** | **0.36** |
| 2KPB | 27.2 | 0.52 | 62.3 | 1.35 | 57.7 | 1.28 | 45.3 | 1.24 | **40.7** | **0.63** |
| 5L82 | 31.0 | 0.71 | 58.8 | 0.95 | 57.0 | 1.19 | 49.4 | 0.86 | **44.0** | **0.62** |
| 2MRW | 23.3 | 0.10 | 38.4 | 0.14 | 33.9 | 0.12 | **22.6** | 0.30 | 24.5 | **0.06** |
| 1D3Z | 24.9 | 0.39 | 23.8 | 0.34 | 24.0 | 0.36 | 13.5 | 0.18 | **11.3** | **0.14** |
| 2B5B | 30.6 | 0.39 | 29.7 | 5.80 | 40.7 | 4.87 | 29.6 | 3.58 | **28.7** | **0.73** |
| 1RKL | 53.0 | 0.36 | 57.2 | 0.63 | 56.7 | 0.59 | 54.0 | 0.42 | **48.8** | **0.33** |
| 2KNS | 58.0 | 0.36 | 65.5 | 0.48 | 58.0 | 0.59 | 50.4 | 0.39 | **49.6** | **0.30** |
| 2RN7 | 11.4 | 0.44 | 9.4 | 0.28 | 11.4 | 0.33 | 5.3 | 0.08 | **4.7** | **0.05** |
| 2K0M | 12.8 | 0.29 | 17.9 | 0.48 | 19.6 | 0.61 | 11.5 | 0.32 | **10.5** | **0.22** |
| 2HEQ | 18.2 | 0.36 | 17.7 | 0.52 | 31.2 | 0.62 | 10.0 | **0.12** | **9.5** | 0.31 |
| 2K57 | 12.5 | 0.50 | 17.8 | 0.52 | 19.2 | 0.51 | 8.2 | 0.19 | **7.9** | **0.06** |
| 1PQX | 1.2 | 0.05 | 5.1 | 0.71 | 5.2 | 0.82 | 3.3 | 0.43 | **2.2** | **0.08** |
| 1YEZ | 12.7 | 0.46 | 13.8 | 0.41 | 17.0 | 0.46 | 7.5 | 0.24 | **7.0** | **0.12** |
| 2KCD | 23.9 | 0.55 | 21.2 | 0.46 | 39.7 | 1.38 | 13.8 | 0.25 | **10.5** | **0.09** |
| 2HN8 | 40.5 | 0.08 | 54.0 | 0.34 | 49.6 | 0.32 | 41.6 | 0.24 | **34.2** | **0.15** |

*Table 4.* **Quantitative evaluation of NOE restraint violations for structures from Table 1**. The violation percentage (**Viol % (↓)**) denotes the fraction of NOE distance restraints that are not satisfied, while the violation distance (**Viol. Å (↓)**) reports the median distance of unsatisfied NOE restraints. Here, *Guidance (Energy)* refers to ensembles derived using NOE-guided AlphaFold3 (Maddipatla et al., 2025b), with Boltzmann-weighted ensemble statistics at 300K, whereas *IT-Opt (Energy)* denotes ensembles obtained via NOE-based inference-time optimization, with Boltzmann weighting at 300K. Entries highlighted in **green** correspond to ensembles that outperform all baseline methods, excluding the PDB, while entries highlighted in **blue** outperform all baselines, including the PDB.

| | PDB | | Guidance | | Guidance (Energy) | | IT-Opt | | IT-Opt (Energy) | |
|---|---|---|---|---|---|---|---|---|---|---|
| **PDB ID** | **Viol. %** | **Viol. Å** | **Viol. %** | **Viol. Å** | **Viol. %** | **Viol. Å** | **Viol. %** | **Viol. Å** | **Viol. %** | **Viol. Å** |
| 2K53 | 29.6 | 0.35 | 28.1 | 0.20 | 32.7 | 0.29 | 20.6 | 0.13 | **19.0** | **0.18** |
| 6SVC | 51.8 | 0.94 | 51.4 | 0.38 | 42.3 | 0.38 | 44.4 | **0.38** | **38.3** | 0.42 |
| 1U0P | 57.5 | 0.13 | 50.2 | 0.22 | 49.3 | 0.22 | 44.4 | 0.20 | **41.1** | **0.15** |
| 1DEC | 12.4 | 0.07 | 19.9 | 0.36 | 18.7 | 0.58 | 14.4 | 0.23 | **11.6** | **0.16** |
| 6SOW | 32.0 | 0.51 | 30.0 | 0.42 | 30.2 | 0.43 | 28.6 | 0.36 | **20.0** | **0.27** |
| 2KPB | 27.2 | 0.52 | 45.3 | 1.24 | 35.7 | 0.56 | 40.7 | 0.63 | **34.8** | **0.44** |
| 5L82 | 31.0 | 0.71 | 49.4 | 0.86 | 49.2 | 1.07 | 44.0 | 0.62 | **34.4** | **0.42** |
| 2MRW | 23.3 | 0.10 | 22.6 | 0.30 | **17.6** | 0.16 | 24.5 | 0.06 | 18.2 | **0.05** |
| 1D3Z | 24.9 | 0.39 | 13.5 | 0.18 | 17.3 | 0.25 | **11.3** | **0.14** | 16.4 | 0.25 |
| 2B5B | 30.6 | 0.39 | 29.6 | 3.58 | 30.6 | 4.12 | 28.7 | **0.73** | **27.8** | 2.04 |
| 1RKL | 53.0 | 0.36 | 54.0 | 0.42 | 54.4 | 0.44 | 48.8 | 0.33 | **42.3** | **0.23** |
| 2KNS | 58.0 | 0.36 | 50.4 | 0.39 | 47.1 | 0.43 | 49.6 | 0.30 | **39.1** | **0.28** |
| 2RN7 | 11.4 | 0.44 | 5.3 | 0.08 | 4.4 | 0.26 | 4.7 | **0.05** | **4.4** | 0.07 |
| 2K0M | 12.8 | 0.29 | 11.5 | 0.32 | 10.5 | 0.28 | 10.5 | 0.22 | **4.4** | **0.11** |
| 2HEQ | 18.2 | 0.36 | 10.0 | 0.12 | 6.1 | 0.28 | 9.5 | 0.31 | **4.3** | **0.09** |
| 2K57 | 12.5 | 0.50 | 8.2 | 0.19 | 11.1 | 0.29 | 7.9 | 0.06 | **5.9** | **0.09** |
| 1PQX | 1.2 | 0.05 | 3.3 | 0.43 | 1.1 | 0.08 | 2.2 | 0.08 | **1.2** | **0.05** |
| 1YEZ | 12.7 | 0.46 | 7.4 | 0.24 | 12.6 | 0.41 | 7.0 | 0.12 | **4.5** | **0.10** |
| 2KCD | 23.9 | 0.55 | 13.8 | 0.25 | 12.8 | 0.46 | 10.5 | 0.09 | **6.9** | **0.09** |
| 2HN8 | 40.5 | 0.08 | 41.6 | 0.24 | 43.3 | 0.29 | 34.2 | 0.15 | **30.3** | **0.14** |

*Table 5.* **Quantitative evaluation of NOE violation percentage for structures from Table 1 on additional benchmarks**. The violation percentage (**Viol. %** (↓)) denotes the fraction of NOE distance restraints that are not satisfied. Here, *Guidance* refers to ensembles derived using NOE-guided AlphaFold3 (Maddipatla et al., 2025b), whereas *IT-Opt* denotes ensembles derived using NOE-based inference-time optimization. Entries highlighted in **bold** indicate the best result.

| PDB ID | AlphaFlow | ESMFlow | AFCluster | BioEMU | Guidance | IT-Opt |
|---|---|---|---|---|---|---|
| 2K53 | 40.5 | 46.6 | 47.4 | 52.9 | 28.1 | **20.6** |
| 6SVC | 49.7 | 54.6 | 52.2 | 54.6 | 51.4 | **44.4** |
| 1U0P | 54.1 | 56.5 | 62.3 | 72.9 | 50.2 | **44.4** |
| 1DEC | 45.6 | 36.5 | 47.9 | 49.9 | 19.9 | **14.4** |
| 6SOW | 43.7 | 45.3 | 53.7 | 68.5 | 30.0 | **28.6** |
| 2KPB | 57.7 | 59.3 | 55.1 | 77.0 | 45.3 | **40.7** |
| 5L82 | 57.0 | 61.9 | 60.4 | 73.6 | 49.4 | **44.0** |
| 2MRW | 33.9 | 38.4 | 39.6 | 50.3 | **22.6** | 24.5 |
| 1D3Z | 24.0 | 26.3 | 28.2 | 36.6 | 13.5 | **11.3** |
| 2B5B | 40.7 | 29.6 | 30.6 | 42.6 | 29.6 | **28.7** |
| 1RKL | 56.7 | 58.1 | 54.9 | 56.7 | 54.0 | **48.8** |
| 2KNS | 58.0 | 68.1 | 67.6 | 72.3 | 50.4 | **49.6** |
| 2RN7 | 11.4 | 10.6 | 21.7 | 40.3 | 5.3 | **4.7** |
| 2K0M | 19.6 | 20.7 | 23.7 | 36.2 | 11.5 | **10.5** |
| 2HEQ | 31.2 | 27.0 | 52.3 | 39.1 | 10.0 | **9.5** |
| 2K57 | 19.2 | 21.1 | 26.1 | 56.1 | 8.2 | **7.9** |
| 1PQX | 5.2 | 5.0 | 7.9 | 23.2 | 3.3 | **2.2** |
| 1YEZ | 17.0 | 13.8 | 14.3 | 29.9 | 7.5 | **7.0** |
| 2KCD | 39.7 | 45.8 | 40.7 | 35.5 | 13.8 | **10.5** |
| 2HN8 | 49.6 | 49.2 | 53.6 | 63.7 | 41.6 | **34.2** |

*Table 6.* **Quantitative evaluation of median NOE violation distance for structures from Table 1 on additional benchmarks**. The violation percentage (**Viol Å** (↓)) denotes the median distance of unsatisfied NOE restraints. Here, *Guidance* refers to ensembles derived using NOE-guided AlphaFold3 (Maddipatla et al., 2025b), whereas *IT-Opt* denotes ensembles derived using NOE-based inference-time optimization. Entries highlighted in **bold** indicate the best result.

| PDB ID | AlphaFlow | ESMFlow | AFCluster | BioEMU | Guidance | IT-Opt |
|---|---|---|---|---|---|---|
| 2K53 | 0.47 | 0.46 | 0.54 | 0.61 | 0.20 | **0.13** |
| 6SVC | 0.41 | 0.66 | 0.50 | 0.66 | 0.38 | **0.33** |
| 1U0P | 0.34 | 0.37 | 0.33 | 0.39 | 0.22 | **0.20** |
| 1DEC | 2.84 | 0.92 | 3.89 | 5.48 | 0.36 | **0.23** |
| 6SOW | 0.71 | 0.60 | 0.78 | 1.47 | 0.42 | **0.36** |
| 2KPB | 1.28 | 1.34 | 1.20 | 2.14 | 1.24 | **0.63** |
| 5L82 | 1.19 | 1.26 | 1.41 | 1.83 | 0.86 | **0.62** |
| 2MRW | 0.12 | 0.20 | 0.27 | 1.02 | 0.30 | **0.06** |
| 1D3Z | 0.36 | 0.37 | 0.44 | 0.53 | 0.18 | **0.14** |
| 2B5B | 4.87 | 5.07 | 5.11 | 6.96 | 3.58 | **0.73** |
| 1RKL | 0.59 | 0.55 | 0.64 | 0.64 | 0.42 | **0.33** |
| 2KNS | 0.59 | 0.41 | 0.49 | 0.99 | 0.39 | **0.30** |
| 2RN7 | 0.33 | 0.46 | 0.58 | 1.32 | 0.08 | **0.05** |
| 2K0M | 0.61 | 0.53 | 0.61 | 0.96 | 0.32 | **0.22** |
| 2HEQ | 0.62 | 0.71 | 2.35 | 0.70 | **0.12** | 0.31 |
| 2K57 | 0.51 | 0.42 | 0.74 | 7.19 | 0.19 | **0.06** |
| 1PQX | 0.82 | 0.82 | 0.48 | 0.88 | 0.43 | **0.08** |
| 1YEZ | 0.46 | 0.48 | 0.60 | 0.76 | 0.24 | **0.12** |
| 2KCD | 1.38 | 3.72 | 1.50 | 0.94 | 0.25 | **0.09** |
| 2HN8 | 0.32 | 0.34 | 0.47 | 0.72 | 0.24 | **0.15** |

*Table 7.* **Effective Sample Size (ESS) of ensembles derived using IT-Opt and Guidance, with and without the force field.** The ESS quantifies the number of proteins in the ensemble with high Boltzmann probability under the Protein-EBM force field at 300 K. For each protein, the highest ESS across the four methods is shown in **bold**.

| PDB ID | Guidance | IT-Opt | Guidance (Energy) | IT-Opt (Energy) |
|--------|----------|--------|-------------------|-----------------|
| 2K53 | 1.490 | 1.002 | 2.326 | **2.389** |
| 6SVC | 1.029 | 1.000 | **2.005** | 2.000 |
| 1U0P | 1.000 | 1.877 | **2.591** | 2.010 |
| 1DEC | 1.000 | 1.000 | **2.413** | 2.000 |
| 6SOW | 1.000 | 1.004 | 2.000 | **2.791** |
| 2KPB | 1.207 | 1.427 | 2.001 | **2.003** |
| 5L82 | 1.000 | 1.000 | **2.008** | 2.000 |
| 2MRW | 1.000 | 1.000 | 2.000 | **2.001** |
| 1D3Z | 1.000 | 1.000 | **2.016** | 2.001 |
| 2B5B | 1.000 | 1.000 | 2.110 | **2.423** |
| 1RKL | 1.000 | 1.000 | 2.079 | **2.509** |
| 2KNS | 1.000 | 1.003 | **2.086** | 2.000 |
| 2RN7 | 1.000 | 1.000 | 2.000 | **2.000** |
| 2K0M | 1.000 | 1.269 | 2.000 | **2.000** |
| 2HEQ | 1.025 | 1.038 | 2.000 | **2.000** |
| 2K57 | 1.012 | 1.000 | **2.784** | 2.000 |
| 1PQX | 1.000 | 1.000 | 2.000 | **2.901** |
| 1YEZ | 2.043 | 1.838 | 2.000 | **2.913** |
| 2KCD | 1.001 | 1.002 | 2.000 | **2.688** |
| 2HN8 | 1.004 | 1.000 | 2.000 | **2.000** |

*Table 8.* **Hyperparameter sensitivity for IT-Optimization on `1U0P`.** Each block sweeps one hyperparameter while the others are held at their default values (anchor $\lambda_{\mathrm{p}} = 10^{-4}$, learning rate $\eta_z = 0.05$, outer iterations $K = 20$, batch size $n = 16$). For each block, the best value per metric is shown in bold. Lower is better for all reported metrics. Validity loss is computed using Equation 38.

| Hyperparameter | Viol % ($\downarrow$) | Viol Å($\downarrow$) | Validity loss ($\downarrow$) |
|----------------|----------------------|---------------------|------------------------------|
| $\lambda_{\mathrm{p}} = 1 \times 10^{-7}$ | **40.1** | **0.1** | 4.9537 |
| $\lambda_{\mathrm{p}} = 1 \times 10^{-4}$ (default) | 44.4 | 0.2 | $\mathbf{1.97 \times 10^{-5}}$ |
| $\lambda_{\mathrm{p}} = 1 \times 10^{-2}$ | 50.2 | 0.3 | $2.91 \times 10^{-5}$ |
| $\eta_z = 0.001$ | 49.8 | 0.3 | **0** |
| $\eta_z = 0.05$ (default) | **44.4** | 0.2 | $1.97 \times 10^{-5}$ |
| $\eta_z = 1$ | 48.8 | **0.1** | $5.92 \times 10^{-1}$ |
| $K = 10$ | 50.8 | **0.2** | $2.02 \times 10^{-5}$ |
| $K = 20$ (default) | 44.4 | **0.2** | $\mathbf{1.97 \times 10^{-5}}$ |
| $K = 30$ | **43.8** | **0.2** | $2.94 \times 10^{-5}$ |
| $n = 12$ | 46.4 | **0.2** | $2.77 \times 10^{-5}$ |
| $n = 16$ (default) | 44.4 | **0.2** | $1.97 \times 10^{-5}$ |
| $n = 24$ | **43.9** | **0.2** | $\mathbf{1.75 \times 10^{-5}}$ |

*Table 9.* **Ablation over annealing, smoothing, and inverse temperature** $\beta$ **for IT-Optimization.** For each protein, the default configuration ( shown in red : ✓ annealing, ✓ smoothing, $\beta = 0.6$) is compared against three ablations: disabling annealing, disabling smoothing, and increasing the inverse temperature to $\beta = 1.2$. The best value per metric within each protein is shown in **bold**.

| PDB ID | Annealing | Smooth | $\beta$ | ESS ($\uparrow$) | Viol % ($\downarrow$) | Viol Å ($\downarrow$) |
|--------|-----------|--------|---------|------------------|----------------------|----------------------|
| 1YEZ | ✓ | ✓ | 0.6 | 2.913 | **4.5** | **0.1** |
|      | ✗ | ✓ | 0.6 | 2.000 | 4.8 | 0.6 |
|      | ✓ | ✗ | 0.6 | 2.088 | 5.1 | 0.6 |
|      | ✓ | ✓ | 1.2 | **3.016** | 11.7 | 0.4 |
| 6SOW | ✓ | ✓ | 0.6 | 2.791 | **20.0** | **0.3** |
|      | ✗ | ✓ | 0.6 | 2.000 | 32.0 | 0.7 |
|      | ✓ | ✗ | 0.6 | 2.023 | 33.3 | 0.6 |
|      | ✓ | ✓ | 1.2 | **3.440** | 31.0 | 0.6 |
| 1PQX | ✓ | ✓ | 0.6 | 2.901 | **1.2** | **0.1** |
|      | ✗ | ✓ | 0.6 | 2.001 | 4.8 | 0.1 |
|      | ✓ | ✗ | 0.6 | 2.941 | 5.1 | 0.1 |
|      | ✓ | ✓ | 1.2 | **3.170** | 5.2 | 0.1 |

*Table 10.* **Robustness to restraint sparsity on 2B5B.** We randomly subsample NOE restraints at retention levels from 10% to 100% and report the IT-Opt validity loss (Equation 38) for each. Across all levels, the validity loss is comparable to that of the PDB reference ensemble (0.002), and we observe no signs of overfitting to sparse constraints (e.g., distorted geometries).

| PDB ID | Percent retained (%) | # NOEs | Validity loss |
|--------|----------------------|--------|---------------|
| 2B5B | 10 | 9 | 0.0000 |
|      | 25 | 24 | 0.0000 |
|      | 50 | 48 | 0.0092 |
|      | 75 | 72 | 0.0007 |
|      | 100 | 96 | 0.0080 |

*Table 11.* **Quantitative evaluation of cosine similarity ($\uparrow$) for structures from Table 2.** Here, *Guidance* refers to ensembles derived using electron density guided AlphaFold3 (Maddipatla et al., 2025b), whereas *IT-Opt* refers to ensembles derived using electron density-based inference-time optimization. Entries highlighted in **green** correspond to ensembles that outperform all baseline methods, excluding the PDB, while entries highlighted in **blue** outperform all baselines, including the PDB.

| PDB ID | PDB | AlphaFold3 | AlphaFlow | Guidance | IT-Opt |
|--------|-----|------------|-----------|----------|--------|
| 5D7F:P | 0.696 | 0.435 | 0.523 | 0.638 | **0.656** |
| 6I42:B | 0.905 | 0.487 | 0.664 | 0.817 | **0.876** |
| 7ABT:B | 0.883 | 0.515 | 0.519 | 0.785 | **0.812** |
| 2DF6:C | 0.852 | 0.419 | 0.679 | 0.825 | **0.826** |
| 2DF6:D | 0.853 | 0.388 | 0.559 | 0.832 | **0.878** |
| 1DDV:B | 0.939 | 0.561 | 0.853 | 0.893 | **0.901** |
| 1CKB:B | 0.868 | 0.808 | 0.796 | 0.882 | **0.891** |
| 1CJR:A | 0.928 | 0.672 | 0.861 | 0.893 | **0.921** |
| 5G51:A | 0.884 | 0.598 | 0.502 | 0.877 | **0.898** |
| 6QQF:A | 0.878 | 0.807 | 0.829 | 0.868 | **0.869** |
| 3AZY:A | 0.847 | 0.717 | 0.721 | 0.780 | **0.848** |
| 2YNT:A | 0.885 | 0.822 | 0.871 | 0.895 | **0.905** |
| 3V3S:B | 0.767 | 0.645 | 0.731 | 0.755 | **0.808** |
| 4NPU:B | 0.832 | 0.791 | 0.785 | 0.804 | **0.828** |
| 3QDA:A | 0.893 | 0.594 | 0.584 | 0.889 | **0.891** |
| 4RMW:A | 0.904 | 0.660 | 0.687 | 0.899 | **0.900** |

*Table 12.* **Quantitative evaluation of $R_{\text{work}}$ (↓) for structures from Table 2**. Here, *Guidance* refers to ensembles derived using Electron density guided AlphaFold3 (Maddipatla et al., 2025b), whereas *IT-Opt* refers to ensembles derived using electron density-based inference-time optimization. Entries highlighted in **green** correspond to ensembles that outperform all baseline methods, excluding the PDB, while entries highlighted in **blue** outperform all baselines, including the PDB.

| PDB ID | PDB | AlphaFold3 | AlphaFlow | Guidance | IT-Opt |
|--------|--------|------------|-----------|----------|--------|
| 5D7F:P | 0.211 | 0.237 | 0.223 | 0.220 | **0.216** |
| 6I42:B | 0.169 | 0.213 | 0.222 | 0.195 | **0.175** |
| 7ABT:B | 0.169 | 0.208 | 0.216 | 0.200 | **0.179** |
| 2DF6:C | 0.174 | 0.254 | 0.207 | 0.188 | **0.183** |
| 2DF6:D | 0.174 | 0.211 | 0.253 | 0.179 | **0.177** |
| 1DDV:B | 0.210 | 0.263 | 0.243 | 0.233 | **0.227** |
| 1CKB:B | 0.1767 | 0.196 | 0.204 | 0.186 | **0.181** |
| 1CJR:A | 0.201 | 0.279 | 0.250 | 0.227 | **0.208** |
| 5G51:A | 0.191 | 0.205 | 0.215 | 0.192 | **0.191** |
| 6QQF:A | 0.159 | 0.163 | 0.163 | 0.161 | **0.161** |
| 3AZY:A | 0.173 | 0.174 | 0.174 | 0.174 | **0.173** |
| 2YNT:A | 0.158 | 0.160 | 0.159 | 0.158 | **0.158** |
| 3V3S:B | 0.163 | 0.164 | 0.164 | **0.162** | 0.163 |

*Table 13.* **Quantitative evaluation of $R_{\text{free}}$ (↓) for structures from Table 2**. Here, *Guidance* refers to ensembles derived using Electron density guided AlphaFold3 (Maddipatla et al., 2025b), whereas *IT-Opt* refers to ensembles derived using electron density-based inference-time optimization. Entries highlighted in **green** correspond to ensembles that outperform all baseline methods, excluding the PDB, while entries highlighted in **blue** outperform all baselines, including the PDB.

| PDB ID | PDB | AlphaFold3 | AlphaFlow | Guidance | IT-Opt |
|--------|--------|------------|-----------|----------|--------|
| 5D7F:P | 0.230 | 0.234 | 0.236 | 0.238 | **0.233** |
| 6I42:B | 0.183 | 0.231 | 0.237 | 0.212 | **0.192** |
| 7ABT:B | 0.193 | 0.228 | 0.234 | 0.219 | **0.205** |
| 2DF6:C | 0.192 | 0.270 | 0.229 | 0.207 | **0.204** |
| 2DF6:D | 0.192 | 0.231 | 0.269 | 0.198 | **0.197** |
| 1DDV:B | 0.257 | 0.289 | 0.200 | 0.274 | **0.268** |
| 1CKB:B | 0.234 | 0.259 | 0.257 | 0.243 | **0.235** |
| 1CJR:A | 0.2634 | 0.346 | 0.319 | 0.289 | **0.270** |
| 5G51:A | 0.211 | 0.223 | 0.231 | 0.215 | **0.211** |
| 6QQF:A | 0.208 | 0.211 | 0.211 | **0.205** | 0.206 |
| 3AZY:A | 0.194 | 0.195 | 0.196 | 0.195 | **0.194** |
| 2YNT:A | 0.200 | **0.191** | 0.200 | 0.200 | 0.200 |
| 3V3S:B | 0.202 | 0.202 | **0.201** | 0.202 | 0.202 |

*Table 14.* **Quantitative evaluation of cosine similarity (↑) for structures from Table 2 on additional benchmarks**. Here, *Guidance* refers to ensembles derived using electron density guided AlphaFold3 (Maddipatla et al., 2025b), whereas *IT-Opt* refers to ensembles derived using electron density-based inference-time optimization. Entries highlighted in **bold** indicate the best result.

| PDB ID | AlphaFlow | ESMFlow | AFCluster | BioEMU | Guidance | IT-Opt |
|--------|-----------|---------|-----------|--------|----------|--------|
| 5D7F:P | 0.523 | 0.532 | 0.412 | 0.568 | 0.638 | **0.656** |
| 6I42:B | 0.664 | 0.596 | 0.608 | 0.664 | 0.817 | **0.876** |
| 7ABT:B | 0.519 | 0.522 | 0.551 | 0.647 | 0.785 | **0.812** |
| 2DF6:C | 0.679 | 0.654 | 0.622 | 0.678 | 0.825 | **0.826** |
| 2DF6:D | 0.559 | 0.466 | 0.730 | 0.495 | 0.832 | **0.878** |
| 1DDV:B | 0.853 | 0.709 | 0.761 | 0.821 | 0.893 | **0.901** |
| 1CKB:B | 0.796 | 0.852 | 0.802 | 0.849 | 0.882 | **0.891** |
| 1CJR:A | 0.861 | 0.768 | 0.817 | 0.700 | 0.893 | **0.921** |
| 5G51:A | 0.502 | 0.535 | 0.494 | 0.669 | 0.877 | **0.898** |
| 6QQF:A | 0.829 | 0.872 | 0.832 | **0.888** | 0.868 | 0.869 |
| 3AZY:A | 0.721 | 0.717 | 0.576 | 0.739 | 0.780 | **0.848** |
| 2YNT:A | 0.871 | 0.873 | 0.814 | 0.888 | 0.895 | **0.905** |
| 3V3S:B | 0.731 | 0.750 | 0.617 | 0.728 | 0.755 | **0.800** |

*Table 15.* **Quantitative evaluation of $R_{\mathrm{work}}$ (↓) for structures from Table 2**. Here, *Guidance* refers to ensembles derived using Electron density guided AlphaFold3 (Maddipatla et al., 2025b), whereas *IT-Opt* refers to ensembles derived using electron density-based inference-time optimization. Entries highlighted in **bold** indicate the best result.

| PDB ID | AlphaFlow | ESMFlow | AFCluster | BioEMU | Guidance | IT-Opt |
|--------|-----------|---------|-----------|--------|----------|--------|
| 5D7F:P | 0.223 | 0.228 | 0.220 | 0.226 | 0.220 | **0.216** |
| 6I42:B | 0.222 | 0.228 | 0.213 | 0.223 | 0.195 | **0.175** |
| 7ABT:B | 0.216 | 0.218 | 0.204 | 0.213 | 0.200 | **0.179** |
| 2DF6:C | 0.207 | 0.208 | 0.195 | 0.209 | 0.188 | **0.183** |
| 2DF6:D | 0.253 | 0.257 | 0.210 | 0.260 | 0.179 | **0.177** |
| 1DDV:B | 0.243 | 0.268 | 0.249 | 0.251 | 0.233 | **0.227** |
| 1CKB:B | 0.204 | 0.196 | 0.203 | 0.193 | 0.186 | **0.181** |
| 1CJR:A | 0.250 | 0.274 | 0.239 | 0.324 | 0.227 | **0.208** |
| 5G51:A | 0.215 | 0.214 | 0.209 | 0.211 | 0.192 | **0.191** |
| 6QQF:A | 0.163 | 0.163 | 0.162 | 0.165 | **0.161** | 0.161 |
| 3AZY:A | 0.174 | 0.179 | 0.180 | 0.175 | 0.174 | **0.173** |
| 2YNT:A | 0.159 | 0.161 | 0.160 | 0.160 | **0.158** | 0.158 |
| 3V3S:B | 0.164 | 0.163 | 0.163 | 0.165 | **0.162** | 0.163 |

*Table 16.* **Quantitative evaluation of $R_{\text{free}}$ (↓) for structures from Table 2.** Here, *Guidance* refers to ensembles derived using Electron density guided AlphaFold3 (Maddipatla et al., 2025b), whereas *IT-Opt* refers to ensembles derived using electron density-based inference-time optimization. Entries highlighted in **bold** indicate the best result.

| PDB ID | AlphaFlow | ESMFlow | AFCluster | BioEMU | Guidance | IT-Opt |
|---|---|---|---|---|---|---|
| 5D7F:P | 0.236 | 0.238 | 0.230 | 0.242 | 0.238 | **0.233** |
| 6I42:B | 0.237 | 0.248 | 0.225 | 0.244 | 0.212 | **0.192** |
| 7ABT:B | 0.234 | 0.248 | 0.221 | 0.227 | 0.219 | **0.205** |
| 2DF6:C | 0.229 | 0.229 | 0.214 | 0.231 | 0.207 | **0.204** |
| 2DF6:D | 0.269 | 0.277 | 0.228 | 0.277 | 0.198 | **0.197** |
| 1DDV:B | **0.200** | 0.306 | 0.287 | 0.280 | 0.274 | 0.268 |
| 1CKB:B | 0.257 | 0.257 | 0.263 | 0.253 | 0.243 | **0.235** |
| 1CJR:A | 0.319 | 0.345 | 0.308 | 0.372 | 0.289 | **0.270** |
| 5G51:A | 0.231 | 0.218 | 0.218 | 0.217 | 0.215 | **0.211** |
| 6QQF:A | 0.211 | 0.208 | 0.209 | 0.208 | **0.205** | 0.206 |
| 3AZY:A | 0.196 | 0.196 | 0.195 | 0.196 | 0.195 | **0.194** |
| 2YNT:A | **0.200** | 0.201 | **0.200** | 0.201 | **0.200** | **0.200** |
| 3V3S:B | 0.201 | **0.196** | 0.201 | 0.204 | 0.200 | 0.202 |

*Table 17.* **MolProbity scores for structures from Table 2.** For each method, we report the MolProbity score (MolP), clash score (Clash), and Ramachandran outlier percentage (Rama) determined using `phenix.molprobity` (Chen et al., 2010; Williams et al., 2018); lower is better for all three metrics. The best value per protein and metric is shown in **bold**. Here, *Guidance* refers to ensembles derived using Electron density guided AlphaFold3 (Maddipatla et al., 2025b), whereas *IT-Opt* refers to ensembles derived using electron density-based inference-time optimization.

| | PDB | | | AlphaFold3 | | | Guidance | | | IT-Opt | | |
|---|---|---|---|---|---|---|---|---|---|---|---|---|
| PDB ID | MolP | Clash | Rama | MolP | Clash | Rama | MolP | Clash | Rama | MolP | Clash | Rama |
| 5D7F:P | **1.5** | **6.6** | **0.0** | 2.7 | 78.4 | 0.0 | 2.1 | 27.3 | 0.0 | 1.8 | 14.0 | 0.0 |
| 6I42:B | **1.3** | **4.8** | **0.0** | 2.4 | 68.4 | 0.0 | 2.3 | 27.3 | 0.0 | 1.6 | 11.7 | 0.0 |
| 7ABT:B | **1.3** | **2.2** | **0.0** | 2.1 | 29.9 | 0.0 | 2.7 | 38.8 | 0.0 | 2.2 | 15.1 | 0.0 |
| 2DF6:C | **1.4** | **7.3** | **0.0** | 3.3 | 190.8 | 0.0 | 2.0 | 22.2 | 0.0 | 1.6 | 12.9 | 0.0 |
| 2DF6:D | **1.4** | **7.3** | **0.0** | 2.1 | 40.5 | 0.0 | 1.9 | 22.7 | 0.0 | 1.9 | 18.5 | 0.0 |
| 1DDV:B | **1.3** | **1.7** | **0.0** | 2.0 | 13.2 | 0.0 | 1.9 | 7.6 | 0.0 | 1.9 | 9.8 | 0.0 |
| 1CKB:B | **1.0** | **1.0** | **0.0** | 1.6 | 7.5 | 0.0 | 1.6 | 9.0 | 0.0 | 1.8 | 8.3 | 0.0 |
| 1CJR:A | **1.7** | **7.9** | **0.0** | 2.8 | 53.1 | 0.0 | 2.2 | 28.1 | 0.0 | 2.0 | 16.9 | 1.8 |
| 5G51:A | **1.6** | 8.8 | **0.0** | 1.9 | 27.3 | 0.0 | 1.8 | 11.4 | 0.0 | 1.7 | 13.4 | 0.0 |
| 6QQF:A | 1.3 | 3.6 | **0.0** | **1.2** | **3.5** | 0.0 | 1.2 | 3.6 | 0.0 | 1.3 | 4.5 | 0.0 |
| 3AZY:A | 1.3 | **2.2** | 0.1 | 1.4 | 2.6 | 0.0 | 1.4 | 2.7 | 0.0 | 1.4 | 2.9 | **0.0** |
| 2YNT:A | 1.3 | 3.6 | 0.8 | 1.3 | **2.9** | 0.9 | 1.3 | **2.9** | 0.9 | 1.3 | **2.9** | 0.6 |
| 3V3S:B | 1.3 | **3.8** | 0.2 | 1.3 | 3.7 | 0.2 | 1.3 | 5.0 | 0.2 | 1.5 | 4.7 | 0.2 |

*Table 18.* **RMSD (↓) and cosine similarity (↑) with respect to deposited PDB structures on homologous proteins where AlphaFold3 exhibits mispredictions.** We evaluate IT-Opt against Guidance and AlphaFold3 on a pair of homologous proteins ($\geq$ 280 residues). Without providing any pre-modeled structures (i.e., optimize entire protein with $F_o$), IT-Opt achieves lower RMSD wrt PDB conformation and higher cosine similarity than baseline methods *at mispredicted regions*. **Bold** values indicate the best result.

| | | RMSD wrt PDB | | | Cosine similarity | | |
|---|---|---|---|---|---|---|---|
| PDB ID | Seq Len | IT-Opt | Guidance | AF3 | IT-Opt | Guidance | AF3 |
| 4NE4 | 286 | **0.217** | 0.248 | 0.230 | **0.724** | 0.711 | 0.717 |
| 5TEU | 301 | **0.194** | 0.356 | 1.302 | **0.783** | 0.762 | 0.725 |

*Table 19.* **Runtime analysis across all baselines on four PDB targets**. For each method we report peak GPU memory usage (MiB, ↓) and wall-clock time per optimization loop (seconds, ↓), along with the NOE violation percentage (Viol %, ↓) measured on the final ensemble. All runs use a batch size of 16. Bold values indicate the best result per metric.

| | | | Guidance (Uniform) | | | IT-Optimization (Uniform) | | | AlphaFlow | | | AlphaFold3 | | |
|---|---|---|---|---|---|---|---|---|---|---|---|---|---|---|
| PDB ID | #NOEs | Seq len | Memory (MiB) | Time/loop (s) | Viol % | Memory (MiB) | Time/loop (s) | Viol % | Memory (MiB) | Time/loop (s) | Viol % | Memory (MiB) | Time/loop (s) | Viol % |
| 2B3W | 1214 | 168 | 11898 | 716 | 13.8 | 13918 | 800 | **7.1** | **7466** | 217 | 37.8 | 9330 | **176** | 39.8 |
| 2M47 | 1549 | 163 | 11468 | 692 | 6.1 | 13438 | 769 | **2.9** | **7286** | 217 | 18.1 | 9078 | **180** | 16.8 |
| 2LF2 | 1853 | 175 | 12642 | 772 | 7.4 | 14802 | 830 | **3.8** | **8770** | 336 | 19.6 | 11960 | **208** | 17.8 |
| 2L06 | 1693 | 155 | 10958 | 718 | 10.3 | 12802 | 777 | **7.2** | **6946** | 210 | 29.2 | 9012 | **180** | 24.8 |

*Table 20.* **Compute-normalized comparison of Guidance and IT-Optimization at matched denoiser-call budgets on four PDB targets**. We report the NOE violation percentage (Viol %, ↓) under two compute budgets: 200 denoiser calls (top) and 800 (equivalent to 4 outer loops in IT-Opt) denoiser calls (bottom). Bold values indicate the best result per row.

| PDB ID | #NOEs | seq len | # Denoiser Calls | Guidance Viol % | IT-Opt Viol % |
|---|---|---|---|---|---|
| 2B3W | 1214 | 168 | | **13.7** | 14.4 |
| 2M47 | 1549 | 163 | 200 | **5.8** | 6.2 |
| 2LF2 | 1853 | 175 | | **7.3** | 8.7 |
| 2L06 | 1693 | 155 | | **9.8** | 11.8 |
| 2B3W | 1214 | 168 | | 13.7 | **8.4** |
| 2M47 | 1549 | 163 | 800 | 5.8 | **3.4** |
| 2LF2 | 1853 | 175 | | 7.3 | **4.8** |
| 2L06 | 1693 | 155 | | 9.8 | **8.3** |

*Table 21.* **Inter-chain hydrogen-bond recovery in the `1YCS` complex.** This table summarizes hydrogen-bond donor-acceptor pairs observed in the crystal structure of `1YCS` and compares their recovery across sampled predictions. The AlphaFold3 column reports the number of AlphaFold3 samples (out of 5) in which a given hydrogen bond is present, while IT-Opt reports the corresponding count for ipTM-based inference-time optimization. $\Delta$ denotes the difference (IT-Opt − AlphaFold3). Across all listed contacts, unguided AlphaFold3 recovers 16 out of 50 possible hydrogen-bond occurrences (average recovery rate 0.32), whereas ipTM-based inference-time optimization recovers 23 out of 50 (average recovery rate 0.46), corresponding to an absolute increase of 0.14, and a relative enrichment of 44%.

| Donor | Acceptor | AlphaFold3 | IT-Opt | $\Delta$ |
|---|---|---|---|---|
| A:ARG280:NH1 | B:GLU493:O | 0/5 | 3/5 | +3 |
| A:ARG248:NH1 | B:GLU495:OE2 | 4/5 | 5/5 | +1 |
| B:ASN473:ND2 | A:ASN247:O | 0/5 | 1/5 | +1 |
| B:TRP498:NE1 | A:SER241:O | 4/5 | 5/5 | +1 |
| A:ARG248:NH2 | B:ASP475:OD1 | 4/5 | 5/5 | +1 |
| A:ASN247:ND2 | B:TYR469:OH | 0/5 | 1/5 | +1 |
| A:HIS178:NE2 | B:MET422:O | 0/5 | 0/5 | 0 |
| A:ARG248:NH1 | B:ASP494:OD2 | 0/5 | 0/5 | 0 |
| A:SER183:N | B:SER425:O | 0/5 | 0/5 | 0 |
| B:ASN473:ND2 | A:ARG248:O | 4/5 | 3/5 | −1 |

*Table 22.* **Inter-chain hydrogen-bond recovery in the `2LY4` complex.** This table summarizes hydrogen-bond donor-acceptor pairs observed in the NMR-determined structure of `2LY4` at the helical region in the peptide (highlighted in Figure 11, Panel B), and compares their recovery across sampled predictions. The AlphaFold3 column reports the number of AlphaFold3 samples (out of 100) in which a given hydrogen bond is present, while IT-Opt reports the corresponding count for ipTM-based inference-time optimization. $\Delta$ denotes the difference (IT-Opt − AlphaFold3). Across the listed contacts, unguided AlphaFold3 recovers 74/900 hydrogen-bond occurrences (0.082), whereas ipTM-based inference-time optimization recovers 116/900 (0.129), corresponding to a relative enrichment of 57%.

| Donor | Acceptor | AlphaFold3 | IT-Opt | $\Delta$ |
|---|---|---|---|---|
| A:LYS27:NZ | B:ASP57:OD2 | 8/100 | 21/100 | +13 |
| A:ARG23:NE | B:PHE54:O | 11/100 | 22/100 | +11 |
| A:LYS27:NZ | B:GLU56:OE2 | 10/100 | 17/100 | +7 |
| A:ARG23:NE | B:TRP53:O | 20/100 | 24/100 | +4 |
| A:ARG23:NE | B:GLU56:OE2 | 9/100 | 12/100 | +3 |
| A:GLY1:N | B:PRO60:O | 0/100 | 1/100 | +1 |
| A:ARG9:NH1 | B:TRP53:O | 16/100 | 17/100 | +1 |
| A:LYS27:NZ | B:TRP53:O | 0/100 | 1/100 | +1 |
| A:GLY1:N | B:GLY59:O | 0/100 | 1/100 | +1 |

## A.3. Pseudocodes

---

**Algorithm 2** Inference-time Optimization

---

**Input:** Sequence $\mathbf{a}$; input features $\{\mathbf{f}^*\}$, $\{\mathbf{s}_i^{\text{inputs}}\}$; initial trunk embeddings $\mathbf{Z}_{\text{init}}$; experimental observation $\mathbf{y}$; noise schedule $[\sigma_0, \sigma_1, \ldots, \sigma_T]$; noise factor $\gamma_0 = 0.8$, minimum noise factor $\gamma_{\min} = 1.0$; noise scale $\lambda$; step scale $\kappa$; ensemble size $n$; outer iterations $K$; inner MSA steps $M = 1$; learning rate $\eta_z$; prior weight $\lambda_p$

**Output:** Optimized batch trunk embeddings $\mathcal{Z}$

$\mathcal{Z} = [\mathbf{Z}_{\text{init}}, \ldots \mathbf{Z}_{\text{init}}]$ $\qquad\qquad\qquad\qquad\qquad\qquad\qquad$ ▷ Initialize from Pairformer

**for** $k = 1$ to $K$ $\qquad\qquad\qquad\qquad\qquad\qquad\qquad\qquad\qquad\qquad$ ▷ Outer optimization loops

$\quad \mathcal{X}_l \sim \sigma_T \cdot [\mathbf{N}^1, \ldots, \mathbf{N}^n]^T$ $\qquad\qquad\qquad\qquad$ $\mathbf{N}^i \sim \mathcal{N}(\mathbf{0}, \mathbf{I}), \mathcal{X}_l \in \mathbb{R}^{n \times m \times 3}$

$\quad$ **for** $\sigma_\tau \in [\sigma_{T-1}, \ldots, \sigma_0]$ $\qquad\qquad\qquad\qquad\qquad\qquad$ ▷ Reverse diffusion schedule

$\quad\quad \mathcal{X}_l \leftarrow \text{CentreRandomAugmentation}(\mathcal{X}_l)$

$\quad\quad \gamma \leftarrow \gamma_0$ if $\sigma_\tau > \gamma_{\min}$ else $0$

$\quad\quad \hat{t} \leftarrow \dfrac{\sigma_{\tau-1}(\gamma + 1)}{\sqrt{t^2 - \sigma_\tau^2/\sigma_{\tau-1}}}$

$\quad\quad \boldsymbol{\xi}_l \leftarrow \lambda\sqrt{\hat{t}^2 - \sigma_\tau^2} \cdot [\mathbf{N}^1, \ldots, \mathbf{N}^n]^T$ $\qquad\qquad\qquad\qquad\qquad$ $\mathbf{N}^i \sim \mathcal{N}(\mathbf{0}, \mathbf{I})$

$\quad\quad \mathcal{X}_l^{\text{noisy}} \leftarrow \mathcal{X}_l + \boldsymbol{\xi}_l$

$\quad\quad$ **for** $j = 1$ to $M$ $\qquad\qquad\qquad\qquad\qquad\qquad\qquad$ ▷ Inner embedding optimization

$\quad\quad\quad \hat{\mathcal{X}}_0 \leftarrow \text{DiffusionModule}(\{\mathcal{X}_l^{\text{noisy}}\}, \hat{t}, \{\mathbf{f}^*\}, \{\mathbf{s}_i^{\text{inputs}}\}, \mathcal{Z})$

$\quad\quad\quad \mathcal{Z} \leftarrow \mathcal{Z} + \nabla_{\mathcal{Z}}(\eta_z \log p(\mathbf{y}|\hat{\mathcal{X}}_0) + \lambda_p \log p(\mathcal{Z}|\mathbf{a}))$ $\qquad$ ▷ Embedding update

$\quad\quad$ **end for**

$\quad\quad \hat{\mathcal{X}}_0 \leftarrow \text{DiffusionModule}(\{\mathcal{X}_l^{\text{noisy}}\}, \hat{t}, \{\mathbf{f}^*\}, \{\mathbf{s}_i^{\text{inputs}}\}, \mathcal{Z})$

$\quad\quad \delta_l \leftarrow (\mathcal{X}_l - \hat{\mathcal{X}}_0)/\hat{t}$

$\quad\quad dt \leftarrow \beta_\tau - \hat{t}$

$\quad\quad \mathcal{X}_l \leftarrow \mathcal{X}_l^{\text{noisy}} + \kappa \cdot dt \cdot \delta_l$ $\qquad\qquad\qquad\qquad\qquad\qquad$ ▷ Reverse diffusion step

$\quad$ **end for**

**end for**

**return** $\mathcal{Z}$

---

---

**Algorithm 3** Inference-time Optimization with Boltzmann Reweighting

---

**Input:** Same as Algorithm 2, plus: energy function $E_\phi \colon \mathbb{R}^{m \times 3} \to \mathbb{R}$; inverse temperature $\beta = 1.68 \, \text{kcal}^{-1}\text{mol}$
**Output:** Optimized batch trunk embeddings $\mathcal{Z}$
$\mathcal{Z} = [\mathbf{Z}_{\text{init}}, \ldots \mathbf{Z}_{\text{init}}]$      $\triangleright$ Initialize from Pairformer
**for** $k = 1$ to $K$      $\triangleright$ Outer optimization loops
    $\mathcal{X}_l \sim \sigma_T \cdot [\mathbf{N}^1, \ldots, \mathbf{N}^n]^T$      $\mathbf{N}^i \sim \mathcal{N}(\mathbf{0}, \mathbf{I}), \mathcal{X}_l \in \mathbb{R}^{n \times m \times 3}$
    **for** $\sigma_\tau \in [\sigma_{T-1}, \ldots, \sigma_0]$      $\triangleright$ Reverse diffusion schedule
        $\mathcal{X}_l \leftarrow \text{CentreRandomAugmentation}(\mathcal{X}_l)$
        $\gamma \leftarrow \gamma_0$ if $\sigma_\tau > \gamma_{\min}$ else $0$
        $\hat{t} \leftarrow \dfrac{\sigma_{\tau-1}(\gamma + 1)}{\sqrt{t^2 - \sigma_\tau^2/\sigma_{\tau-1}}}$
        $\boldsymbol{\xi}_l \leftarrow \lambda \sqrt{\hat{t}^2 - \sigma_\tau^2} \cdot [\mathbf{N}^1, \ldots, \mathbf{N}^n]^T$      $\mathbf{N}^i \sim \mathcal{N}(\mathbf{0}, \mathbf{I})$
        $\mathcal{X}_l^{\text{noisy}} \leftarrow \mathcal{X}_l + \boldsymbol{\xi}_l$
        **for** $j = 1$ to $M$      $\triangleright$ Inner optimization loop with reweighting
            $\hat{\mathcal{X}}_0 \leftarrow \text{DiffusionModule}(\{\mathcal{X}_l^{\text{noisy}}\}, \hat{t}, \{\mathbf{f}^*\}, \{\mathbf{s}_i^{\text{inputs}}\}, \mathcal{Z})$
            $w^i \leftarrow \dfrac{\exp(-\beta E_\phi(\hat{\mathbf{X}}_0^i))}{\sum_{k=1}^n \exp(-\beta E_\phi(\hat{\mathbf{X}}_0^k))}$ for $i \in [n]$      $\triangleright$ Boltzmann weights
            $\mathcal{Z} \leftarrow \mathcal{Z} + \nabla_{\mathcal{Z}}(\eta_z \log p(\mathbf{y}|\hat{\mathcal{X}}_0; \mathbf{w}) + \lambda_p \log p(\mathcal{Z}|\mathbf{a}))$      $\triangleright$ Embedding update
        **end for**
        $\hat{\mathcal{X}}_0 \leftarrow \text{DiffusionModule}(\{\mathcal{X}_l^{\text{noisy}}\}, \hat{t}, \{\mathbf{f}^*\}, \{\mathbf{s}_i^{\text{inputs}}\}, \mathcal{Z})$
        $\delta_l \leftarrow (\mathcal{X}_l - \hat{\mathcal{X}}_0)/\hat{t}$
        $dt \leftarrow \beta_\tau - \hat{t}$
        $\mathcal{X}_l \leftarrow \mathcal{X}_l^{\text{noisy}} + \kappa \cdot dt \cdot \delta_l$      $\triangleright$ Reverse diffusion step
    **end for**
**end for**
**return** $\mathcal{Z}$

---

## B. Theoretical analysis of IT-Optimization

We provide a formal convergence analysis of the inference-time optimization procedure described in Algorithm 2, modeling the outer loop as Biased Stochastic Gradient Ascent (BSGA) (Asi et al., 2021). Specifically, we study the convergence properties of inference-time updates to the diffusion model's input embeddings under a surrogate objective defined as the diffusion trajectory-averaged log-likelihood. Assuming local Lipschitz smoothness and bounded inner-loop updates – conditions justified by a Gaussian prior and gradient clipping that restrict optimization to a compact region with finite curvature – we formalize IT-Opt as biased stochastic gradient ascent consistent with our implementation. The dominant estimator bias, introduced by the stochastic denoiser, is bounded via the Cauchy-Schwarz inequality in terms of the Fisher information trace, while variance is controlled through inner-loop early stopping. Despite the presence of bias, we establish an $\mathcal{O}(1/K)$ convergence rate to a near-stationary region after $K$ outer iterations and derive a sufficient condition for expected monotonic ascent.

### B.1. Formalizing the Surrogate Objective and its True Gradient

To strictly match the evaluation mechanism of our algorithmic proxy, we formally redefine our target as the trajectory-averaged surrogate objective $\tilde{\mathcal{J}}(\boldsymbol{\mathcal{Z}})$:

$$\tilde{\mathcal{J}}(\boldsymbol{\mathcal{Z}}) \; = \; \mathbb{E}_{t, \boldsymbol{\mathcal{X}}_t | \boldsymbol{\mathcal{Z}}}\Big[\log p\big(\mathbf{y} \mid D_\theta(\boldsymbol{\mathcal{X}}_t; \boldsymbol{\mathcal{Z}}, t)\big)\Big] \; + \; \lambda_p \log p(\boldsymbol{\mathcal{Z}} \mid \mathbf{a}). \tag{11}$$

By operating directly on the denoiser predictions over the sampled diffusion schedule $t$, this surrogate objective circumvents the intractable analytic bounds of the exact evidence lower bound (ELBO) when subjected to severe, non-linear experimental log-likelihood evaluations.

To perform optimization, we compute the gradient $\nabla_{\boldsymbol{\mathcal{Z}}}\tilde{\mathcal{J}}(\boldsymbol{\mathcal{Z}})$. Applying the log-derivative trick to the conditional expectation yields the *exact* analytic gradient of the surrogate:

$$\begin{aligned}
\nabla_{\boldsymbol{\mathcal{Z}}}\tilde{\mathcal{J}}(\boldsymbol{\mathcal{Z}}) \; = \; & \mathbb{E}_{t, \boldsymbol{\mathcal{X}}_t | \boldsymbol{\mathcal{Z}}}\Big[\underbrace{\nabla_{\boldsymbol{\mathcal{Z}}} \log p\big(\mathbf{y} \mid D_\theta(\boldsymbol{\mathcal{X}}_t; \boldsymbol{\mathcal{Z}}, t)\big) \; + \; \lambda_p \nabla_{\boldsymbol{\mathcal{Z}}} \log p(\boldsymbol{\mathcal{Z}} \mid \mathbf{a})}_{\text{Our Computed Proxy Gradient } (\hat{g}_t)}\Big] \\
& + \; \mathbb{E}_{t, \boldsymbol{\mathcal{X}}_t | \boldsymbol{\mathcal{Z}}}\Big[\underbrace{\log p\big(\mathbf{y} \mid D_\theta(\boldsymbol{\mathcal{X}}_t; \boldsymbol{\mathcal{Z}}, t)\big)\nabla_{\boldsymbol{\mathcal{Z}}} \log p_t(\boldsymbol{\mathcal{X}}_t \mid \boldsymbol{\mathcal{Z}})}_{\text{Dropped REINFORCE Score Term Bias}}\Big]
\end{aligned} \tag{12}$$

Algorithm 2 purposefully utilizes pathwise (backpropagation) automatic differentiation strictly through the differentiable denoiser operator $D_\theta$. This computational choice generates the tractable proxy gradient defined in Equation (13), purposefully dropping the reinforcing score dynamics dependent on the sampling distribution. The nature of this resulting bias is algebraically characterized in Lemma B.1.

### B.2. Formalizing the Proxy Gradient and Sources of Bias

In Algorithm 2, the inner loop computes a proxy gradient using the pretrained denoiser network $D_\theta(\boldsymbol{\mathcal{X}}_t; \boldsymbol{\mathcal{Z}}, t)$. Let $\hat{g}_t(\boldsymbol{\mathcal{Z}})$ denote the stochastic gradient estimator evaluated at a single diffusion timestep $t$:

$$\hat{g}_t(\boldsymbol{\mathcal{Z}}) \; = \; \nabla_{\boldsymbol{\mathcal{Z}}} \log p\big(\mathbf{y} \mid D_\theta(\boldsymbol{\mathcal{X}}_t; \boldsymbol{\mathcal{Z}}, t)\big) \; + \; \lambda_p \nabla_{\boldsymbol{\mathcal{Z}}} \log p(\boldsymbol{\mathcal{Z}} \mid \mathbf{a}). \tag{13}$$

**Lemma B.1** (Sources of Estimator Bias)**.** *The stochastic estimator $\hat{g}_t(\boldsymbol{\mathcal{Z}})$ defined in* (13) *is a* biased *estimator of the true gradient $\nabla_{\boldsymbol{\mathcal{Z}}}\tilde{\mathcal{J}}(\boldsymbol{\mathcal{Z}})$.*

*Proof.* Because the marginal distribution $p_t(\boldsymbol{\mathcal{X}}_t \mid \boldsymbol{\mathcal{Z}})$ depends on the conditioning vector $\boldsymbol{\mathcal{Z}}$, the linear gradient operator $\nabla_{\boldsymbol{\mathcal{Z}}}$ does not commute with the sampling expectation $\mathbb{E}_{\boldsymbol{\mathcal{X}}_t | \boldsymbol{\mathcal{Z}}}$. Differentiating purely via pathwise backpropagation exactly omits how parameter shifts in $\boldsymbol{\mathcal{Z}}$ influence the underlying base sampling densities. The explicit formulation of this omission yields the REINFORCE score-function product (Williams, 1992): $\mathbb{E}_{\boldsymbol{\mathcal{X}}_t | \boldsymbol{\mathcal{Z}}}\big[\log p(\mathbf{y} \mid D_\theta(\boldsymbol{\mathcal{X}}_t; \boldsymbol{\mathcal{Z}}, t)) \nabla_{\boldsymbol{\mathcal{Z}}} \log p_t(\boldsymbol{\mathcal{X}}_t \mid \boldsymbol{\mathcal{Z}})\big]$.

Our optimization procedure systematically drops this computationally unstable term, trading rigorous unbiasedness for scalable, lower-variance structural sampling trajectories targeting the proxy gradient. Consequently, $\mathbb{E}_{t, \boldsymbol{\mathcal{X}}_t | \boldsymbol{\mathcal{Z}}}[\hat{g}_t(\boldsymbol{\mathcal{Z}})] \neq \nabla_{\boldsymbol{\mathcal{Z}}}\tilde{\mathcal{J}}(\boldsymbol{\mathcal{Z}})$, yielding the dropped score term as the explicit score-function source of bias before accounting for the additional nested-drift bias analyzed below. $\square$

Since $\hat{G}(\boldsymbol{\mathcal{Z}}^k)$ aggregates these individually biased estimators $\hat{g}_t$ sequentially over the inner loop evaluations, the composite optimization oracle structurally inherits a boundedly finite systematic target bias, formalized by $\mathcal{B}(\boldsymbol{\mathcal{Z}}) \neq 0$.

**Definition B.2** (Oracle Displacement). We formally define the net embedding displacement oracle $\hat{G}(\boldsymbol{\mathcal{Z}}^k)$ produced by the inner loop at outer iteration $k$:

$$\hat{G}(\boldsymbol{\mathcal{Z}}^k) \;=\; \frac{\boldsymbol{\mathcal{Z}}_0^k - \boldsymbol{\mathcal{Z}}_T^k}{\eta_{\mathrm{inner}}} \;=\; \frac{1}{\eta_{\mathrm{inner}}} \sum_t \alpha_t \hat{g}_t(\boldsymbol{\mathcal{Z}}_t^k, \boldsymbol{\mathcal{X}}_t) \tag{14}$$

where the weights $\alpha_t = \eta_z$ correspond to the inner-loop step size, making the effective inner-loop integration horizon $\eta_{\mathrm{inner}} = T\eta_z$.

Furthermore, because the outer loop updates by carrying forward the final optimized embedding (equivalent to the update $\boldsymbol{\mathcal{Z}}^{k+1} = \boldsymbol{\mathcal{Z}}^k + \eta_{\mathrm{inner}}\hat{G}$), the effective outer-loop learning rate governing the ascent is exactly $\eta = \eta_{\mathrm{inner}}$. Instead of empirically assuming this oracle has bounded variance and bias, we derive these critical properties directly from the problem geometry.

## B.3. Derivation of the Oracle Properties: Nested Optimization and Stochasticity

The displacement $\hat{G}(\boldsymbol{\mathcal{Z}}^k)$ is accumulated over multiple inner-loop steps evaluated at dynamically changing non-stationary coordinates $\boldsymbol{\mathcal{Z}}_t$. To rigorously deploy the theory of Biased Stochastic Gradient Ascent for the outer loop, we must mathematically account for the *tracking drift* originating from the nested updates, as well as the variance injected by the denoiser stochasticity $\boldsymbol{\mathcal{X}}_t \sim p_t(\boldsymbol{\mathcal{X}}_t \mid \boldsymbol{\mathcal{Z}})$.

**Assumption B.3** (Local Lipschitz Smoothness of the Proxy Gradient). We assume the proxy gradient $\hat{g}_t(\boldsymbol{\mathcal{Z}}, \boldsymbol{\mathcal{X}}_t)$ is locally $L_Z$-Lipschitz almost surely with respect to the embedding $\boldsymbol{\mathcal{Z}}$ on the compact sublevel set $\{\boldsymbol{\mathcal{Z}} : \|\boldsymbol{\mathcal{Z}} - \boldsymbol{\mathcal{Z}}_0\| \leq R\}$ visited by the initialized algorithm:

$$\|\hat{g}_t(\boldsymbol{\mathcal{Z}}_1, \boldsymbol{\mathcal{X}}_t) - \hat{g}_t(\boldsymbol{\mathcal{Z}}_2, \boldsymbol{\mathcal{X}}_t)\| \;\leq\; L_Z\|\boldsymbol{\mathcal{Z}}_1 - \boldsymbol{\mathcal{Z}}_2\| \quad a.s. \tag{15}$$

and locally $L_X$-Lipschitz with respect to the noisy coordinate $\boldsymbol{\mathcal{X}}_t$:

$$\|\hat{g}_t(\boldsymbol{\mathcal{Z}}, \boldsymbol{\mathcal{X}}_1) - \hat{g}_t(\boldsymbol{\mathcal{Z}}, \boldsymbol{\mathcal{X}}_2)\| \;\leq\; L_X\|\boldsymbol{\mathcal{X}}_1 - \boldsymbol{\mathcal{X}}_2\|. \tag{16}$$

For any non-differentiable criteria like the $L_1$ norm in X-ray data, we assume standard continuous smoothing (e.g., pseudo-Huber penalty) such that its gradients are globally Lipschitz continuous. Furthermore, we assume standard gradient clipping inner-loop bounding limits the maximum per-step update divergence strictly by a constant $C_{\mathrm{clip}}$.

**Lemma B.4** (Variance from Denoiser Stochasticity). *Under Assumption B.3, the variance of the inner-loop estimator $\hat{G}(\boldsymbol{\mathcal{Z}}^k)$ is bounded proportionally to the diffusion noise scale.*

*Proof.* By the Law of Total Variance conditioned on the clean structure $\boldsymbol{\mathcal{X}}_0$, the variance of the estimator decomposes as $\mathrm{Var}(\hat{g}_t(\boldsymbol{\mathcal{Z}})) = \mathbb{E}_{\boldsymbol{\mathcal{X}}_0}[\mathrm{Var}_{\boldsymbol{\mathcal{X}}_t|\boldsymbol{\mathcal{X}}_0}(\hat{g}_t(\boldsymbol{\mathcal{Z}}))] + \mathrm{Var}_{\boldsymbol{\mathcal{X}}_0}(\mathbb{E}_{\boldsymbol{\mathcal{X}}_t|\boldsymbol{\mathcal{X}}_0}[\hat{g}_t(\boldsymbol{\mathcal{Z}})])$. For the first term, by the $L_X$-Lipschitz condition (Assumption B.3), the variance with respect to the Gaussian transition kernel $p(\boldsymbol{\mathcal{X}}_t \mid \boldsymbol{\mathcal{X}}_0)$ is bounded by $L_X^2\sigma_t^2$, where $\sigma_t^2$ is the marginal noise variance at step $t$. For the second term, we assume a bounding constant $V_{\mathrm{struct}}$ to capture the intrinsic structural heterogeneity. This is well-posed because the unperturbed coordinates evaluated by the neural network natively reside on a compact manifold of valid protein structures, bounding the variance of the gradients over the clean domain:

$$\mathrm{Var}(\hat{g}_t(\boldsymbol{\mathcal{Z}})) \;\leq\; L_X^2\sigma_t^2 \;+\; V_{\mathrm{struct}}. \tag{17}$$

The outer-loop oracle $\hat{G}$ is a convex combination of the per-step estimators $\hat{G} = \sum_t w_t\hat{g}_t$ with weights $w_t = \alpha_t/\eta_{\mathrm{inner}}$ summing to 1. Defining the variance of a vector estimator as the expected squared deviation from its mean, $\mathrm{Var}(X) = \mathbb{E}\|X - \mathbb{E}X\|^2$, and the covariance as the expected inner product, the Cauchy-Schwarz inequality holds boundedly. The total variance of the convex combination is bounded by the supremum of the individual variances: $\mathrm{Var}(\sum_t w_t\hat{g}_t) = \sum_{t,s} w_t w_s \mathrm{Cov}(\hat{g}_t, \hat{g}_s) \leq \sum_{t,s} w_t w_s |\mathrm{Cov}(\hat{g}_t, \hat{g}_s)| \leq \sum_{t,s} w_t w_s \sqrt{\mathrm{Var}(\hat{g}_t)\mathrm{Var}(\hat{g}_s)} \leq \sup_t \mathrm{Var}(\hat{g}_t)(\sum_t w_t)^2 = \sup_t \mathrm{Var}(\hat{g}_t)$. Thus, the total variance satisfies:

$$\mathbb{E}\left[\left\|\hat{G}(\boldsymbol{\mathcal{Z}}^k) - \mathbb{E}[\hat{G}(\boldsymbol{\mathcal{Z}}^k)]\right\|^2\right] \;\leq\; L_X^2 \sup_{t \leq t_{\mathrm{stop}}} \sigma_t^2 \;+\; V_{\mathrm{struct}} \;:=\; \sigma^2. \tag{18}$$

While the SDE noise scale $\sigma_t^2$ grows large as $t \to T$, dynamically executing the inner-loop inference over a restricted integration horizon (stopping optimization early at $t_{\text{stop}} = 160$) explicitly truncates the supremum strictly to $\sup_{t \leq 160} \sigma_t^2$, ensuring the overall estimator variance $\sigma^2$ remains properly bounded. $\qquad\square$

**Lemma B.5** (Cauchy-Schwarz Bound on the Dropped Score Bias). *Let the stationary expected proxy displacement evaluated at the initialization be defined as $\bar{g}(\boldsymbol{\mathcal{Z}}^k) = \mathbb{E}_{t, \boldsymbol{\mathcal{X}}_t}[\hat{g}_t(\boldsymbol{\mathcal{Z}}^k, \boldsymbol{\mathcal{X}}_t)]$. The error induced exclusively by omitting the score term from the exact trajectory surrogate gradient $\nabla \tilde{\mathcal{J}}(\boldsymbol{\mathcal{Z}}^k)$ is uniformly bounded by the trace of the Fisher Information Matrix:*

$$\|\bar{g}(\boldsymbol{\mathcal{Z}}^k) - \nabla \tilde{\mathcal{J}}(\boldsymbol{\mathcal{Z}}^k)\| \leq \sup_{t \leq t_{\text{stop}}} \sqrt{\mathbb{E}_{\boldsymbol{\mathcal{X}}_t \mid \boldsymbol{\mathcal{Z}}}[(\log p(\mathbf{y} \mid D_\theta))^2]} \cdot \sqrt{\text{Tr}\left(\mathcal{I}_t(\boldsymbol{\mathcal{Z}})\right)} := B_{\text{score}}. \tag{19}$$

*Proof.* Let the per-timestep residual bias vector be exactly the dropped REINFORCE score term: $b_t = \|\mathbb{E}_{\boldsymbol{\mathcal{X}}_t \mid \boldsymbol{\mathcal{Z}}}[\log p(\mathbf{y} \mid D_\theta) \nabla_{\boldsymbol{\mathcal{Z}}} \log p_t(\boldsymbol{\mathcal{X}}_t \mid \boldsymbol{\mathcal{Z}})]\|_2$. Applying the generalized triangle inequality for expectations (or Jensen's inequality applied to the convex Euclidean norm) maps the evaluation inside the expectation:

$$b_t \leq \mathbb{E}_{\boldsymbol{\mathcal{X}}_t \mid \boldsymbol{\mathcal{Z}}}\left[\left|\log p(\mathbf{y} \mid D_\theta)\right| \cdot \left\|\nabla_{\boldsymbol{\mathcal{Z}}} \log p_t(\boldsymbol{\mathcal{X}}_t \mid \boldsymbol{\mathcal{Z}})\right\|_2\right]. \tag{20}$$

Applying the classical Cauchy-Schwarz inequality against expectations completely isolates the log-likelihood proxy magnitude from the marginal sampling sensitivities:

$$b_t \leq \sqrt{\mathbb{E}_{\boldsymbol{\mathcal{X}}_t \mid \boldsymbol{\mathcal{Z}}}[(\log p(\mathbf{y} \mid D_\theta))^2]} \cdot \sqrt{\mathbb{E}_{\boldsymbol{\mathcal{X}}_t \mid \boldsymbol{\mathcal{Z}}}\left[\left\|\nabla_{\boldsymbol{\mathcal{Z}}} \log p_t(\boldsymbol{\mathcal{X}}_t \mid \boldsymbol{\mathcal{Z}})\right\|_2^2\right]}. \tag{21}$$

Leveraging the continuous analytic formulation of the diffusion conditionals, the expected squared magnitude of the sampling score gradient collapses precisely to the generalized multidimensional identity formulation for the Fisher Information evaluated at trace capacity:

$$\mathbb{E}_{\boldsymbol{\mathcal{X}}_t \mid \boldsymbol{\mathcal{Z}}}\left[\left\|\nabla_{\boldsymbol{\mathcal{Z}}} \log p_t(\boldsymbol{\mathcal{X}}_t \mid \boldsymbol{\mathcal{Z}})\right\|_2^2\right] = \text{Tr}\left(\mathcal{I}_t(\boldsymbol{\mathcal{Z}})\right).^3 \tag{22}$$

Substituting this directly yields the resulting scalar bound over $b_t$. We take the supremum strictly over the truncated diffusion integration horizon $t \leq t_{\text{stop}}$. Because the Gaussian prior penalty and gradient clipping jointly confine the optimization trajectory of $\boldsymbol{\mathcal{Z}}$ to a compact ball, and the continuous diffusion base densities $p_t(\boldsymbol{\mathcal{X}}_t \mid \boldsymbol{\mathcal{Z}})$ inherit bounded parametrizations across this restricted domain, the Fisher information guarantees strict regularity ($\text{Tr}(\mathcal{I}_t(\boldsymbol{\mathcal{Z}})) < \infty$). Furthermore, on this confined domain, the denoiser outputs and smoothly evaluated log-likelihoods (including bounded ipTM neural confidence scores) remain strictly finite, ensuring the second moment of the log-likelihood is analytically bounded. Hence, the truncated supremum yields a strictly bounded, absolute stationary bias $B_{\text{score}}$. $\qquad\square$

**Lemma B.6** (Total Bias from Nested Drift). *Under Assumption B.3, the total expected displacement bias tracked along the dynamic nested updates, defined as $\mathcal{B}(\boldsymbol{\mathcal{Z}}^k) = \mathbb{E}[\hat{G}(\boldsymbol{\mathcal{Z}}^k)] - \nabla \tilde{\mathcal{J}}(\boldsymbol{\mathcal{Z}}^k)$, is strictly bounded.*

*Proof.* The algorithmic proxy update computes the expected displacement along the non-stationary trajectory $\{\boldsymbol{\mathcal{Z}}_t\}$ indexed by the diffusion steps:

$$\mathbb{E}[\hat{G}(\boldsymbol{\mathcal{Z}}^k)] = \frac{1}{\eta_{\text{inner}}} \sum_t \alpha_t \mathbb{E}[\hat{g}_t(\boldsymbol{\mathcal{Z}}_t, \boldsymbol{\mathcal{X}}_t)]. \tag{23}$$

We decompose this accumulated update into a stationary term around the root state vector $\boldsymbol{\mathcal{Z}}^k$ and a residual accumulation drift term:

$$\mathbb{E}[\hat{G}(\boldsymbol{\mathcal{Z}}^k)] = \underbrace{\frac{1}{\eta_{\text{inner}}} \sum_t \alpha_t \mathbb{E}[\hat{g}_t(\boldsymbol{\mathcal{Z}}^k, \boldsymbol{\mathcal{X}}_t)]}_{:=\bar{g}(\boldsymbol{\mathcal{Z}}^k)} + \underbrace{\frac{1}{\eta_{\text{inner}}} \sum_t \alpha_t \mathbb{E}[\hat{g}_t(\boldsymbol{\mathcal{Z}}_t, \boldsymbol{\mathcal{X}}_t) - \hat{g}_t(\boldsymbol{\mathcal{Z}}^k, \boldsymbol{\mathcal{X}}_t)]}_{:=\mathcal{B}_{\text{drift}}(\boldsymbol{\mathcal{Z}}^k)}. \tag{24}$$

Lemma B.5 establishes the score term bias as $\|\bar{g}(\boldsymbol{\mathcal{Z}}^k) - \nabla \tilde{\mathcal{J}}(\boldsymbol{\mathcal{Z}}^k)\| \leq B_{\text{score}}$. For the drift error $\mathcal{B}_{\text{drift}}$, since the optimization uses gradient clipping with constant $C_{\text{clip}}$ which restricts the divergence per inner-step to be well-behaved, the Lipschitz

---

[3] This standard Fisher information identity strictly holds under the assumption that the parametric family $p_t(\boldsymbol{\mathcal{X}}_t \mid \boldsymbol{\mathcal{Z}})$ satisfies standard regularity conditions (i.e., dominated convergence permitting the interchange of integration and differentiation w.r.t. $\boldsymbol{\mathcal{Z}}$). Because the diffusion marginals are analytically intractable SDEs, our regularity appeal relies strictly on the fact that $\boldsymbol{\mathcal{Z}}$ is embedded through a smooth neural network over a compact, gradient-clipped domain (Assumption B.3), which structurally ensures dominated convergence.

condition (Assumption B.3) yields an upper bound matching the maximal displacement magnitude: $\|\mathcal{B}_{\text{drift}}(\boldsymbol{\mathcal{Z}}^k)\| \leq L_Z \eta_{\text{inner}} C_{\text{clip}}$.

Finally, combining these terms using the triangle inequality bounds the total bias of the proxy oracle:

$$\|\mathcal{B}(\boldsymbol{\mathcal{Z}}^k)\| \leq B_{\text{score}} + L_Z \eta_{\text{inner}} C_{\text{clip}} := B. \tag{25}$$

$\square$

**Remark.** Lemmas B.4, B.5, and B.6 map the empirical algorithmic choices of nested optimization, gradient clipping, and score dropping directly to theoretical bounds $B$ and $\sigma^2$. Consequently, we can establish non-asymptotic outer-loop convergence guarantees identically to generic BSGA over the chosen surrogate bounds.

### B.4. Convergence Theorem

**Assumption B.7** (Local Lipschitz Smoothness of Surrogate Objective). The global surrogate objective $\tilde{\mathcal{J}}(\boldsymbol{\mathcal{Z}})$ is locally $L$-smooth in $\boldsymbol{\mathcal{Z}}$ on the compact sublevel set $\mathcal{S} = \{\boldsymbol{\mathcal{Z}} : \|\boldsymbol{\mathcal{Z}} - \boldsymbol{\mathcal{Z}}_0\| \leq R\}$ visited by the initialized algorithm: there exists a finite $L > 0$ such that for all $\boldsymbol{\mathcal{Z}}_1, \boldsymbol{\mathcal{Z}}_2 \in \mathcal{S}$,

$$\tilde{\mathcal{J}}(\boldsymbol{\mathcal{Z}}_2) \geq \tilde{\mathcal{J}}(\boldsymbol{\mathcal{Z}}_1) + \langle \nabla \tilde{\mathcal{J}}(\boldsymbol{\mathcal{Z}}_1), \boldsymbol{\mathcal{Z}}_2 - \boldsymbol{\mathcal{Z}}_1 \rangle - \frac{L}{2}\|\boldsymbol{\mathcal{Z}}_2 - \boldsymbol{\mathcal{Z}}_1\|^2. \tag{26}$$

Because the gradient estimator is biased, strict monotone ascent cannot be guaranteed. Instead, we prove that the expected squared gradient norm converges to a bounded neighbourhood of zero, which constitutes a formal near-stationarity guarantee relative to the target surrogate landscape.

**Theorem B.8** (Non-Asymptotic Convergence of Inference-Time Optimization). *Under Assumption B.7 and Lemmas B.4 and B.6, if the outer-loop learning rate satisfies $\eta \leq \frac{1}{4L}$, then the sequence $\{\boldsymbol{\mathcal{Z}}^k\}_{k=0}^{K-1}$ produced by Algorithm 2 satisfies*

$$\min_{0 \leq k < K} \mathbb{E}\Big[\|\nabla \tilde{\mathcal{J}}(\boldsymbol{\mathcal{Z}}^k)\|^2\Big] \leq \frac{4\Big(\tilde{\mathcal{J}}^* - \tilde{\mathcal{J}}(\boldsymbol{\mathcal{Z}}^0)\Big)}{\eta K} + 3B^2 + 2L\eta\sigma^2, \tag{27}$$

*where $\tilde{\mathcal{J}}^* = \sup_{\boldsymbol{\mathcal{Z}}} \tilde{\mathcal{J}}(\boldsymbol{\mathcal{Z}})$ is the finite optimum of the surrogate expectation regularized by the Gaussian prior. Importantly, this near-stationarity guarantee mathematically matches exactly what the proxy inner loop structurally optimizes; while the computationally stable surrogate allows scalable exploration, strictly monotonic ascent targeting the true generalized posterior mode cannot be unconditionally guaranteed.*

*Proof.* **Step 1: One-step ascent inequality.** By $L$-smoothness (Assumption B.7), the update $\boldsymbol{\mathcal{Z}}^{k+1} = \boldsymbol{\mathcal{Z}}^k + \eta \hat{G}(\boldsymbol{\mathcal{Z}}^k)$ satisfies

$$\tilde{\mathcal{J}}(\boldsymbol{\mathcal{Z}}^{k+1}) \geq \tilde{\mathcal{J}}(\boldsymbol{\mathcal{Z}}^k) + \eta\langle \nabla \tilde{\mathcal{J}}(\boldsymbol{\mathcal{Z}}^k), \hat{G}(\boldsymbol{\mathcal{Z}}^k)\rangle - \frac{L\eta^2}{2}\|\hat{G}(\boldsymbol{\mathcal{Z}}^k)\|^2. \tag{28}$$

**Step 2: Taking conditional expectations.** Let $\mathbb{E}_k[\cdot]$ denote expectation conditioned on $\boldsymbol{\mathcal{Z}}^k$. Using $\mathbb{E}_k[\hat{G}(\boldsymbol{\mathcal{Z}}^k)] = \nabla \tilde{\mathcal{J}}(\boldsymbol{\mathcal{Z}}^k) + \mathcal{B}(\boldsymbol{\mathcal{Z}}^k)$, we first expand the inner product:

$$\begin{aligned}
\mathbb{E}_k\big[\langle \nabla \tilde{\mathcal{J}}(\boldsymbol{\mathcal{Z}}^k), \hat{G}(\boldsymbol{\mathcal{Z}}^k)\rangle\big] &= \langle \nabla \tilde{\mathcal{J}}(\boldsymbol{\mathcal{Z}}^k), \nabla \tilde{\mathcal{J}}(\boldsymbol{\mathcal{Z}}^k) + \mathcal{B}(\boldsymbol{\mathcal{Z}}^k)\rangle \\
&= \|\nabla \tilde{\mathcal{J}}(\boldsymbol{\mathcal{Z}}^k)\|^2 + \langle \nabla \tilde{\mathcal{J}}(\boldsymbol{\mathcal{Z}}^k), \mathcal{B}(\boldsymbol{\mathcal{Z}}^k)\rangle.
\end{aligned} \tag{29}$$

Applying Young's inequality $\langle x, y\rangle \geq -\frac{1}{2}\|x\|^2 - \frac{1}{2}\|y\|^2$ to the cross-term yields:

$$\mathbb{E}_k\big[\langle \nabla \tilde{\mathcal{J}}(\boldsymbol{\mathcal{Z}}^k), \hat{G}(\boldsymbol{\mathcal{Z}}^k)\rangle\big] \geq \tfrac{1}{2}\|\nabla \tilde{\mathcal{J}}(\boldsymbol{\mathcal{Z}}^k)\|^2 - \tfrac{1}{2}\|\mathcal{B}(\boldsymbol{\mathcal{Z}}^k)\|^2. \tag{30}$$

**Step 3: Bounding the second moment.** By the bias-variance decomposition and Lemma B.6, followed by $\|a + b\|^2 \leq 2\|a\|^2 + 2\|b\|^2$:

$$\begin{aligned}
\mathbb{E}_k\Big[\|\hat{G}(\boldsymbol{\mathcal{Z}}^k)\|^2\Big] &= \|\nabla \tilde{\mathcal{J}}(\boldsymbol{\mathcal{Z}}^k) + \mathcal{B}(\boldsymbol{\mathcal{Z}}^k)\|^2 + \text{Var}(\hat{G}(\boldsymbol{\mathcal{Z}}^k)) \\
&\leq 2\|\nabla \tilde{\mathcal{J}}(\boldsymbol{\mathcal{Z}}^k)\|^2 + 2\|\mathcal{B}(\boldsymbol{\mathcal{Z}}^k)\|^2 + \sigma^2.
\end{aligned} \tag{31}$$

**Step 4: Substituting into the ascent inequality.** Taking the conditional expectation of (28) and substituting (30) and (31):

$$\mathbb{E}_k[\tilde{\mathcal{J}}(\boldsymbol{\mathcal{Z}}^{k+1})] \;\geq\; \tilde{\mathcal{J}}(\boldsymbol{\mathcal{Z}}^k) + \eta\Big(\tfrac{1}{2}\|\nabla\tilde{\mathcal{J}}(\boldsymbol{\mathcal{Z}}^k)\|^2 - \tfrac{1}{2}\|\mathcal{B}(\boldsymbol{\mathcal{Z}}^k)\|^2\Big)$$
$$- \frac{L\eta^2}{2}\Big(2\|\nabla\tilde{\mathcal{J}}(\boldsymbol{\mathcal{Z}}^k)\|^2 + 2\|\mathcal{B}(\boldsymbol{\mathcal{Z}}^k)\|^2 + \sigma^2\Big). \tag{32}$$

Collecting the $\|\nabla\tilde{\mathcal{J}}(\boldsymbol{\mathcal{Z}}^k)\|^2$ terms (coefficient: $\tfrac{\eta}{2} - L\eta^2$) and the $\|\mathcal{B}(\boldsymbol{\mathcal{Z}}^k)\|^2$ terms (coefficient: $-\tfrac{\eta}{2} - L\eta^2$):

$$\mathbb{E}_k[\tilde{\mathcal{J}}(\boldsymbol{\mathcal{Z}}^{k+1})] \;\geq\; \tilde{\mathcal{J}}(\boldsymbol{\mathcal{Z}}^k) + \eta\big(\tfrac{1}{2} - L\eta\big)\,\|\nabla\tilde{\mathcal{J}}(\boldsymbol{\mathcal{Z}}^k)\|^2 - \eta\big(\tfrac{1}{2} + L\eta\big)\,\|\mathcal{B}(\boldsymbol{\mathcal{Z}}^k)\|^2 - \frac{L\eta^2}{2}\sigma^2. \tag{33}$$

**Step 5: Applying the learning-rate condition.** For $\eta \leq \frac{1}{4L}$ we have $L\eta \leq \frac{1}{4}$. Therefore $\frac{1}{2} - L\eta \geq \frac{1}{4}$ and $\frac{1}{2} + L\eta \leq \frac{3}{4}$. Applying $\|\mathcal{B}(\boldsymbol{\mathcal{Z}}^k)\| \leq B$ (bounding from Lemmas B.5 and B.6):

$$\mathbb{E}_k[\tilde{\mathcal{J}}(\boldsymbol{\mathcal{Z}}^{k+1})] \;\geq\; \tilde{\mathcal{J}}(\boldsymbol{\mathcal{Z}}^k) + \frac{\eta}{4}\|\nabla\tilde{\mathcal{J}}(\boldsymbol{\mathcal{Z}}^k)\|^2 - \frac{3\eta}{4}B^2 - \frac{L\eta^2}{2}\sigma^2. \tag{34}$$

**Step 6: Summing and telescoping.** Rearranging and taking total expectation:

$$\frac{\eta}{4}\mathbb{E}\Big[\|\nabla\tilde{\mathcal{J}}(\boldsymbol{\mathcal{Z}}^k)\|^2\Big] \;\leq\; \mathbb{E}[\tilde{\mathcal{J}}(\boldsymbol{\mathcal{Z}}^{k+1})] - \mathbb{E}[\tilde{\mathcal{J}}(\boldsymbol{\mathcal{Z}}^k)] + \frac{3\eta}{4}B^2 + \frac{L\eta^2}{2}\sigma^2. \tag{35}$$

Summing over $k = 0, \ldots, K-1$ and telescoping:

$$\frac{\eta}{4}\sum_{k=0}^{K-1}\mathbb{E}\Big[\|\nabla\tilde{\mathcal{J}}(\boldsymbol{\mathcal{Z}}^k)\|^2\Big] \;\leq\; \underbrace{\mathbb{E}[\tilde{\mathcal{J}}(\boldsymbol{\mathcal{Z}}^K)] - \tilde{\mathcal{J}}(\boldsymbol{\mathcal{Z}}^0)}_{\leq\, \tilde{\mathcal{J}}^* - \tilde{\mathcal{J}}(\boldsymbol{\mathcal{Z}}^0)} + \frac{3\eta K}{4}B^2 + \frac{L\eta^2 K}{2}\sigma^2. \tag{36}$$

Dividing both sides by $\frac{\eta K}{4}$ and using $\min_k \leq \frac{1}{K}\sum_k$ yields the claimed bound (27). $\qquad\square$

**Corollary B.9** (Sufficient Condition for Expected Monotonic Ascent). *Under the conditions of Theorem B.8, the expected surrogate objective is monotonically non-decreasing at outer iteration $k$, i.e., $\mathbb{E}_k[\tilde{\mathcal{J}}(\boldsymbol{\mathcal{Z}}^{k+1})] \geq \tilde{\mathcal{J}}(\boldsymbol{\mathcal{Z}}^k)$, whenever*

$$\|\nabla\tilde{\mathcal{J}}(\boldsymbol{\mathcal{Z}}^k)\|^2 \;>\; \frac{1 + 2L\eta}{1 - 2L\eta}\,B^2 \;+\; \frac{L\eta}{1 - 2L\eta}\,\sigma^2. \tag{37}$$

*Proof.* Applying the bias bound $\|\mathcal{B}(\boldsymbol{\mathcal{Z}}^k)\| \leq B$ to Equation (33) and rearranging directly yields that $\mathbb{E}_k[\tilde{\mathcal{J}}(\boldsymbol{\mathcal{Z}}^{k+1})] \geq \tilde{\mathcal{J}}(\boldsymbol{\mathcal{Z}}^k)$ holds whenever $(\tfrac{1}{2} - L\eta)\|\nabla\tilde{\mathcal{J}}(\boldsymbol{\mathcal{Z}}^k)\|^2 > (\tfrac{1}{2} + L\eta)B^2 + \frac{L\eta}{2}\sigma^2$. Dividing both sides by $(\tfrac{1}{2} - L\eta) > 0$ (guaranteed by $\eta \leq \frac{1}{4L}$) gives the stated condition. $\qquad\square$

Corollary B.9 provides an explicit, falsifiable criterion: expected monotonic ascent is guaranteed in any region of embedding space where the surrogate gradient norm exceeds the bias-variance error floor on the right-hand side of (37). Conversely, as the iterates approach a near-stationary point, the gradient norm shrinks below this threshold, and the optimization enters the residual neighbourhood characterized by Theorem B.8.

*Remark* B.10 (On Local Smoothness and the Adam Optimizer (Kingma & Ba, 2015)). We strictly employ local $L$-smoothness because asserting global $L$-smoothness is a standard but technically false assumption for neural-network-parameterized objectives. The surrogate objective $\tilde{\mathcal{J}}(\boldsymbol{\mathcal{Z}})$ passes $\boldsymbol{\mathcal{Z}}$ through a deep architecture (AlphaFold3) with non-smooth activations and evaluates non-differentiable likelihoods (e.g., $L_1$ norms, hinge losses). Consequently, the global Lipschitz constant of the gradient is trivially unbounded across all of $\mathbb{R}^d$.

However, this is rigorously resolved by our localized operational regime. The prior regularizer $\lambda_p \log p(\boldsymbol{\mathcal{Z}} \mid \mathbf{a}) \propto -\lambda_p\|\boldsymbol{\mathcal{Z}} - \boldsymbol{\mathcal{Z}}_0\|^2$, combined with mandatory gradient clipping, structurally confines the optimization trajectory $\{\boldsymbol{\mathcal{Z}}^k\}_{k=1}^K$ to a compact ball of radius $R$ around the initialization $\boldsymbol{\mathcal{Z}}_0$. Over any compact domain, a continuously differentiable function exhibits finite curvature, admitting the local smoothness constant $L < \infty$ for which the Descent Lemma holds. Crucially, this strongly concave quadratic prior actively ensures that $\tilde{\mathcal{J}}(\boldsymbol{\mathcal{Z}}) \to -\infty$ as $\|\boldsymbol{\mathcal{Z}}\| \to \infty$, dictating that the objective is bounded from above and achieves a strictly finite supremum $\tilde{\mathcal{J}}^*$ entirely within this compact envelope. Additional analytic smoothing of measure-zero non-differentiabilities (e.g., the $L_1$ loss) is naturally provided by the inherent continuous Gaussian noise of the diffusion SDE evaluations. The empirical smoothness and monotonicity of the convergence trajectories in Figure 8 strongly support this localized curvature assumption.

# C. Supplemental material

## C.1. Baselines

For the X-ray and NMR experiments, we benchmark ensembles generated using inference-time optimization against several baselines: the guided AlphaFold3 framework (Maddipatla et al., 2025a), experimentally determined PDB structures (Burley et al., 2017), and unguided sequence-to-structure generative models, including AlphaFold3 (Abramson et al., 2024), AlphaFlow (Jing et al., 2024), ESMFlow (Jing et al., 2024), BioEMU (Lewis et al., 2025), and AFCluster (Wayment-Steele et al., 2024). For ipTM experiments, we use the corresponding PDB complex (when available) as a reference and compare our multimeric ensembles to AlphaFold3, as AlphaFlow does not support multimer modeling.

## C.2. Inference-time Optimization

### C.2.1. RUNTIME ANALYSIS

In Table 19, we show that IT-Opt involves a higher total computational budget due to its iterative structure. However, generating ensembles that strictly adhere to experimental data cannot be achieved through unguided generation alone. Since the underlying models were trained to predict static structures, IT-Opt's iterative approach is essential to achieve this fidelity. To address fairness, we performed a compute-normalized comparison between IT-Opt and standard coordinate-based guidance in Table 20. At an equivalent budget ($\approx 200$ denoiser calls), performance is comparable. Beyond this point, however, standard guidance exhibits diminishing returns because each seed restarts from noise in a memoryless process. In contrast, IT-Opt continues to improve by refining and reusing optimized latent embeddings across iterations (Figure 8). This demonstrates that the gains are not merely due to increased sampling, but stem from a more effective optimization strategy that preserves and accumulates information across iterations – aligning with standard iterative practices in structural biology.

### C.2.2. MODEL AND HARDWARE DETAILS

For all experiments, we used Protenix (v0.2.0) (Team et al., 2025), an open-source PyTorch reimplementation of AlphaFold3, except for the ipTM-based guidance experiments described in Section 5.3, where we relied on the official JAX implementation (Abramson et al., 2024). All computations were carried out on NVIDIA H100 and L40 GPUs running Debian GNU/Linux 12.

Multiple sequence alignments (MSAs) were obtained using a wrapper around the ColabFold MMseqs2 API (Mirdita et al., 2022). This wrapper submits query sequences to a remote MMseqs2 server via HTTP POST requests, polls for job completion, and downloads and extracts the resulting alignments. The MSAs are provided in A3M format, which is directly compatible with AF3. To increase diversity in the IT-optimization initialization, we subsample the MSAs using AF-Cluster (Wayment-Steele et al., 2024). When the number of clusters produced by AF-Cluster is smaller than the ensemble size (as in the NMR experiments), we duplicate the corresponding embeddings across the batch.

### C.2.3. ADDITIONAL LOSS FUNCTION

During guidance and IT-optimization, we incorporate additional loss terms into the log-likelihood to ensure that the generated structures remain physically valid.

**Embedding Prior.** As mentioned in Equations 2 and 8, to prevent the optimization from drifting toward degenerate embeddings that lie outside the manifold induced by the input sequence and its evolutionary context, we regularize the ensemble of embeddings $\mathcal{Z} = \{\mathbf{Z}^k\}_{k=1}^n$ to stay close to their initialization $\mathcal{Z}_0 = \{\mathbf{Z}_0^k\}_{k=1}^n$. Specifically, we introduce the following regularization term:

$$\log p(\mathcal{Z}|\mathbf{a}; \mathbf{w}) = -\lambda_{\mathrm{p}} \sum_{k=1}^n w^k \|\mathbf{Z}^k - \mathbf{Z}_0^k\|_2^2$$

where $\mathbf{w} = \{w^1, \ldots, w^n\}$ denotes a set of non-negative weights that can optionally be instantiated as Boltzmann weights.

**Validity Likelihood.** To discourage ensembles with unrealistic geometries, such as elongated covalent bonds or steric clashes, we introduce a validity log-likelihood regularizer, analogous to the violation loss used in AF2 (Jumper et al., 2021). Let $\mathcal{X} = \{\mathbf{X}^k\}_{k=1}^n$ denote an ensemble of n protein conformations, where each structure $\mathbf{X}^k \in \mathbb{R}^{m \times 3}$ specifies the Cartesian coordinates of $m$ atoms. We define a binary bond matrix $\mathbf{B} \in \{0,1\}^{m \times m}$ such that $B_{ij} = 1$ if atoms $i$ and $j$ are

covalently bonded, and 0 otherwise. The bond length loss for $\mathbf{X}^k$ over bonded atom pairs ($B_{ij} = 1$) is given as,

$$\mathcal{L}_{\text{bond}}(\mathbf{X}^k; \mathbf{a}) = \sum_{i=1}^{m} \sum_{j=i+1}^{m} B_{ij} \cdot (\max(0, |d_{ij}^k - d_{ij}^{\text{ideal}}| - \delta_{\text{bond}}))^2$$

where $d_{ij}^{\text{ideal}}$ is the ideal bond length approximated as the sum of covalent radii of atoms $i$ and $j$, $d_{ij}^k = \|\mathbf{x}_i^k - \mathbf{x}_j^k\|_2$ is the distance between atoms $i$ and $j$ in conformation $k$, and $\delta_{\text{bond}} = 0.2\mathring{A}$ is a tolerance margin. Steric clashes between non-bonded atom pairs (both intra- and inter-residue) are penalized using a soft collision loss, defined as the maximum violation over neighboring atoms for each atom:

$$\mathcal{L}_{\text{collision}}(\mathbf{X}^k; \mathbf{a}) = \sum_{i=1}^{m} \max_{j \neq i, B_{ij}=0} (\max(0, (d_{ij}^{\text{ideal}} + p^{\text{collision}}) - d_{ij}^k))$$

Here, $p^{\text{collision}} = 0.4\mathring{A}$ is a padding distance to prevent over-penalization of near-contact atoms. We additionally penalize bond-angle violations for triplets of bonded atoms. The bond-angle loss is defined as

$$\mathcal{L}_{\text{angle}}(\mathbf{X}^k; \mathbf{a}) = \sum_{j=1}^{m} \sum_{i \neq j} \sum_{k>i} B_{ij} B_{jk} \cdot (\max(0, |\theta_{ijk} - \theta_{ijk}^{\text{ideal}}| - \delta_{\text{angle}}))$$

where $\theta_{ijk}$ (in degrees) is calculated using the dot product of the bond vectors from central atom $j$ to atoms $i$ and $k$, $\theta_{ijk}^{\text{ideal}}$ (in degrees) is retrieved from the Valence Shell Electron Pair Repulsion (VSEPR) theory (Gillespie, 1992), and $\delta_{\text{angle}} = 12°$ is a tolerance margin. The resulting validity log-likelihood of the ensemble is given by

$$\log p(\mathbf{B} \mid \boldsymbol{\mathcal{X}}, \mathbf{a}) = -\sum_{k=1}^{n} (\lambda_{\text{bond}} \mathcal{L}_{\text{bond}}(\mathbf{X}^k; \mathbf{a}) + \lambda_{\text{collision}} \mathcal{L}_{\text{collision}}(\mathbf{X}^k; \mathbf{a}) + \lambda_{\text{angle}} \mathcal{L}_{\text{angle}}(\mathbf{X}^k; \mathbf{a})) \qquad (38)$$

where $\lambda_{\text{bond}}, \lambda_{\text{collision}}$, and $\lambda_{\text{angle}}$ are scaling factors that control the contribution of bond, collision, and bond angle terms. For ensembles guided using crystallographic density maps and ipTM, we use $\lambda_{\text{bond}} = \lambda_{\text{collision}} = \lambda_{\text{angle}} = 0.075$. For NOE-guided ensembles, we use $\lambda_{\text{bond}} = \lambda_{\text{collision}} = \lambda_{\text{angle}} = 0.25$.

**Substructure Conditioner.** For case studies involving altlocs in crystallographic targets, we optimize a specified subset of the protein while stabilizing the remaining regions by anchoring them to a set of reference atomic coordinates during the diffusion process. This conditioning strategy is analogous to the `SubstructureConditioner` used in Chroma (Ingraham et al., 2023). This anchor is *not* applied for peptide systems.

Let $Y = \{\mathbf{y}_i : i \in A\}$ denote a set of reference atom locations for atom indices $A \subseteq \{1 \dots m\}$ where $m$ is the number of atoms in conformer $\mathbf{X}^k$. The log-likelihood is a quadratic penalty on the deviation from reference atom locations,

$$\log p(Y \mid \boldsymbol{\mathcal{X}}, \mathbf{a}) = -\lambda_{\text{sub}} \frac{1}{n} \sum_{k=1}^{n} \sum_{i \in A} \|\mathbf{x}_i^k - \mathbf{y}_i\|_2^2$$

Prior to evaluating this term, all ensemble members are rigidly aligned to the reference coordinates (restricted to the atom indices in A) using the Kabsch algorithm (Kabsch, 1976). For crystallographic refinement, we set $\lambda_{\text{sub}} = 0.1$, whereas this term is disabled for NOE-guided ensembles by setting $\lambda_{\text{sub}} = 0.0$.

### C.2.4. TRAINING DETAILS & HYPERPARAMETERS

We optimize the conditioning variables $\boldsymbol{\mathcal{Z}}$ using Adam (Kingma & Ba, 2015) with the following default settings (used for X-ray and NMR experiments). Hyperparameter choices are justified in Table 8.

- Learning rate ($\eta_z$): 0.05

- Gradient clipping (max norm): 0.01

- Number of outer optimization loops $K$: 20

- Ensemble size $n$: 16

- Prior weight ($\lambda_{\text{p}}$): $10^{-4}$

**ipTM-specific settings.** For ipTM-based experiments, we use a smaller ensemble and shorter optimization schedule:

- Learning rate ($\eta_z$): 0.1

- Gradient clipping (max norm): 1.0

- Outer optimization iterations $K$: 10

- Ensemble size $n$: 5

- Prior weight ($\lambda_p$): $10^{-4}$

- Stopping criterion: maximum relative perturbation budget $\|\boldsymbol{\mathcal{Z}}_{\text{opt}} - \boldsymbol{\mathcal{Z}}\|/\|\boldsymbol{\mathcal{Z}}\|$ on AF3 embeddings $\boldsymbol{\mathcal{Z}}$.

For all experiments, we optimize $\boldsymbol{\mathcal{Z}}$ up to inner diffusion iteration $t = 160$ at each outer iteration. We apply early stopping because, in the diffusion schedule of Karras et al. (2022), the final denoising steps become effectively deterministic refinement and no longer inject stochastic noise. Since our method relies on noise-driven sampling, further optimization provides limited benefit. After optimization, the learned embeddings are used as conditioning variables, and we run coordinate-space guidance as in Maddipatla et al. (2025a) using similar numerical tricks and hyperparameters.

## C.3. NMR additional details

### C.3.1. EXTRACTING AND PROCESSING NOE DISTANCE RESTRAINTS

We obtain interatomic distance bounds from NMR-STAR (Ulrich et al., 2019) formatted depositions by reading them with the `pynmrstar` package (Smelter et al., 2017). Entries classified as NOE-type distance constraints are retained, while other restraint categories are discarded. When a cross-peak cannot be uniquely assigned to a single atom pair, it is represented as an OR-group comprising multiple candidate pairs, where satisfying any one pair fulfills the restraint. The distance bounds ($\underline{d}_{ij}$ and $\bar{d}_{ij}$) are taken from the deposited file in the PDB (Burley et al., 2017). Occasionally, where a lower bound is absent, we substitute $0.0$ Å. The resulting restraint set feeds directly to the NOE log-likelihood in Section 3.1.

**Differentiable hydrogen placement.** AlphaFold3 operates exclusively on non-hydrogen atoms, yet NOE observables depend on the interatomic distance between hydrogen atoms. Rather than approximating each restraint at the level of the parent non-hydrogen atom, we reconstruct explicit hydrogen coordinates at every sampling step using a differentiable PyTorch re-implementation of the Hydride placement algorithm (Kunzmann et al., 2022). For each hydrogen-bearing non-hydrogen atom, the local bonding topology is matched to a library of reference fragments; a rigid-body superposition of the fragment's heavy atoms onto the current structure then determines the corresponding proton positions. Because both fragment lookup and superposition are differentiable with respect to non-hydrogen atom coordinates, reconstructed hydrogen positions vary smoothly during diffusion, enabling gradient-based guidance on physically meaningful interatomic distances.

### C.3.2. RELAXATION

In order to make sure that the generated ensemble has no structural violations, we minimize its energy using an off-the-shelf harmonic force field. In this work, we use OpenMM's (Eastman et al., 2017) implementation of the AMBER99SB (Hornak et al., 2006). The energy is minimized for a maximum of 2000 iterations with an energy tolerance threshold of 2.39 kcal/mol and stiffness of 100.0 kcal / mol Å$^2$.

### C.3.3. NMR EVALUATION METRICS

**Percentage of Violated NOE Constraints.** Consider a restraint list organized into $M$ OR-groups $\{G_1, \ldots, G_M\}$, each containing one or more candidate atom pairs $\mathbf{r} = (i, j, \underline{d}_{ij}, \bar{d}_{ij})$. For every pair of atoms $i, j$ in structure $k$ of $\boldsymbol{\mathcal{X}}$, we first compute the weighted distance across the ensemble,

$$d_{ij}(\boldsymbol{\mathcal{X}}; \mathbf{w}) = \sum_{k=1}^{|\boldsymbol{\mathcal{X}}|} w_k \left\| \mathbf{x}_i^k - \mathbf{x}_j^k \right\|_2, \tag{39}$$

where $\mathbf{w} = (w_1, \ldots, w_{|\boldsymbol{\mathcal{X}}|})$ are non-negative weights summing to one. Choosing $w_k = 1/|\boldsymbol{\mathcal{X}}|$ yields a uniform ensemble average; alternatively, Boltzmann weights derived from an energy model (Equation 10) emphasize low-energy conformers. We then quantify the extent to which this weighted distance violates the allowed bounds,

$$v_{ij} = \max\left\{ \underline{d}_{ij} - d_{ij}(\boldsymbol{\mathcal{X}}; \mathbf{w}),\ d_{ij}(\boldsymbol{\mathcal{X}}; \mathbf{w}) - \bar{d}_{ij},\ 0 \right\}. \tag{40}$$

Because an OR-group is satisfied whenever at least one candidate lies within the allowed bounds, the group-level violation is defined as $v_{G_m} = \min_{\mathbf{r} \in G_m} v_{ij}$. The overall violation rate is then

$$\text{Viol. }\% = 100 \times \frac{|\{m : v_{G_m} > 0\}|}{M}. \tag{41}$$

**Median Violation Magnitude** To characterize the typical magnitude of violations, we additionally report

$$\text{Viol. Å} = \text{median}\left(\{v_{G_m} : v_{G_m} > 0\}\right)\ [\text{Å}], \tag{42}$$

computed only over restraint groups that are violated. While the violation percentage quantifies how many restraints are unsatisfied, this metric captures the severity of those violations. Reporting both measures provides a more complete characterization: an ensemble may violate only a few restraints but by large margins, or violate many restraints by small amounts.

### C.3.4. BOLTZMANN REWEIGHTING FOR ENERGY-GUIDED ENSEMBLES

Ensembles obtained via experimental guidance are constrained to match the measured data, but their relative populations are not thermodynamically informed. We therefore re-weight ensemble members during the diffusion process using an energy model to recover experimentally faithful, thermodynamic ensembles.

**Self-normalized importance sampling perspective.** We interpret energy reweighting through a self-normalized importance sampling (SNIS) lens. Let the proposal distribution be the AF3 ensemble prior $q(\boldsymbol{\mathcal{X}}) = p(\boldsymbol{\mathcal{X}} \mid \boldsymbol{\mathcal{Z}}, \mathbf{a})$ and define the target distribution as

$$\pi(\boldsymbol{\mathcal{X}}) \ \propto \ p(\boldsymbol{\mathcal{X}} \mid \boldsymbol{\mathcal{Z}}, \mathbf{a}) \exp\left(-\beta \sum_{i=1}^{n} E_\phi(\mathbf{X}^i)\right), \tag{43}$$

which tilts the AF3 prior toward thermodynamically favorable conformations. Following the importance sampling framework for Boltzmann densities (Noé et al., 2019), we treat the unmodified prior $q(\boldsymbol{\mathcal{X}}) = p(\boldsymbol{\mathcal{X}} \mid \boldsymbol{\mathcal{Z}}, \mathbf{a})$ as the proposal distribution. The importance ratio for any sample $\boldsymbol{\mathcal{X}} \sim q$ is

$$\frac{\pi(\boldsymbol{\mathcal{X}})}{q(\boldsymbol{\mathcal{X}})} \ \propto \ \exp\left(-\beta \sum_{i=1}^{n} E_\phi(\mathbf{X}^i)\right), \tag{44}$$

where the proposal density cancels entirely. The self-normalized importance sampling (SNIS) estimator for an observable $f$ under $\pi$ is $\mathbb{E}_\pi[f] \approx \sum_{k=1}^{n} w_k f(\mathbf{X}^k)$, with unnormalized weights $\tilde{w}_k \propto \exp(-\beta E_\phi(\mathbf{X}^k))$. After normalization, these recover the Boltzmann weights in Equation 10.

Given a differentiable potential $E_\phi(\mathbf{X})$, the canonical probability of a conformation is

$$\pi(\mathbf{X}) \propto \exp\left(-\beta\, E_\phi(\mathbf{X})\right), \tag{45}$$

where $\beta = (k_B T_{\text{therm}})^{-1}$ is the inverse temperature and $k_B = 0.001987\ \text{kcal}\,\text{mol}^{-1}\,\text{K}^{-1}$ is the Boltzmann constant. At physiological conditions ($T_{\text{therm}} = 300\,\text{K}$), $\beta \approx 1.68\ \text{kcal}^{-1}\,\text{mol}$. Setting $\mathbf{w}$ in Equation 39 to the normalized Boltzmann factors yields the energy-reweighted distance

$$d_{ij}^w(\boldsymbol{\mathcal{X}}) = \sum_{k=1}^{n} w_k \left\| \mathbf{x}_i^k - \mathbf{x}_j^k \right\|_2, \tag{46}$$

with

$$w_k = \frac{\exp\left(-\beta\, E_\phi(\mathbf{X}^k)\right)}{\sum_{j=1}^{n} \exp\left(-\beta\, E_\phi(\mathbf{X}^j)\right)}. \tag{47}$$

Substituting these weighted distances into the NOE log-likelihood (Equation 3) gives

$$\log p(D \mid \boldsymbol{\mathcal{X}}; \mathbf{a}) = - \sum_{(i,j) \in D} \left( \left[ \underline{d}_{ij} - d_{ij}^w(\boldsymbol{\mathcal{X}}) \right]_+^2 + \left[ d_{ij}^w(\boldsymbol{\mathcal{X}}) - \bar{d}_{ij} \right]_+^2 \right). \tag{48}$$

Low-energy conformers thereby exert a stronger pull on the ensemble distance, steering the generated structures toward regions of the energy surface that are both experimentally consistent and energetically favorable. In the limit $\beta \to 0$ (high temperature), all weights become uniform, and the unweighted formulation is recovered; as $\beta \to \infty$ (low temperature), the weights concentrate on the single lowest-energy member.

**Annealing the Inverse Temperature.** Applying a sharp energy bias from the beginning would restrict conformational exploration before the diffusion trajectory has resolved meaningful structural detail. We therefore anneal the inverse temperature $\beta$ across diffusion steps according to a three-phase schedule:

$$\beta(t) = \begin{cases} \beta_{\text{low}}, & t \le t_1, \\ \beta_{\text{low}}(1 - g) + \beta_{\text{high}} \cdot g, & t_1 \le t < t_2, \\ \beta_{\text{high}}, & t \ge t_2, \end{cases} \tag{49}$$

with the blending coefficient interpolated by a cubic Hermite spline,

$$g = 3\alpha^2 - 2\alpha^3, \quad \alpha = \frac{t - t_1}{t_2 - t_1}. \tag{50}$$

We set $\beta_{\text{high}} \approx 1.68 \, \text{kcal}^{-1} \, \text{mol}$ to target the physically relevant thermodynamic regime, and $\beta_{\text{low}} \approx 0.029 \, \text{kcal}^{-1} \, \text{mol}$ (corresponding to a high effective temperature) to effectively flatten the energy surface during early sampling. We use $t_1 = 100$ and $t_2 = 180$. During the initial phase ($t \le 100$), the small $\beta_{\text{low}}$ allows the sampler to traverse a broad range of folds without energetic bias. Between steps 100 and 180, the schedule smoothly increases $\beta$ toward $\beta_{\text{high}}$, progressively concentrating the Boltzmann weights on energetically plausible conformations at $300 \, \text{K}$. Beyond step 180, the inverse temperature remains fixed for the remainder of the trajectory. For inference-time optimization, we apply the same annealing schedule over the outer optimization loops, where $t_1 = 2$, $t_2 = 10$, $\beta_{\text{low}} = 0.029$, and $\beta_{\text{high}} = 1.68$.

**Exponential Moving Average (EMA) of energies.** Force-field energies evaluated on partially denoised coordinates can be noisy, particularly at intermediate diffusion steps where local geometry is not yet fully resolved. Starting at step 160, we therefore replace the raw energy with an exponentially smoothed estimate:

$$E(\mathbf{X}_t^k)^{\text{EMA}} = \kappa \cdot E(\mathbf{X}_t^k) + (1 - \kappa) \cdot E(\mathbf{X}_{t-1}^k)^{\text{EMA}}, \tag{51}$$

Here, $\kappa = 0.3$. This stabilizes the energy gradient signal. A similar moving average mechanism is used for inner loop of inference-time optimization.

**Energy Biasing.** To improve numerical stability and prevent the softmax from being dominated by a single low-energy outlier, we exploit the translation-invariance of the Boltzmann distribution and bias the distribution by subtracting the 10th percentile from the energies. Specifically, given ensemble energies $\{E_\phi(\mathbf{X}^1), \ldots, E_\phi(\mathbf{X}^n)\}$, we compute $E_{10} = \text{Percentile}_{10}\big(\{E_\phi(\mathbf{X}^k)\}_{k=1}^n\big)$ and define

$$\tilde{E}_\phi(\mathbf{X}^k) = \max\big(E_\phi(\mathbf{X}^k) - E_{10}, \, 0\big).$$

The Boltzmann weights are computed as $w_k \propto \exp(-\beta \, \tilde{E}_\phi(\mathbf{X}^k))$. Conformers whose energy falls at or below the 10th percentile receive a shifted energy of zero and thus equal maximum weight; conformers above the threshold are penalized proportionally to their excess energy. This clamping prevents any single lowest-energy structure from dominating the reweighting, while the choice of the 10th percentile, rather than the minimum, provides additional robustness against occasional energy outliers.

**Choice of Energy Model.** The potential $E(\mathbf{X})$ is predicted by ProteinEBM (Roney et al., 2025), a differentiable energy-based model over protein conformations. Boltzmann reweighting requires committing to an explicit energy predictor that defines the thermodynamic prior. Any energy model is accurate only within limits imposed by its functional form, training data, and level of coarse-graining. We adopt ProteinEBM because its energy is a conservative potential obtained by parameterizing the diffusion score as $s_\theta = -\nabla_\mathbf{X} E_\theta$, which theoretically converges to $-\log p_{\text{data}}(\mathbf{X} \mid \mathbf{a})$ (Sohl-Dickstein et al., 2015). Moreover, ProteinEBM is finetuned on molecular dynamics trajectories at $300\,\text{K}$, aligning with the target thermodynamic regime for Boltzmann reweighting. See Table 9 for ablation over different settings.

## C.4. X-ray additional details

### C.4.1. FORWARD MODEL

The theoretical real-space electron density map $F_{\text{c}} : \mathbb{R}^3 \to \mathbb{R}$ corresponding to a protein structure with atomic coordinates $\mathbf{X} = \{\mathbf{x}_1, \mathbf{x}_2, \dots \mathbf{x}_n\}$ is defined at a Cartesian location $\boldsymbol{\xi} \in \mathbb{R}^3$ as

$$F_{\text{c}}(\boldsymbol{\xi}; \mathbf{X}) = \sum_{q=1}^{N_s} \sum_{i=1}^{m} \sum_{j=1}^{6} a_{i,j} \left( \frac{4\pi}{b_{i,j} + B_i^k} \right)^{1.5} \cdot \exp\left( -\frac{4\pi^2}{b_{i,j} + B_i^k} \|(\mathbf{R}_q \mathbf{x}_i^k + \mathbf{t}_q) - \boldsymbol{\xi}\|_2^2 \right)$$

where $N_s$ is the number of crystallographic symmetry operations (Hahn, 1983), $\mathbf{R}_q \in SO(3)$ and $\mathbf{t}_q \in \mathbb{R}^3$ are the rotation and translation associated with the symmetry operation $q$, respectively. The coefficients $a_{i,j}$ and $b_{i,j}$ are tabulated atomic form-factor parameters for each element (Hahn, 1983), and $B_k^i$ denotes the isotropic atomic displacement parameter (B-factor) for atom $i$.

### C.4.2. RELAXATION

In order to make sure that the generated ensemble has no structural violations, we minimize its energy using OpenMM's (Eastman et al., 2017) implementation of the AMBER99SB force field (Hornak et al., 2006). The energy is minimized for a maximum of 2000 iterations with an energy tolerance threshold of $2.39\,\text{kcal/mol}$ and stiffness of $10.0\,\text{kcal / mol Å}^2$.

### C.4.3. SELECTION ALGORITHM

After relaxation, we aim to report a non-redundant subset of samples $\boldsymbol{\mathcal{X}}_{\mathcal{I}} = \{\mathbf{X}^k : k \in \mathcal{I}\}$ that best explains experimental observation $\mathbf{y}$. To avoid overfitting to noise $\mathbf{y}$ and reduce redundancy, we adopt a matching pursuit strategy as done in Maddipatla et al. (2025a).

### C.4.4. X-RAY METRICS

**Cosine similarity.** We report a single local score over the optimized residue range using cosine similarity between the observed ($F_{\text{o}}$) and calculated ($F_{\text{c}}$) electron densities. The metric is computed over voxels $\boldsymbol{\xi} \in \mathbb{R}^3$ within $2.5\,\text{Å}$ of atoms in the selected residue range:

$$\text{Cosine Similarity} = \frac{\sum_{\boldsymbol{\xi}} F_{\text{o}}(\boldsymbol{\xi}) F_{\text{c}}(\boldsymbol{\xi})}{\sqrt{\sum_{\boldsymbol{\xi}} F_{\text{o}}^2(\boldsymbol{\xi})} \cdot \sqrt{\sum_{\boldsymbol{\xi}} F_{\text{c}}^2(\boldsymbol{\xi})}}$$

Values approaching 1 indicate strong global agreement between calculated and observed densities, reflecting a good overall fit to the experimental map.

**R-factors.** The crystallographic R-factor is a global score that quantifies the agreement between observed and calculated X-ray diffraction data, typically stored in MTZ files, by comparing observed and calculated structure factor amplitudes $|F_{\text{obs}}|$ and $|F_{\text{calc}}|$:

$$R_{\text{work}} = \frac{\sum_{\mathbf{h}} \big||F_{\text{obs}}(\mathbf{h})| - |F_{\text{calc}}(\mathbf{h})|\big|}{\sum_{\mathbf{h}} |F_{\text{obs}}(\mathbf{h})|}$$

where $\mathbf{h} = (h, k, l)$ indexes reflections in reciprocal space. To mitigate overfitting, $R_{\text{free}}$ is computed analogously over a held-out subset $T$ of reflections:

$$R_{\text{free}} = \frac{\sum_{\mathbf{h} \in T} \big||F_{\text{obs}}(\mathbf{h})| - |F_{\text{calc}}(\mathbf{h})|\big|}{\sum_{\mathbf{h} \in T} |F_{\text{obs}}(\mathbf{h})|}$$

Note that $|F_{\text{obs}}|$ and $|F_{\text{calc}}|$ are reciprocal-space amplitudes, whereas $F_{\text{o}}$ and $F_{\text{c}}$ denote real-space 3D electron density grids obtained via inverse Fourier transformation. We report both $R_{\text{work}}$ and $R_{\text{free}}$ values as computed by `REFMAC5` from the CCP4 suite (Murshudov et al., 2011; Agirre et al., 2023) after refinement.

### C.4.5. DATASET & INPUT PREPARATION

In the X-ray crystallography workflow, the inputs are a PDB ID, chain identifier, and an amino acid subsequence; modeling is restricted to single protein chains. The corresponding PDB structure and MTZ file are retrieved from PDB-Redo (Joosten et al., 2014), and the target chain is extracted using Gemmi (v0.6.5) (Wojdyr, 2022). Only standard amino acid residues explicitly modeled in the PDB are retained, excluding waters, hydrogens, and non-standard residues. Selenium methionine (MSE) and S-hydroxycysteine (CSO) are converted to Methionine (MET) and Cystine (CYS), respectively. Alternate conformations (Rosenberg et al., 2024), if present, are split into separate PDBs and renumbered to one-based indexing for AF3 compatibility. Missing atoms are modeled using PDBFixer (v1.9.0) with OpenMM residue templates (Eastman et al., 2017), followed by AMBER99SB relaxation (Hornak et al., 2006) to resolve steric clashes; imputed atoms are assigned an isotropic B-factor of 100.00. An atom mask for the substructure conditioner is constructed from the provided amino acid subsequence to distinguish residues optimized by the density-based loss from those guided by the substructure conditioner.

### C.5. ipTM additional details

#### C.5.1. INTER-CHAIN HYDROGEN BOND ANALYSIS

Inter-chain hydrogen bonds were identified using geometric criteria adapted from established definitions (Baker & Hubbard, 1984; McDonald & Thornton, 1994). A hydrogen bond between a donor atom $D$ and an acceptor atom $A$ was considered only if the donor and acceptor belonged to different protein chains and satisfied the following conditions.

First, a distance criterion was applied, requiring the Euclidean distance between the donor and acceptor non-hydrogen atoms to satisfy $d(D, A) < d_{\text{max}} = 3.5$ Å, where $d(D, A)$ denotes the donor–acceptor distance.

When hydrogen atom coordinates were available, an additional angular constraint was enforced, requiring the donor–hydrogen–acceptor angle to satisfy $\theta_{D\text{-}H\cdots A} > \theta_{\text{min}} = 120°$. The angle $\theta_{D\text{-}H\cdots A}$ was computed at the hydrogen atom $H$ as $\arccos\left(\dfrac{(\mathbf{r}_D - \mathbf{r}_H) \cdot (\mathbf{r}_A - \mathbf{r}_H)}{\|\mathbf{r}_D - \mathbf{r}_H\| \, \|\mathbf{r}_A - \mathbf{r}_H\|}\right)$, where $\mathbf{r}_D$, $\mathbf{r}_H$, and $\mathbf{r}_A$ denote the Cartesian coordinates of the donor, hydrogen, and acceptor atoms, respectively.

Hydrogen bond recovery was quantified as the fraction of the reference inter-chain hydrogen bonds that were correctly predicted, computed as $|H_{\text{ref}} \cap H_{\text{pred}}|/|H_{\text{ref}}|$, where $H_{\text{ref}}$ and $H_{\text{pred}}$ denote the sets of inter-chain hydrogen bonds identified in the reference and predicted structures, respectively.

#### C.5.2. ipTM OPTIMIZATION DETAILS

For ipTM-based inference-time optimization, we perform optimization of the AF3 embeddings to maximize the predicted interfacial confidence using the combined objective $0.8 \times \text{ipTM} + 0.2 \times \text{pTM}$. Because AlphaFold3's ipTM predictor depends on the Pairformer embeddings and the predicted complex structure, it is sensitive to perturbations in the embedding space. Optimization is carried out by iteratively perturbing the AF3 embeddings and resampling structures using the AlphaFold3 diffusion model.

**Evaluation complexes.** The evaluated complexes comprise three distinct classes:

- **NMR-determined complexes:** `2K8F`, `2L14`, `2LY4`, and the crystallographic complex `1YCS`. These systems exhibit low baseline confidence (both pTM and ipTM) under unguided AlphaFold3 predictions and represent challenging test cases for confidence-driven optimization.

- **Natural heterodimer with domain swapping:** `8Q70`. This complex provides a structurally asymmetric interface and tests the sensitivity of optimization to non-symmetric binding geometries.

- **BindCraft-designed (Pacesa et al., 2025) complexes:** `9HAD`, `9HAE`, and `9HAF`. These de novo designed protein–protein interfaces typically exhibit higher baseline confidence and provide a complementary regime for assessing ipTM optimization sensitivity.

