# OpenReview forum: "Inference-time optimization for experiment-grounded protein ensemble generation"
_ICML.cc/2026/Conference — ICML 2026 regular_

### Official Review · Reviewer_JDDy · 2026-03-01

**Soundness:** 4
**Presentation:** 2
**Significance:** 3
**Originality:** 1
**Overall Recommendation:** 5
**Confidence:** 4

**Summary:**

The paper introduces a novel inference-time optimization (IT-Opt) framework designed to generate protein structural ensembles that accurately reflect experimental data. Current sequence-conditioned diffusion models like AlphaFold3 often fail to capture alternate functional states or agree with structural measurements such as NOE distances or crystallographic density maps. Previous approaches attempted to solve this via coordinate-space guidance during the reverse diffusion trajectory, which is sensitive to initialization and constrained by the noise schedule. The authors propose shifting the optimization to the latent representation space. They iteratively update the Pairformer conditioning embeddings (Z) to maximize the ensemble log-likelihood of the experimental observation. To ensure the resulting ensembles are thermodynamically plausible, they incorporate a differentiable force field (ProteinEBM) to apply Boltzmann-weighted sampling via self-normalized importance sampling (SNIS). The method demonstrates improved constraint satisfaction on NMR datasets and better density map alignment on X-ray crystallography targets compared to unguided and coordinate-guided baselines. The authors conduct a perturbation analysis showing that the interfacial predicted TM-score (ipTM) can be artificially inflated by minute ($\approx 0.01\%$) updates to the MSA embedding space without corresponding improvements in structural accuracy.

**Compliance With Llm Reviewing Policy:**

Affirmed.

**Final Justification:**

I raised some questions and they are all addressed by the authors with additional explanations or experimental results.

**Key Questions For Authors:**

- Would the authors mind moving some important results from the supplements to the main text?
- What is the wall-clock time and memory consumption of the IT-Opt procedure compared to unguided AlphaFold3 and coordinate-guided baselines?
- In the ipTM perturbation experiments, you mention that $0.01\%$ relative changes to the embedding space drastically alter the confidence metrics. Can you characterize the nature of these embedding shifts? Do they correspond to specific evolutionary features or structural regions?
- How does the framework perform on larger protein-protein interfaces or multi-domain proteins where long-range allosteric effects must be modeled?
See the "Weaknesses" part.

**Limitations:**

The paper does not address the computational scalability of the nested optimization algorithm, nor does it discuss the reliance on the accuracy of the selected differentiable force field (ProteinEBM). Adding a clear discussion on computational bottlenecks and force-field dependencies would strengthen the submission. There are a lot of potential issues but I would be very happy to raise my rating if the authors successfully handle my raised issues.

**Strengths And Weaknesses:**

Strengths:
- Shifting the guidance objective from the coordinate space $X$ during the denoising trajectory to the latent Pairformer embedding space $\mathcal{Z}$ is a highly creative and effective approach. Treating AlphaFold3 as a learned structural prior and modifying the upstream conditioning variables elegantly sidesteps the finite-horizon limitations of standard diffusion guidance.
- The findings regarding the adversarial susceptibility of ipTM and related confidence metrics are highly impactful for the protein design community. Since many current binder-design workflows (e.g., BindCraft) heavily rely on these metrics to rank generated complexes, demonstrating that confidence can decouple from structural accuracy exposes a critical vulnerability.
- The integration of the experimental likelihood optimization with a physically meaningful thermodynamic prior is methodologically rigorous. Using SNIS to bypass the intractable partition function of the tilted distribution ensures the theoretical validity of the Boltzmann reweighting. The experimental design using NMR and X-ray benchmarks is thorough and demonstrates consistent empirical gains.
- The inference-time optimization, experiments-based design, and protein structure flexibility (ensembles) are all important questions in the science area and AI4Sci applications. The authors accurately capture those questions and handle with the help of existing strong methods such as AF3. The illustration of the conformational space is very clear.

Weaknesses:
- By rating Presentation of a "fair", I think the authors may consider moving some results to the main text.
- The computational overhead of the nested optimization loops (K outer loops and M inner loops) is a significant omission. AlphaFold3 inference is already resource-intensive; running multiple reverse SDE trajectories while continuously computing gradients through the structure module and force field likely introduces substantial latency. This proposed framework potentially costs even more than AF3.
- The convergence comparisons (Figure 7) highlight that standard guidance is limited to a fixed budget of 200 iterations, while IT-Opt benefits from multiple outer loops and cumulative refinement. The evaluation lacks a compute-matched baseline, making it difficult to disentangle whether IT-Opt's success stems from the representation-space paradigm itself or simply a vastly larger budget of forward/backward passes.
- The method updates $\mathcal{Z}$ using $\nabla_{\mathcal{Z}} \log p(y | D_\theta(\mathcal{X}; \mathcal{Z}, t))$. However, AF3's score network is highly sensitive to its conditioning. Pushing $\mathcal{Z}$ too far might result in out-of-distribution (OOD) latents that degrade the diffusion model's structural priors. The authors rely on a simple $L_{2}$ penalty to anchor the embeddings, but a more critical ablation on how much the manifold structure degrades under heavy optimization is missing.
- IT-Opt relies on Protenix (v0.2.0), for the majority of experiments, but uses the official AF3 implementation for the ipTM optimization. The potential domain shift or performance discrepancies between the replica and the official model are not discussed, which obscures whether some structural generation artifacts stem from the method or the replica implementation.
- The evaluation focuses heavily on short sequences, loops, and small peptides (e.g., a 13-residue bound peptide). It remains unclear how robust IT-Opt is when applied to large macromolecular complexes where embedding updates might introduce complex, non-local structural perturbations.
- Currently, IT-Opt compares its method against "Unguided AF3" and a "Coordinate-guided AF3" baseline. However, the community has already moved past unguided AF3 for ensemble generation. Models like AlphaFlow (which fine-tunes AF2 using flow matching) and BioEmu (a diffusion model for equilibrium ensembles) are explicitly designed to map the protein conformational landscape. If a structural biologist wants to resolve an NMR ensemble, why should they use the computationally heavy IT-Opt nested-loop framework over a rapid, out-of-the-box ensemble generator like AlphaFlow or BioEmu? I suggest that IT-Opt should run BioEmu and AFsample2 (or even ESMflow, optionally) on its NMR and X-ray benchmark targets. Even if these models are run without explicit experimental guidance, comparing them to IT-Opt would rigorously demonstrate the exact added value of inference-time experimental guidance. Alternatively, applying the coordinate-guidance baseline to AlphaFlow would create a true SOTA comparison.
- The authors claim their energy-reweighted framework produces "thermodynamically plausible" ensembles. However, satisfying macroscopic distance constraints (NOEs) or density maps does not guarantee physical realism at the atomic level. Recent benchmarking (e.g., ProteinConformers) demonstrates that while many generative models easily match $C_\alpha$ distance distributions, they frequently fail at producing physically valid inter-residue torsion angles (resulting in high Ramachandran outlier rates). Because IT-Opt updates the Pairformer embeddings using gradients that primarily target distances and coarse metrics, there is a high risk that the latent optimization degrades the strict stereochemical prior of AF3. The authors may report Ramachandran outlier percentages and perform a comprehensive torsional angle evaluation (e.g., using a metric like the Protein Conformation Plausibility Score) to prove the atomic-level physics remain intact.
- The current evaluation of ensemble diversity relies primarily on qualitative visualizations of bimodal loops (e.g., Figure 3, 3AZY). To substantiate the claim that IT-Opt successfully explores the manifold without collapsing to a single mode, the authors should adopt quantitative landscape-coverage metrics.
- The inference time is not new for the sampling of the conformational distributions (see ConforMix paper below), which greatly undermine the novelty of this work. ConforMix also should be considered adding into the performance comparison.


References:
- AFsample2 predicts multiple conformations and ensembles with AlphaFold2
- Scalable emulation of protein equilibrium ensembles with generative deep learning
- AlphaFold Meets Flow Matching for Generating Protein Ensembles
- ProteinConformers: Benchmark Dataset for Simulating Protein Conformational Landscape Diversity and Plausibility
- Structure language models for protein conformation generation.
- Unlocking hidden biomolecular conformational landscapes in diffusion models at inference time

---

> ### Author Rebuttal · Authors · 2026-03-31
>
> We thank the reviewer for their comments.
>
> > reorganize results
>
> Yes, we will move important baselines from the Appendix to main text for improved readability. Specifically, Tables 3-7 will be summarized with consolidated plots for improved readability.
>
> > runtime evaluation and compute-matched baseline
>
> While IT-Opt introduces additional overhead via outer-loop refinement, its per-iteration cost is comparable to coordinate guidance (https://postimg.cc/pywmrscJ). For a fair comparison, we include a compute-normalized evaluation (https://postimg.cc/7Jky8JX9), matching total denoiser calls between IT-Opt and guidance. At ~200 calls, both methods perform similarly; beyond this, guidance shows diminishing returns due to restarting from noise, whereas IT-Opt continues to improve by reusing optimized latent embeddings. Since current models are trained to produce static structures, iterative refinement is necessary to achieve maximum consistency with experimental data, aligning with standard structure refinement workflow.
>
> > ablation on how manifold structure degrades under heavy optimization
>
> Ablations over $\lambda_p$ (https://postimg.cc/rzZ9MqF9) show weak anchoring causes overfitting/collisions, while strong anchoring limits improvement; similar trends hold for learning rate. Similar trends are observed with learning rate. In addition to the embedding prior, validity losses (Line 1479) during optimization enforce physical plausibility. Hence, we do not observe degradation in structural validity.
>
> > AF3 replica artifacts
>
> To assess replica artifacts, we evaluate IT-Opt across Protenix, AF3, and Boltz2. We observe consistent improvements in experimental objectives (https://postimg.cc/mcGFt8dL, https://postimg.cc/SJMsCcYT, https://postimg.cc/w16jZMBY, and Figure 7) despite independent implementations. This indicates improvements and failure modes (e.g., confidence inflation) arise from the method (latent optimization), not replica-specific effects.
>
> > robustness to larger proteins
>
> We agree that evaluating on more complex systems is important. To partially address this, we include experiments on homologous proteins (≥280 residues), where AF3 exhibits mispredictions. IT-Opt successfully refines these structures toward experimentally consistent conformations (https://postimg.cc/DWCScC8C), indicating robustness to longer sequences and non-local structural corrections. While large multi-chain interfaces are beyond this work's scope, these results suggest latent optimization propagates corrections globally
>
> > Additional baselines
>
> We show an expanded benchmark on NMR (https://postimg.cc/T5nPCZfG) and X-ray targets (https://postimg.cc/MXYHfJPn), comparing against AFCluster, BioEmu, AlphaFlow, and ESMFlow. While these generate diverse ensembles efficiently, they lack experimental conditioning. IT-Opt consistently improves agreement with experimental observables, demonstrating the benefit of inference-time guidance.
>
> > latent optimization degrades stereochemical prior of AF3
>
> We preserve stereochemical quality via the embedding prior and validity regularizers (Line 1479), which constrain optimization to physically plausible regions of the AF3 manifold. The ProteinConformers plausibility score is designed for large MD ensembles and is not directly applicable in our setting. Instead, we use `phenix.molprobity`, a standard all-atom validation tool. As shown in (https://postimg.cc/Mvv47sFc), IT-Opt achieves lower clashscores, MolProbity scores, and Ramachandran outliers than AF3 and is comparable to PDB structures.
>
> > adopt quantitative landscape-coverage metrics
>
> While we agree on the value of quantitative metrics, the sheer size of protein systems makes exhaustively quantifying "landscape coverage" computationally intractable. Therefore, our evaluation targets PDB altlocs—highly curated, experimentally derived physical macrostates.
>
> > characterize the nature of these embedding shifts
>
> This is a very interesting question. In this work, our primary goal was to demonstrate the sensitivity of confidence metrics to small embedding perturbations, rather than to attribute these shifts to specific evolutionary or structural features. We are actively investigating the nature of these embedding changes in ongoing follow-up work.
>
> > Lack of novelty compared to ConforMix
>
> ConforMix uses coordinate-space guidance with an RMSD-based repulsive potential to promote diversity, without modifying learned representations or incorporating experimental constraints. In contrast, IT-Opt optimizes Pairformer embeddings using experimental likelihoods. Since ConforMix is not designed for experimental conditioning, a direct comparison is not fully applicable; in this setting, it reduces to unguided sampling, which is already included as a baseline. We will cite ConforMix and clarify these distinctions.
>
> References
>
> [1] MolProbity: all-atom contacts and structure validation for proteins and nucleic acids (2007)

---

> > ### Author Rebuttal · Reviewer_JDDy · 2026-04-02
> >
> > Thanks for the answers, which resolved my questions. I will raise my rating.

---

### Official Review · Reviewer_vXFn · 2026-03-11

**Soundness:** 4
**Presentation:** 3
**Significance:** 3
**Originality:** 3
**Overall Recommendation:** 4
**Confidence:** 3

**Summary:**

This paper introduces IT-Opt, a framework designed to improve AlphaFold3 (AF3) for experiment-grounded protein ensemble generation. Standard AF3 often struggles to produce structural ensembles that align with specific experimental constraints like cryo-EM, NMR and X-ray data. Instead of traditional steering in 3D coordinate space, the authors propose optimizing within the Pairformer latent embedding space. By treating AF3 as a learnable structural prior and incorporating Boltzmann-weighting via physical force fields, the method generates ensembles that are both experimentally consistent and thermodynamically plausible.

**Compliance With Llm Reviewing Policy:**

Affirmed.

**Final Justification:**

I would like to keep my positive rating to accept this paper without other major concerns.

**Key Questions For Authors:**

1. The paper mentions that IT-Opt uses nested optimization. For a structural biologist working with a standard workstation (e.g., a single A100 or 4090 GPU), how long does it take to optimize a medium-sized protein (e.g., 300 residues)?

2. If a user only has a small amount of experimental data (e.g., just a few NMR distance restraints), is there a risk that the model will "force" the protein into a weird shape just to satisfy those few points, while ignoring the rest of the protein's natural structure? How does the framework prevent the structure from becoming physically distorted when data is sparse?

**Limitations:**

Yes. The authors have appropriately identified the challenges in extending the framework to Cryo-EM data and acknowledged the potential for "confidence inflation" when optimizing latent representations.

**Strengths And Weaknesses:**

### Strengths
1. The core idea of shifting optimization from coordinate space to latent representation space is well-motivated. Coordinate-based steering often leads to structural distortion, but optimizing the embeddings leverages AF3's internal structural "logic," making the generation much more stable.

2. Many AI models generate "pretty" structures that fail basic physics, like having significant atomic clashes. By integrating the Amber99 force field for reweighting, this paper ensures the generated ensembles make sense from a thermodynamic perspective.

3. The empirical results across NMR and X-ray datasets are impressive. The framework successfully captures conformational heterogeneity, such as alternative side-chain conformations, which raw AF3 usually misses. Additionally, the analysis of "confidence inflation" provides a timely warning to the protein design community about the risks of over-optimizing confidence metrics like ipTM without actual structural improvement.

### Weaknesses
1. The most obvious drawback is the computational cost. The nested optimization loops (inner and outer) likely add significant overhead. It would be helpful to see exactly how much slower this is compared to standard AF3 sampling.

2. While it works well on medium proteins, the scalability to very large, multi-chain complexes—where the latent space becomes much more complex—remains to be fully proven with larger datasets. Would cryo-EM results be helpful in this case or not?

---

> ### Author Rebuttal · Authors · 2026-03-31
>
> We thank the reviewer for their comments.
>
> > Runtime
>
> We acknowledge that IT-Opt incurs additional cost due to its nested optimization; (https://postimg.cc/pywmrscJ; runtime analysis). Generating ensembles that strictly adhere to experimental data cannot be achieved through unguided generation alone. To ensure fairness, we perform a compute-normalized comparison (https://postimg.cc/7Jky8JX9), matching total denoiser calls between IT-Opt and coordinate guidance. At ~200 calls, performance is comparable; beyond this, guidance shows diminishing returns due to restarting from noise, whereas IT-Opt treats the embeddings as a persistent state, enabling accumulation of refinement across iterations. Importantly, IT-Opt is designed as a scientific workflow rather than a real-time application. Since these models are trained to predict static structures, iterative refinement is necessary to achieve consistency with experimental data, and aligns with standard practices in structural biology.
>
> > X-ray structure fitting time
>
> We agree that scaling to large multi-chain complexes is an important direction. While a comprehensive study on cryo-EM is beyond the scope of this work, we include experiments on a pair of homologous proteins (each >280 residues), where AF3 exhibits mispredictions. Without providing any pre-modeled structures (i.e. optimization happens on the full density), IT-Opt successfully refines these structures toward experimentally consistent conformations in <1 hour (https://postimg.cc/DWCScC8C).
>
> > Behavior under sparse experimental constraints
>
> IT-Opt operates in the Pairformer embedding space, which is constrained by the AF3 diffusion prior. To prevent arbitrary distortions under sparse supervision, we stabilize optimization using (i) _validity regularizers_ (Line 1479) to penalize physically implausible conformations, and (ii) _an embedding prior_, to anchor each optimized embedding to its initialization and prevent adversarial deviations.
>
> To evaluate robustness, we subsample NOE restraints for 2B5B (10–100% retention; https://postimg.cc/Y4c08HPK). Across all sparsity levels, the validity loss remains comparable to the PDB ensemble (validity: 0.002), indicating that structures remain physically consistent.

---

> > ### Author Rebuttal · Reviewer_vXFn · 2026-04-03
> >
> > I appreciate authors' rebuttal. My concerns have been resolved.

---

### Official Review · Reviewer_uE5Y · 2026-03-11

**Soundness:** 2
**Presentation:** 2
**Significance:** 2
**Originality:** 3
**Overall Recommendation:** 3
**Confidence:** 3

**Summary:**

The paper proposes an inference-time optimization (IT-Opt) framework for experiment-grounded ensemble generation with AlphaFold3-like diffusion models. Instead of applying guidance in coordinate space during the reverse SDE, the method optimizes AlphaFold’s conditioning embeddings (Pairformer trunk) to maximize an ELBO-style surrogate of ensemble log-likelihood, and augments sampling with Boltzmann reweighting using force-field energies to yield thermodynamically plausible ensembles. Empirically, the approach reports improved agreement with X-ray electron density and NMR NOE data compared to coordinate-space guidance baselines, and a systematic study shows that ipTM can be artificially inflated by tiny latent perturbations without guaranteed accuracy gains.

**Compliance With Llm Reviewing Policy:**

Affirmed.

**Final Justification:**

The rebuttal provides some additional information, but overall it does not remove my doubts about the paper’s main contributions. At this stage, I am still not convinced that the evidence presented is strong enough to support the claimed significant improvements, and these issues are not suitable for resolution through a brief rebuttal alone.

**In addition, using links to display experimental results is not compliant with the regulations, because it will exceed the word limit.**

**Key Questions For Authors:**

* What are the sensitivities to key hyperparameters (embedding prior variance σ^2, λp, learning rates, K outer loops, ensemble size n)? Please include ablations and compute-normalized comparisons against coordinate guidance (same total denoiser calls).
* For Boltzmann reweighting, what are the effective sample sizes and weight variances across datasets? How do results change with β, energy smoothing, and annealing? Are energies all-atom and clash-minimized before evaluation? Any MolProbity/clashscore statistics?
* In the ipTM study, how were confidence gains calibrated to structural accuracy across the benchmark (e.g., DockQ or contact recovery vs ipTM under increasing perturbation budgets)? Can you quantify average correlation pre- vs post-optimization and report cases of large confidence increases with no accuracy gain?
* In the ipTM study, how were confidence gains calibrated to structural accuracy across the benchmark (e.g., DockQ or contact recovery vs ipTM under increasing perturbation budgets)? Can you quantify average correlation pre- vs post-optimization and report cases of large confidence increases with no accuracy gain?
* How does IT-Opt perform on lower-resolution or noisier density maps and on sparse/incomplete NOE sets? Does the method overfit to artifacts in these regimes, and can the embedding prior mitigate this?
* Are the AF3 internals necessary for reproduction, or can the method be run fully on Protenix or other open backbones? Will code, data, and configuration files be released to enable replication?

**Limitations:**

yes

**Strengths And Weaknesses:**

# Strengths
* Introduces a nested inference-time optimization scheme that updates conditioning embeddings across diffusion trajectories, decoupling conditioning from the specific reverse SDE schedule and reducing sensitivity to initialization.
* Integrates Boltzmann reweighting via self-normalized importance sampling to combine the AF3 structural prior with differentiable force-field energies, moving beyond uniform ensembles to thermodynamically plausible populations.
* Provides an informative analysis of optimizing learned confidence metrics (ipTM), demonstrating susceptibility to small embedding-space manipulations—of practical importance for binder design workflows.
* Evidence of improved stability and reduced variance across random seeds relative to coordinate-space guidance, and qualitative examples where multimodal density is better recovered.
* Addresses a central limitation in current experiment-guided protein generative pipelines—sensitivity to fixed diffusion horizons and initialization—while providing a general mechanism applicable across modalities.
# Weaknesses
* The variational relaxation uses a simple Gaussian prior p(Z|a) around AF3 embeddings; the choice of variance and its effect on drift/off-manifold behavior is under-specified and may be critical to stability and generalization.
* Boltzmann reweighting via SNIS can suffer from high-variance weights for small ensemble sizes; the paper lacks diagnostics on effective sample size or variance control, and potential double-counting/competition between AF3 priors and the energy model is not discussed.
* No theoretical convergence or monotonicity guarantees for the nested optimization are provided (beyond Jensen), and the impact of denoiser stochasticity on gradient estimators is not analyzed.

---

> ### Author Rebuttal · Authors · 2026-03-31
>
> We thank the reviewer for their comments.
>
> > Gaussian prior
>
> For X-ray and ipTM, we use a Gaussian prior over the embeddings, while for NMR we employ a GMM prior derived from embeddings of subsampled MSAs [1] (App. B.5.1). The GMM provides a flexible, general representation of the embedding distribution and thus does not restrict the formulation. We impose this structure to improve stability and convergence [1]. A discussion will be included in the final manuscript.
>
> > Theoretical justifications
>
> We study the convergence properties of inference-time updates to the diffusion model’s input embeddings. We assume local Lipschitz smoothness of the surrogate objective (diffusion-trajectory–averaged log-likelihood) and bounded inner-loop updates. Under a Gaussian prior and gradient clipping, these conditions are reasonable and confine the optimization to a compact region with finite curvature. We formalize IT-Opt as biased stochastic gradient ascent over a surrogate objective consistent with our implementation. The dominant bias is bounded via Cauchy–Schwarz in terms of the Fisher information trace, while estimator variance is controlled through inner-loop early stopping. Despite the bias, we establish an O(1/K) convergence rate to a near-stationary region (with K outer iterations) and derive a sufficient condition for expected monotonic ascent (https://postimg.cc/4KwX6r7D, https://postimg.cc/qgW4xcqM, https://postimg.cc/2qjDS2cH, https://postimg.cc/kRSrpZGq, https://postimg.cc/fJxn7664). This analysis will be included in the revised version of the paper.
>
> > Compute normalization
>
> We include a compute-normalized comparison against coordinate guidance, matching total denoiser calls (https://postimg.cc/7Jky8JX9; normalized runtime table). At an equivalent budget (~200 function evaluations), performance is comparable. Beyond this point, however, standard guidance exhibits diminishing returns because each seed restarts from noise in a memoryless process. This indicates that gains arise from more efficient use of compute rather than increased budget alone.
>
> > IT-opt hyperparams
>
> In practice $\sigma^2$ is absorbed into $\lambda_p$, and we treat it as a single parameter. We ablate $\lambda_p$, η, $K$, and $n$ (https://postimg.cc/rzZ9MqF9).
>
> We observe (i) strong $\lambda_p$ limits optimization, while weak $\lambda_p$ leads to overfitting; (ii) small η slows convergence, whereas large η introduces unphysical structures; (iii) both $K$ and $n$ improve performance up to point with diminishing returns.
>
> > ESS/energy ablation
>
> All-atom energies are evaluated post-relaxation. IT-Opt achieves higher ESS (300K) than uniform weighting (https://postimg.cc/478GdYYk; ESS). Ablations over temperature, annealing, and smoothing (https://postimg.cc/JsKPSZv0) show that IT-Opt yields a higher ESS, and a better accuracy.
>
> > Molprobity
>
> Using `phenix.molprobity` on Xray (https://postimg.cc/Mvv47sFc)[3], IT-Opt achieves better clashscores and MolProbity scores than AF3 and is comparable to PDB.
>
> > sparse NOEs
>
> We randomly subsample restraints for 2B5B (10-100% retention; https://postimg.cc/Y4c08HPK). Across levels, the validity loss (line 1479 of manuscript) is comparable to the PDB ensemble (validity: 0.002), and we do not observe overfitting to sparse constraints (e.g., distorted geometries).
>
> > limited to Protenix?
>
> IT-Opt operates in Pairformer embedding space and applies to any differentiable generative AF3-like model. We demonstrate this across Protenix, AF3, and Boltz2.
>
> IT-Opt in Boltz2 exhibits improvements in both NMR (https://postimg.cc/SJMsCcYT) and iPTM (https://postimg.cc/mcGFt8dL) objectives, matching Fig. 7's trends. Similar behavior is observed also in Protenix (https://postimg.cc/w16jZMBY).
> This shows that IT-Opt is not model specific. We will release the code as a unified framework that can work with any AF3-like models.
>
> > iPTM
>
> To address confidence–accuracy calibration, we evaluated 7 benchmark complexes at a perturbation budget of 0.1, using DockQ, Fnat, and H-bond recovery to assess structural quality. ipTM-guided IT-Opt uniformly increases confidence (0.8×ipTM+0.2×pTM; Δ=+0.10 to +0.74, all >0.96), but structural accuracy is largely unchanged in 6/7 cases (ΔDockQ: −0.04 to +0.01), directly demonstrating decoupling between confidence and accuracy. Importantly, no substantial degradation is observed (min ΔDockQ=−0.04; https://postimg.cc/K3DCxQ8Z). In the single improving case (1YCS), confidence gains align with genuine structural recovery (ΔDockQ=+0.53, Fnat=0.69, H-bond=0.46). Overall, these results show that ipTM can be inflated via minor (~0.1%) embedding perturbations without corresponding structural gains, suggesting potential false positives when used for ranking or optimization.
>
> Refs
> [1] Predicting multiple conformations via sequence clustering and AlphaFold2 (2024)
> [2] Simple statistical gradient-following algorithms for connectionist reinforcement learning (1992)
> [3] MolProbity for the masses-of data (2015)

---

> > ### Author Rebuttal · Reviewer_uE5Y · 2026-04-02
> >
> > I thank the authors for the comprehensive rebuttal. Two points remain open: (1) all additional results (ablations, convergence proofs, MolProbity, ipTM calibration) are provided via external image links rather than the manuscript — please confirm these will be fully incorporated in the revision; (2) the reported MolProbity scores and clashscores of IT-Opt surpassing PDB reference structures is surprising, as PDB represents experimentally determined ground truth — does this reflect low resolution or poorly refined entries in the test set, or is the metric not a reliable proxy for structural accuracy here? This result requires explicit discussion to avoid overclaiming.

---

> > > ### Author Response · Authors · 2026-04-03
> > >
> > > We thank the reviewer for their response.
> > >
> > > 1. We confirm that all additional results will be fully incorporated into the final manuscript. We provided these via external links currently due to conference restrictions on updating the manuscript during the rebuttal phase.
> > >
> > > 2. We note that deposited PDB coordinates are models derived from the ground truth signal of raw experimental observations, such as electron density maps or NMR restraints. Our case studies—specifically protein-bound peptides and loops with alternative conformations—often exhibit significant local disorder or conformational heterogeneity. In these regimes, traditional model fitting can be subject to localized ambiguity, where manually refined models may occasionally retain minor stereochemical tensions or struggle to resolve precise atomic positions. IT-Opt leverages the structural prior from AlphaFold (AF) to navigate this uncertainty, identifying conformations that are both faithful to the experimental signal and physically plausible. This is evidenced by the competitive metrics achieved by IT-Opt relative to original deposited PDB entries, notably in NMR restraint satisfaction (e.g., PDBs 1D3Z and 2K0M in Table 3) and X-ray model fitting (e.g., PDBs 6QQF and 1CKB in Table 5). Furthermore, while clashscores necessarily vary depending on the original refinement quality of a given deposition, the use of an AF-based prior can help mitigate atomic overlaps that may persist in legacy PDB entries. This suggests that our framework can yield models that are consistent with the experimental signal without distorting stereochemical priors.

---

### Official Review · Reviewer_ecTA · 2026-03-13

**Soundness:** 3
**Presentation:** 3
**Significance:** 3
**Originality:** 3
**Overall Recommendation:** 4
**Confidence:** 4

**Summary:**

This work focuses on protein ensemble generation via a nested two stage inference only approach called IT-Optimization on AlphaFold3. The first component leverages a latent pairformer optimization to maximize log likelihood of the structure prediction then enhances the conditioning with weighted diffusion sampling. This work also shows how perturbing MSA embeddings can artificially inflate interface confidence scores. IT-Optimization is evaluated on NMR-based, x-ray-based, and ipTM-based guidance via their meta optimization framework showing strong results compared to the base model and prior guidance methods.

**Compliance With Llm Reviewing Policy:**

Affirmed.

**Final Justification:**

The rebuttal addressed my main concerns so I maintain my score.

**Key Questions For Authors:**

1. How does the run time compare for each methods? AF3 vs AlphaFlow vs AF3 + Guidance vs AF3 + IT-Opt (w + w/out energy?
2. Similar how does compute normalization impact the improvements?
3. What is the significance and error bars for Table 5-7?

**Limitations:**

yes

**Strengths And Weaknesses:**

Strengths
- Well motivated and inner sections clearly written.
- This work shows a novel extension of structure prediction to mitigate issues in non dynamic models.
- Improvements across multiple tasks with different conditioning. Clear improvement across all baselines.

Weaknesses
- Most of the benchmarks are in the Appendix. Paper is well written but not structured in an easy to digest manner. Results in a lot of back and forth to understand the impacts.
- The paper describes AF3 but in 2/3 experiments Protenix was used. These models are similar but perform quite differently in practice. These details should be made more clear outside the appendix. Results would benefit by cross evaluating with Boltz/RF3 to more rigorously asses generalizability.
- How does the number for function evaluations compare between guidance and IT-Optimization? Are the improvements in performance due to just sampling more or the improved guidance itself?

---

> ### Author Rebuttal · Authors · 2026-03-31
>
> We thank the reviewer for their comments.
>
> > Most of the benchmarks are in the Appendix...results in back and forth
>
> We agree that the current organization can be improved. We will augment Tables 3–7 with consolidated violin/box plots to summarize the quantitative results (Tables 3-4 -- NMR, and Tables 5–7 -- X-ray crystallography) and include them in the main paper.
>
> > AF3 vs Protenix usage and generalizability
>
> We appreciate this point and will clarify model usage explicitly in the main paper (including per-experiment labeling). While some experiments use Protenix, our method is not tied to a specific implementation, and can be applied to any sequence-conditioned generative model (e.g. AlphaFold3, Boltz-2). To demonstrate this, we implemented IT-Opt within the Boltz-2 framework. As shown in the NMR & iPTM convergence plots (https://postimg.cc/mcGFt8dL; Boltz iPTM, https://postimg.cc/SJMsCcYT; Boltz NMR), iterative latent refinement improves both NMR restraint satisfaction and iPTM-based objectives in Boltz2, analogous to the trends observed with AF3/Protenix (Figure 7). We also notice similar iPTM trends with Protenix (https://postimg.cc/w16jZMBY; Protenix iPTM). This suggests that IT-Opt operates on architectural properties of AF3-like models rather than model-specific behaviors. We also clarify our model choices: AF3 was used for iPTM-based experiments, as it provides well-established confidence metrics for complex prediction [1], while Protenix was used in other settings to enable direct comparison with prior guidance baselines [2].
>
> We view the Boltz2 results as a proof-of-concept demonstrating that IT-Opt extends beyond a single implementation. In the future, we plan to release a unified implementation of IT-Opt and systematically benchmark it across multiple diffusion-based structure prediction models to further evaluate generality.
>
> > significance and error bars for Table 5-7
>
> We have computed standard deviations across 5 independent runs (different random seeds) for all entries in Tables 5-7 (https://postimg.cc/N9knWJnV; error bars Xray). These show that IT-Opt not only improves mean performance but also exhibits lower variance compared to guidance, indicating greater robustness to initialization.
>
> > Runtime and compute normalization
>
> We acknowledge that IT-Opt involves a higher total computational budget due to its iterative structure (https://postimg.cc/pywmrscJ; runtime analysis). Generating ensembles that strictly adhere to experimental data cannot be achieved through unguided generation alone. Since the underlying models were trained to predict static structures, IT-Opt's iterative approach is essential to achieve this fidelity. To address fairness, we performed a compute-normalized comparison between IT-Opt and standard coordinate-based guidance (https://postimg.cc/7Jky8JX9; compute-normalized runtime). At an equivalent budget (~200 function evaluations), performance is comparable. Beyond this point, however, standard guidance exhibits diminishing returns because each seed restarts from noise in a memoryless process. In contrast, IT-Opt continues to improve by refining and reusing optimized latent embeddings across iterations. This demonstrates that the gains are not merely due to increased sampling, but stem from a more effective optimization strategy that preserves and accumulates information across iterations.
>
> Furthermore, we observe that, in several examples, even a second outer iteration of IT-Opt consistently outperforms coordinate-based guidance. While this requires additional compute, it is important to note that IT-Opt is not an interactive application (such as an LLM or text-guided diffusion model) that requires real-time latency, but rather a rigorous scientific workflow. In structural biology, consistently guaranteeing better adherence to experimental restraints is the primary objective, fully justifying the computational cost. Furthermore, iterative refinement of the modeled structure is standard in both crystallographic and NMR structure resolution workflows [3, 4]. Despite involving multiple runs, our method significantly accelerates the overall structure resolution process compared to traditional experimental pipelines, bringing substantial value to these communities. Finally, this computational footprint aligns with the broader field, as other state-of-the-art conformational ensemble generators (e.g., AlphaFlow, BioEmu) similarly rely on running several generative iterations.
>
> References
>
> [1] Assessing scoring metrics for AlphaFold2 and AlphaFold3 protein complex predictions (2025)
>
> [2] Experiment-guided AlphaFold3 resolves accurate protein ensembles (2026)
>
> [3] Using NMR Chemical Shifts and Cryo-EM Density Restraints in Iterative Rosetta-MD Protein Structure Refinement (2019)
>
> [4] Improving macromolecular atomic models at moderate resolution by automated iterative model building, statistical density modification and refinement (2003)

---

> > ### Author Rebuttal · Reviewer_ecTA · 2026-04-01
> >
> > Thank you for answering my questions.

---

### Decision · Program_Chairs · 2026-04-30

**Decision:**

Accept (regular)

**Comment:**

This paper introduces an effective inference-time optimization framework that operates within the latent embedding space of diffusion-based structural models to generate thermodynamically plausible protein ensembles. The reviewers unanimously recognized the technical rigor and novelty of the approach, and the authors provided a comprehensive rebuttal that successfully resolved initial concerns regarding computational overhead, baseline comparisons, and cross-model generalizability.  This work provides a significant contribution to computational structural biology and merits acceptance.